# GRADIENT-FREE GENERATION FOR HARD-CONSTRAINED SYSTEMS

**Chaoran Cheng**[1]*, **Boran Han**[2], **Danielle C. Maddix**[2], **Abdul Fatir Ansari**[2], **Andrew Stuart**[3], **Michael W. Mahoney**[4], **Yuyang Wang**[2]
[1]University of Illinois Urbana-Champaign
[2]Amazon Web Services
[3]Stores Foundational AI, Amazon
[4]Amazon SCOT
chaoran7@illinois.edu
{boranhan,dmmaddix,ansarnd,andrxstu,zmahmich,yuyawang}@amazon.com

## ABSTRACT

Generative models that satisfy hard constraints are critical in many scientific and engineering applications, where physical laws or system requirements must be strictly respected. Many existing constrained generative models, especially those developed for computer vision, rely heavily on gradient information, which is often sparse or computationally expensive in some fields, e.g., partial differential equations (PDEs). In this work, we introduce a novel framework for adapting pre-trained, unconstrained flow-matching models to satisfy constraints exactly in a zero-shot manner without requiring expensive gradient computations or fine-tuning. Our framework, *ECI sampling*, alternates between extrapolation (E), correction (C), and interpolation (I) stages during each iterative sampling step of flow matching sampling to ensure accurate integration of constraint information while preserving the validity of the generation. We demonstrate the effectiveness of our approach across various PDE systems, showing that ECI-guided generation strictly adheres to physical constraints and accurately captures complex distribution shifts induced by these constraints. Empirical results demonstrate that our framework consistently outperforms baseline approaches in various zero-shot constrained generation tasks and also achieves competitive results in the regression tasks without additional fine-tuning. Our code is available at https://github.com/amazon-science/ECI-sampling.

## 1 INTRODUCTION

Diffusion and flow matching models have achieved remarkable success in generative tasks of image generation (Ho et al., 2020; Esser et al., 2024a), language modeling (Lou et al., 2023; Gat et al., 2024), time series prediction (Lin et al., 2024; Kollovieh et al., 2024), and functional data modeling (Lim et al., 2023; Kerrigan et al., 2023). Constrained generation built upon these generative models for solving various inverse problems has also been explored in the image domain (Kawar et al., 2022; Ben-Hamu et al., 2024). These approaches predominantly rely on gradient information with respect to some cost function as a soft constraint. While such soft-constrained methods have been successful in the image domain, applications in other domains often require the generation to adhere to certain constraints strictly. Such hard-constrained generation, i.e., generative modeling where the natural hard constraints are *requirements* and not just *suggestions*, is crucial for tasks in many scientific domains. For example, in scientific computing, numerical simulations often require generated solutions to adhere to specific physical constraints (energy or mass conservation (Hansen et al., 2023; Mouli et al., 2024)) and satisfy boundary condition constraints on the values or derivatives of the solutions (Saad et al., 2023). An intuitive example of the generation of PDE solutions is demonstrated in Figure 1, where the solution set narrows and shifts when the constraint is imposed. While existing approaches for constrained generation in image inverse problems can always be adapted

---

*Work done during internship at AWS.

(Esser et al., 2024b; Ansari et al., 2024), hard-constrained generation for complex systems like PDE solutions presents the following challenges:

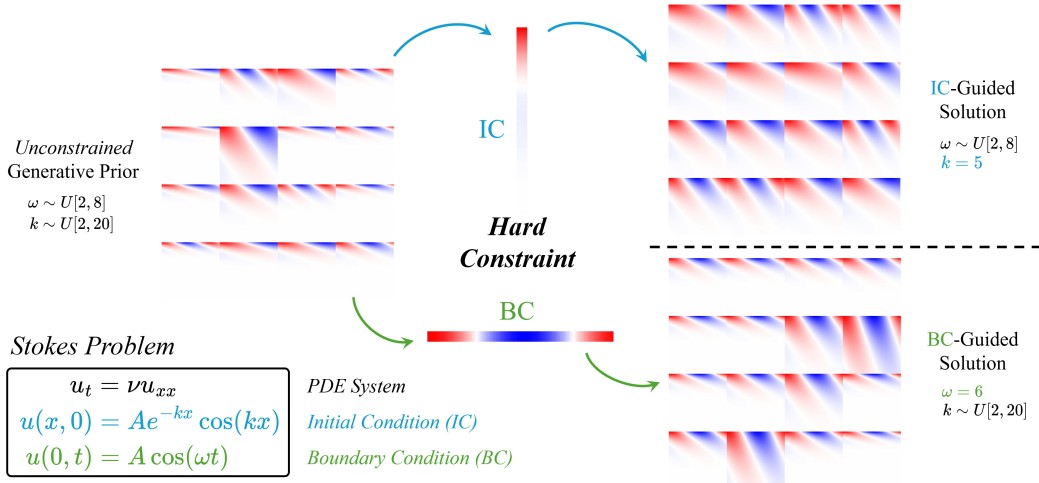

Figure 1: Hard-constrained generation of PDE solutions with the boundary condition (BC) or initial condition (IC) prescribed *a posteriori*. $\omega, k$ are PDE parameters that determine the IC and BC.

**Scarcity of constraint information.** In contrast to computer vision (CV) tasks, e.g., image inpainting, which typically assumes a considerable amount of context information, common constraints like BC and IC in Figure 1 have measure zero with respect to the spatiotemporal domain. For example, a standard setting with a spatiotemporal resolution of $100 \times 100$ leads to only $1\%$ of known pixel values as context. This is significantly less than the context provided in a typical CV application.

**Exact constraint satisfaction.** The exact conservation of mass, energy, or momentum is often essential in the simulation of PDEs (LeVeque, 1990) for ensuring physically feasible and consistent solutions. This is in contrast with inverse problems in CV, e.g., super-resolution and de-blurring, where constraints are often implicitly defined as modeling assumptions and evaluation metrics, e.g., peak signal-to-noise ratio (PSNR) do not directly depend on the exact satisfaction of these constraints. Existing CV approaches fail to guarantee exact satisfaction of constraints.

**Issues with gradient-based methods.** Existing zero-shot frameworks for constrained generation predominantly rely on gradient guidance from a differentiable cost function on the final step of generation. This information can be prohibitively expensive, especially in large 3D PDE spatiotemporal systems. Previous work has also indicated the drawbacks of these soft-constrained gradient-based approaches. Well-known limitations include gradient imbalances in the loss terms (Wang et al., 2020; 2021) and ill-conditioning (Krishnapriyan et al., 2021), which can lead to failure modes in scientific machine learning (SciML) tasks.

To address these challenges for hard-constrained generation, we propose a general framework that adopts a gradient-free and zero-shot approach for guiding unconstrained pre-trained flow matching models. For the diversity of practical hard constraints for PDE systems, our proposed framework provides a unified and efficient approach without the need for expensive fine-tuning or gradient backpropagation. We term our framework *ECI sampling* since it interleaves *extrapolation (E)*, *correction (C)*, and *interpolation (I)* stages at each iterative sampling step. ECI sampling effectively and accurately captures the distribution shift imposed by the constraints and maintains the consistency of the generative prior. We summarize our main contributions as follows:

- **Unified gradient-free generation framework.** We introduce ECI sampling, a unified gradient-free sampling framework for guiding an unconstrained pre-trained flow matching model. By interleaving extrapolation, correction, and interpolation stages at each iterative sampling step, ECI offers fine-grained iterative control over the flow sampling that can accurately capture the distribution shift imposed by the constraints and maintain the consistency of the system.

- **Exact and efficient satisfaction of hard constraints.** ECI sampling addresses the unique challenges imposed by hard constraints. The memory- and time-efficient gradient-free approach guarantees the exact satisfaction of constraints and mitigates gradient issues known with existing gradient-based methods in CV domains.
- **Zero-shot performance on generative and regression tasks.** Comprehensive generative metrics of distributional properties manifest the superior generative performance of our ECI sampling compared with various existing zero-shot guidance methods on various PDE systems. We also show that ECI sampling can be applied to zero-shot regression tasks, still achieving competitive performance with state-of-the-art Neural Operators (NOs) (Li et al., 2020a) that are directly trained on the regression tasks.

## 2 RELATED WORK

**Diffusion and Flow Matching Models.** We first review existing generative models for functional data that serve as the generative prior for the spatiotemporal solutions in our approach. Diffusion models (Song & Ermon, 2020; Ho et al., 2020; Song et al., 2020) rely on a variational bound for the log-likelihood and learn a reverse diffusion process to transform prior noise into meaningful samples. Lim et al. (2023) propose the diffusion denoising operator (DDO) to extend diffusion models to function spaces. Flow matching models (Lipman et al., 2022; Liu et al., 2022) are a family of continuous normalizing flows that learn a time-dependent vector field that defines the data dynamics via a flow ordinary differential equations (ODEs). Kerrigan et al. (2023) proposes functional flow matching (FFM) as an extension of existing flow matching models for image generation. Compared to DDO, FFM has a more concise mathematical formulation and better empirical generation results. We also note the close connection between diffusion and flow models in Lipman et al. (2022); Albergo et al. (2023). Therefore, while we focus on guiding flow-based models in this work, we also compare our approach with a wide range of diffusion-based methods.

**Constrained Generation.** Flow-based generative modeling in function spaces has not been explored until recently, and constrained generation for PDE systems remains largely unexplored. The neural process (NP) (Garnelo et al., 2018; Kim et al., 2019; Sitzmann et al., 2020) is one traditional constrained generation approach that learns to map a context set of observed input-output pairs to a distribution over regression functions. Hansen et al. (2023) introduce a generic model-agnostic approach to control the variance of the generation and enjoy the exact satisfaction of constraints. Lippe et al. (2024) use gradient guidance from the constraint to guide each sampling step of the pre-trained diffusion model in a zero-shot fashion, similar to diffusion posterior sampling in Chung et al. (2022). Huang et al. (2024) further incorporate the gradients from physics-informed losses (Shu et al., 2023). Other approaches focus on controlling NOs in regression tasks (Négiar et al., 2023; Saad et al., 2023; Li et al., 2024; Mouli et al., 2024). As the sampling for flow or diffusion models is an iterative procedure, these methods cannot be directly adopted to provide iterative control. They can be, however, incorporated in our correction stage (see Section 3.2).

**Inverse Problems.** Zero-shot generation for pre-trained diffusion and flow models has been explored in the image domain for solving various inverse problems, including image inpainting, deblurring, and superresolution (Bai et al., 2020). Existing approaches predominantly rely on gradient guidance. Liu et al. (2023b); Ben-Hamu et al. (2024); Wang et al. (2024) propose to modify the prior noise or vector field by differentiating through the ODE solver, thus being extremely time- and memory-consuming. Pan et al. (2023a;b) use the adjoint sensitivity method to mitigate the high memory assumption. Other works propose gradient-free control, usually with strong prior assumptions on the constraints to derive control at each sampling step. Lugmayr et al. (2022) propose to mix forward diffusion steps from the context with the model's prediction, and Kawar et al. (2022) use a similar variational approach to solve linear inverse problems.

Although these existing methods have achieved decent generation results for inverse problems in image generation, the approximate enforcement of physical laws (soft-constrained) (Négiar et al., 2023) failed to address the challenges imposed by hard constraints (Wang et al., 2020; 2021; Krishnapriyan et al., 2021). Our proposed zero-shot guidance approach focuses on the challenges of exact constraint satisfaction, using fine-grain iterative control at each sampling step for more consistent generations. The gradient-free approach is also more efficient and provides a unified framework for various linear and non-linear constraints. We summarize the major differences between the existing methods and our proposed method in Table 1.

Table 1: Comparison between existing constrained generation methods and our ECI sampling.

| | Zero-shot | Gradient-free | Exact constraint | Iterative control |
|---|---|---|---|---|
| Conditional FFM [18] | ✗ | ✓ | ✗ | ✓ |
| ANP [19] | ✓ | ✓ | ✗ | ✗ |
| ProbConserv [13] | ✓ | ✓ | ✓ | ✗ |
| DiffusionPDE [16] | ✓ | ✗ | ✗ | ✓ |
| D-Flow [4] | ✓ | ✗ | ✗ | ✓ |
| ECI (ours) | ✓ | ✓ | ✓ | ✓ |

## 3 METHOD

### 3.1 PRELIMINARY

**Problem Definition.** Consider a PDE system $\mathcal{F}_\phi u(x) = 0, x \in \mathcal{X} \subseteq \mathbb{R}^D$ with PDE parameters $\phi \in \Phi$. For a fixed PDE family $\mathcal{F}$, let $\mathcal{U}_\mathcal{F} = \{u(x) : \exists \phi \in \Phi, \mathcal{F}_\phi u(x) = 0, x \in \mathcal{X}\}$ denote the set of all the plausible PDE solutions by varying the PDE parameters. Consider some constraint operator $\mathcal{G}u(x) = 0, x \in \mathcal{X}_\mathcal{G} \subseteq \mathcal{X}$ defined on a subset of the PDE domain, and let $\mathcal{U}_\mathcal{G} = \{u(x) : \mathcal{G}u(x) = 0, x \in \mathcal{X}_\mathcal{G}\}$ denote the solution set for the constraint. We are interested in finding a subset of solutions $\mathcal{U}_{\mathcal{F}|\mathcal{G}} := \mathcal{U}_\mathcal{F} \cap \mathcal{U}_\mathcal{G} \subseteq \mathcal{U}_\mathcal{F}$ in which both the original PDE and the constraint are satisfied. We assume that the prior solution set $\mathcal{U}_\mathcal{F}$ can be captured by a generative model, e.g., an FFM pretrained on a solution set. We aim to guide the pre-trained model to satisfy the constraint operator in a zero-shot fashion towards the narrowed solution set $\mathcal{U}_{\mathcal{F}|\mathcal{G}}$. A more mathematically rigorous definition using measure theoretic terms is provided in Appendix A.1.

**Prior Generative Model.** FFM (Kerrigan et al., 2023) extends the flow matching framework (Lipman et al., 2022) to model measures over the Hilbert space of continuous functions. Given a set $\mathcal{U}$ of well-behaved (see Appendix C.2) functions $u : \mathcal{X} \to \mathbb{R}$, FFM learns a time-dependent vector field operator $v_t : \mathcal{U} \times [0, 1] \to \mathcal{U}$ that defines a time-dependent diffeomorphism $\psi_t : \mathcal{U} \times [0, 1] \to \mathcal{U}$ called the *flow* via the *flow ODE*:

$$\partial_t \psi_t(u) = v_t(\psi_t(u)), \quad \psi_0(u) = u_0. \tag{1}$$

The flow $\psi_t$ induces a pushforward measure $\hat{\mu}_t := (\psi_t)_* \mu_0$, where $\mu_0$ is the prior noise measure from which $u_0$ can be sampled. FFM showed that a tractable flow matching objective can be derived when the flow is conditioned on the target function $u_1$ sampled from the target measure $\mu_1$. As we are interested in guiding pre-trained generative models, we will assume that FFM can well approximate the target measure $\hat{\mu}_1 \approx \mu_1$ over $\mathcal{U}_\mathcal{F}$. We want to guide the pre-trained FFM towards the conditional measure $\mu_{\mathcal{F}|\mathcal{G}}$ over $\mathcal{U}_{\mathcal{F}|\mathcal{G}}$. The sampling procedure of the FFM, similar to all diffusion and flow models, is an iterative process in which the initial noise function is iteratively refined into target functions via the learned vector field with the dynamics described in Equation 1. Algorithm 1 describes sampling from the FFM method using the Euler method. In practice, the vector field can be parameterized by some discretization-invariant NOs (Lu et al., 2019; Li et al., 2020b;a) such that the whole generative framework is discretization-invariant. This indicates that FFM can be naturally adapted for zero-shot superresolution (Kerrigan et al., 2023).

---

**Algorithm 1** Sampling from FFM (Euler Method)

1: **Input:** Learned vector field $v_\theta$, Euler steps $N$.
2: Sample noise function $u_0 \sim \mu_0(u)$.
3: **for** $t \leftarrow 0, 1/N, 2/N, \ldots, (N-1)/N$ **do**
4: $\quad u_{t+1/N} \leftarrow u_t + v_\theta(u_t, t)/N$
5: **return** $u_1$

---

### 3.2 HARD CONSTRAINT INDUCED GUIDANCE

Although various regression NOs have been proposed to satisfy specific constraints in their prediction (Liu et al., 2023a; Négiar et al., 2023; Saad et al., 2023; Liu et al., 2024; Mouli et al., 2024), the difficulty in applying these existing methods to flow matching models lies in the iterative nature of the flow sampling in Algorithm 1. For flow matching sampling, intermediate generations are noised data instead of final predictions, making the correction algorithm inapplicable as the constraint can only be applied to the last generation step.

Following previous zero-shot guidance frameworks (Lugmayr et al., 2022; Kawar et al., 2022), at each iterative sampling timestep $t$, we model the constrained generation process with the conditional generation probability $p(\hat{u}_t|\mathcal{G}) = p(\hat{u}_t|u_t, \mathcal{G})p(u_t)$, where the unconditional probability $p(u_t)$ can be modeled by a pre-trained generative model. We assume a correction algorithm $C(u_1, \mathcal{G})$ is readily available for *final predictions* that uses an orthogonal/oblique projection (Hansen et al., 2023) to ensure hard constraint satisfaction (see Appendix A.3). To offer iterative control over the sampling process for hard constraints, we note that if the final prediction can be *extrapolated* from intermediate noised generations, the correction can be safely applied. Furthermore, if intermediate noised generations can be *interpolated* back from the corrected prediction, we can "propagate" the constraint information back to each time step. Indeed, let $p_\theta(u_1|u_t)$ denote the probability of extrapolating the final prediction of $u_1$, given the current noise data $u_t$ and the learnable parameter $\theta$; and let $q(\hat{u}_t|\hat{u}_1)$ denote the probability of interpolating the intermediate noised data $\hat{u}_t$, given the corrected data $\hat{u}_1 = C(u_1, \mathcal{G})$; then the constraint-conditional probability can be decomposed in a variational way by marginalizing over the auxiliary variable $u_1$:

$$p(\hat{u}_t|u_t, \mathcal{G}) = \mathbb{E}_{u_1 \sim p_\theta(u_1|u_t)}[q(\hat{u}_t|C(u_1, \mathcal{G}))]. \tag{2}$$

We call such a constraint-guided step in each flow sampling step an **ECI Step** (for *extrapolation-correction-interpolation*). With the deterministic flow ODE formulation, good properties can be naturally deduced with the flow-matching framework. The *extrapolation probability* $p_\theta(u_1|u_t)$ can learned by the pre-trained unconditional model by doing a deterministic one-step extrapolation of the current predicted vector field as $u_1 = u_t + (1-t)v_\theta(u_t)$. The *interpolation*

---

**Algorithm 2** ECI Step
---
1: **Input:** Learned vector field $v_\theta$, constraint $\mathcal{G}$, current noise data $u_t$, timestep $t' \geq t$.
2: $u_1 \leftarrow u_t + (1-t)v_\theta$      ▷ Extrapolation
3: $\hat{u}_1 \leftarrow C(u_1, \mathcal{G})$          ▷ Correction
4: $u_{t'} \leftarrow (1-t')u_0 + t'\hat{u}_1$    ▷ Interpolation
5: **return** $u_{t'}$

---

*probability* $q(\hat{u}_t|\hat{u}_1)$ is also well-defined in the FFM along the *OT-path* of measures (Kerrigan et al., 2023), where the ground truth data are linearly interpolated with random noises as $\hat{u}_t = (1-t)u_0 + t\hat{u}_1, u_0 \sim \mu_0(u)$. In this way, the expectation can be discarded, and the conditional probability can be further simplified as

$$p(\hat{u}_t|u_t, \mathcal{G}) = q(\hat{u}_t|C(u_t + (1-t)v_\theta(u_t), \mathcal{G})). \tag{3}$$

We summarize the ECI step in Algorithm 2, in which we also allow for $t' \geq t$ to advance the solver. Furthermore, we note that our ECI steps can be applied recursively to the same timestep in a manner similar to previous work that applied multiple rounds of guidance (Lugmayr et al., 2022; Ben-Hamu et al., 2024). With the constraint information "backpropagating" into the noise data via one ECI step, a recursive application of such steps can promote further information mixing between the constrained and unconstrained re-

---

**Algorithm 3** ECI Sampling (Euler Method)
---
1: **Input:** Learned vector field $v_\theta$, Euler steps $N$, mixing iterations $M$, constraint $\mathcal{G}$.
2: Sample noise function $u_0 \sim \mu_0(u)$.
3: **for** $t \leftarrow 0, 1/N, 2/N, \ldots, (N-1)/N$ **do**
4:      $u_t^{(0)} \leftarrow u_t$
5:      **for** $m \leftarrow 0, 1, \ldots, M-1$ **do**
6:          **if** $m < M-1$ **then**
7:              $u_t^{(m+1)} \leftarrow \text{ECIStep}(v_\theta, \mathcal{G}, u_t^{(m)}, t)$
8:          **else**        ▷ Advance the flow ODE solver
9:              $u_{t+1/N} \leftarrow \text{ECIStep}(v_\theta, \mathcal{G}, u_t^{(m)}, t+1/N)$
10: **return** $u_1$

---

gions, leading to a more consistent generation. At the last mixing step, we instead interpolate the new time step $t' > t$ with sample $u_{t'}$ to advance the ODE solver. The number of total mixing steps $M$ is a controllable hyperparameter. Algorithm 3 describes the complete sampling process for flow matching models using iterative ECI steps, for which we term **ECI sampling**. ECI guarantees exact satisfaction of the constraint and facilitates information mixing between the constrained and unconstrained regions to produce a more consistent generation. See Appendix A for details on the sampling and proof of exact satisfaction.

### 3.3 CONTROL OVER STOCHASTICITY

ECI sampling provides a unified zero-shot and gradient-free guidance framework for various constraints. Nonetheless, the varieties in the constraint types may impose challenges. For example, if

the constraint contains little information, the variance is expected to match the unconstrained case. On the other hand, with enough constraints to result in a well-posed PDE with a unique solution, the generation variance should be as small as possible (see Section 3.4).

Inspired by the work of the stochastic interpolant that unified flow matching and diffusion (Albergo et al., 2023), we propose to control the generation stochasticity with different sampling strategies of $u_0$ during our ECI sampling. For flow-based models, stochasticity only appears in the sampling process of $u_0 \sim \mu_0(u)$ for interpolating the new $\hat{u}_t$. In the previous context, we assume the initially sampled $u_0$ at $t = 0$ is used throughout the sampling process. This makes the trajectory straighter as the interpolation always starts with the same point. An alternative approach is to sample new noises on each step, effectively marginalizing over the prior noise measure $\mu_0$ and resulting in a more concentrated generation. We use a hyperparameter $R$ to denote the length of the interval for re-sampling the noise function $u_0$ for interpolation. Intuitively, a smaller $R$ leads to a more concentrated marginalized generation, whereas a larger $R$ leads to larger generational diversity. We quantitatively demonstrate the effect of this hyperparameter in Section 4.3 and also propose heuristic rules for choosing it.

## 3.4 GENERATIVE FFM AS A REGRESSION MODEL

We illustrate the potential of our ECI sampling framework for various regression tasks. Specifically, the solution set $\mathcal{U}_{\mathcal{F}|\mathcal{G}}$ defined in Section 3.1 will degenerate to a unique solution given enough constraints $\mathcal{G}$, leading to a well-posed PDE system. In this way, the generative task degenerates into a deterministic regression learning task, with the target distribution now shifted to the Dirac measure over the solution $\delta_{u_{\text{sol}}}$. Although our constrained generative framework is not designed for these regression tasks and has intrinsic stochasticity, in our empirical study, we show that our framework is able to generate competitive predictions to standard regression models, e.g., the Fourier Neural Operator (FNO) (Li et al., 2020a).

## 4 EMPIRICAL EVALUATION

In this section, we present an extensive evaluation of our ECI sampling and other existing zero-shot guidance models for both generative and regression tasks. Covering a wide range of diverse PDE systems, Table 2 summarizes the dataset specifications used in our evaluation. See Appendix B for more details regarding the PDE systems and the data generation process. For most zero-shot guidance methods, we first pre-train an unconditional flow matching model as the common prior generative model (See Appendix C.3). We test our proposed method using more than one mixing iteration in most cases. See Appendix D for the detailed experimental setup and baseline adaptations for flow sampling, and Appendix E for additional visualizations and results.

Table 2: Dataset specifications (IC / BC / CL for initial condition / boundary condition / conservation law). For different resolutions, we test models with the zero-shot superresolution setting.

| Type | Dataset | Underspecification | Spatial Resolution | Temporal Resolution | Constraint |
|---|---|---|---|---|---|
| Generative | Stokes Problem | IC & BC | 100 | 100 | IC / BC |
| | Heat Equation | IC & diffusion | 100 | 100 | IC |
| | Darcy Flow | BC & field | $101 \times 101$ | N/A | BC |
| | NS Equation | IC & forcing | $64 \times 64$ | 50 | IC |
| Regression | Heat Equation | diffusion | 100 / 200 | 100 / 200 | CL |
| | PME | diffusion | 100 / 200 | 100 / 200 | CL |
| | Stefan Problem | shock range | 100 / 200 | 100 / 200 | CL |
| | Stokes Problem | IC & BC | 100 | 100 | IC & BC |
| | NS Forward | IC & forcing | $64 \times 64$ | 50 | first 15 frames |

## 4.1 GENERATIVE TASK

In generative tasks, the solution set is non-degenerate with an infinite cardinality after imposing the constraint. We expect the generation distribution to match the ground truth shifted distribution. The prior FFM is pre-trained on a larger solution set, whereas during the zero-shot constrained

generation, we impose a specific PDE constraint to narrow down the solution set. Additionally, we use a conditionally-trained FFM (CondFFM) that takes known constraints as conditions during both training and sampling as an upper bound for the zero-shot guidance models. We conduct extensive experiments on various 2D and 3D PDE datasets with different types of constraints.

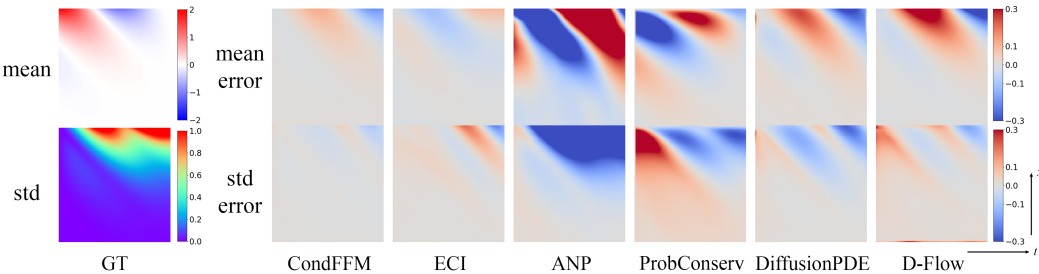

Figure 2: Generation mean and standard deviation errors for the Stokes problem with IC fixed. Note that grey indicates $\approx 0$ error, red for positive error, and blue for negative error. Noticeable IC errors (left column) can be observed for the gradient-based methods.

To compare two distributions over continuous functions, we follow Kerrigan et al. (2023) to calculate the pointwise mean and standard deviation (std) of the ground truth solutions and the solutions generated from various sampling methods. Mean squared errors (MSEs) of the mean (**MMSE**) and of the std (**SMSE**) are then calculated to assess the resemblance in these statistics. To evaluate the satisfaction of the constraint, the constraint error (**CE**) is defined as the MSE for the constraint operator $\text{CE}(u) = \text{MSE}(\mathcal{G}(u), 0)$ on the constrained region $\mathcal{X}_{\mathcal{G}}$. Inspired by the generative metric of Fréchet inception distance (FID) in image and PDE generation domains (Lim et al., 2023), we calculate the Fréchet distance between the hidden representations extracted by the pre-trained PDE foundation model Poseidon (Herde et al., 2024). We call this metric Fréchet Poseidon distance (**FPD**). A lower FPD indicates a closer match to the ground truth distribution regarding the high-level representations. To alleviate the impact of randomness in sampling, we sample 512 solutions for the 2D datasets and 100 solutions for the 3D datasets for all methods to calculate the metrics.

We carry out experiments on 1D **Stokes problem**, 1D **heat equation**, 2D **Darcy flow**, and 2D **Navier-Stokes (NS) equation**. The detailed description of these PDE systems, as well as the set of PDE parameters for pre-training and constrained generation, are provided in Appendix B. Note that we also ensure at least two degrees of freedom (underspecification) in generative tasks, as demonstrated in Table 2. This ensures that distribution over the constrained generations does not collapse to a Dirac distribution as in a regression task (see Figure 1).

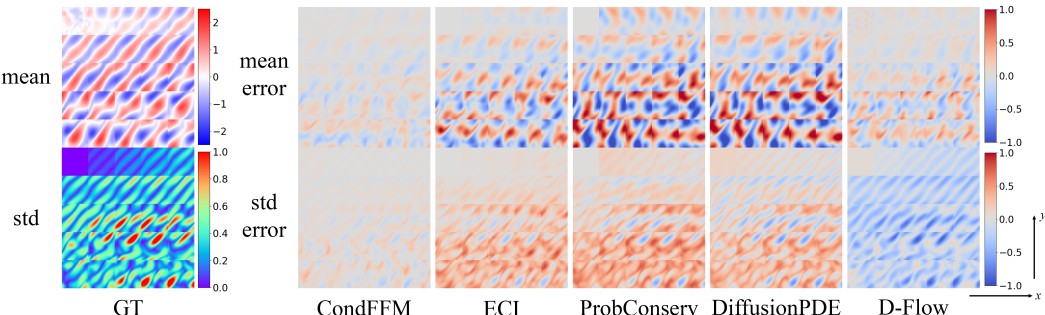

Figure 3: Generation mean and standard deviation errors of the trajectories for the NS equation with IC fixed. 25 downsampled time frames are plotted for each method.

Table 3 summarizes the results for the generative tasks with the best zero-shot method performance highlighted in bold. Figure 2 shows the visualization of pointwise errors for the generation statistics for the Stokes problem with IC fixed together with the ground truth reference. Figure 3 shows the visualization for the NS equation with 25 downsampled time frames. Our proposed ECI sampling achieves state-of-the-art performance on most metrics and also enjoys zero constraint errors. Notice-

Table 3: Generative metrics on various constrained PDEs. The best results for zero-shot methods (CondFFM is not zero-shot) are highlighted in bold. The unconstrained pre-trained FFM is provided.

| Dataset | Metric | ECI | ANP | ProbConserv | DiffusionPDE | D-Flow | CondFFM | FFM |
|---|---|---|---|---|---|---|---|---|
| Stokes IC | MMSE / $10^{-2}$ | **0.090** | 13.750 | 2.014 | 0.601 | 1.015 | 0.121 | 2.068 |
| | SMSE / $10^{-2}$ | **0.127** | 9.183 | 1.528 | 0.330 | 0.219 | 0.016 | 1.572 |
| | CE / $10^{-2}$ | **0** | 3.711 | **0** | 0.721 | 1.021 | 0.017 | 9.792 |
| | FPD | **0.076** | 26.685 | 13.075 | 0.202 | 0.614 | 0.028 | 13.342 |
| Stokes BC | MMSE / $10^{-2}$ | **0.005** | 3.003 | 2.938 | 0.603 | 4.800 | 0.032 | 3.692 |
| | SMSE / $10^{-2}$ | **0.003** | 3.055 | 2.302 | 0.341 | >100 | 0.047 | 3.320 |
| | CE / $10^{-2}$ | **0** | 29.873 | **0** | 0.007 | 53.551 | 0.012 | >100 |
| | FPD | **0.010** | 5.339 | 2.780 | 1.162 | 27.680 | 0.048 | 2.824 |
| Heat Equation | MMSE / $10^{-2}$ | 1.605 | 5.127 | 3.744 | **0.979** | 1.174 | 0.008 | 3.954 |
| | SMSE / $10^{-2}$ | **0.157** | 0.740 | 4.453 | 1.291 | 9.928 | 0.004 | 4.737 |
| | CE / $10^{-2}$ | **0** | 0.183 | **0** | 11.646 | 94.551 | 0.002 | 49.296 |
| | FPD | 1.365 | 2.432 | 2.639 | **0.951** | 3.831 | 0.006 | 2.799 |
| Darcy Flow | MMSE / $10^{-2}$ | 2.314 | 14.647 | 56.608 | 10.442 | **1.728** | 0.043 | 58.879 |
| | SMSE / $10^{-2}$ | **0.592** | 1.269 | 65.967 | 1.097 | 6.343 | 0.037 | 69.432 |
| | CE / $10^{-2}$ | **0** | 3.152 | **0** | 0.656 | 14.553 | 0.071 | >$10^4$ |
| | FPD | **0.946** | 5.805 | >100 | 8.027 | 10.043 | 0.021 | >100 |
| NS Equation | MMSE / $10^{-2}$ | 7.961 | 59.020 | 18.635 | 19.141 | **2.846** | 0.851 | 18.749 |
| | SMSE / $10^{-2}$ | **5.846** | 14.990 | 8.306 | 5.941 | 6.840 | 0.554 | 8.423 |
| | CE / $10^{-2}$ | **0** | 5.527 | **0** | 2.295 | 0.776 | 0.079 | 11.451 |
| | FPD | **1.131** | 57.263 | 2.171 | 5.750 | 1.651 | 0.232 | 2.185 |

ably, ECI sampling excels in capturing the higher-order distributional properties of the shifted conditional measure, e.g., the standard deviation and FPD, and even surpasses the conditional FFM in several simple 2D cases. Specifically, we observe noticeable artifacts around the IC for the gradient-based methods, i.e., DiffusionPDE and D-Flow, as they do not guarantee the exact satisfaction of the constraint. For ProbConserv (Hansen et al., 2023), although the constraint is satisfied exactly, the naive non-iterative correction approach leaves a sudden change around the boundary that violates the physical feasibility. In addition, Table 4 shows that our gradient-free approach enjoys sampling efficiency in both 2D and 3D cases compared with gradient-based methods. Noticeably, ECI-1 achieved $\times 440$ acceleration and $\times 310$ memory saving compared to gradient-based D-Flow on the Stokes problem. More visualizations for the other PDE systems and the different generation statistics are provided in Appendix E.1.

Table 4: Generation setup (batch size vs Euler step), time per sample in seconds, and GPU memory for 2D (Stokes IC) and 3D (NS equation) PDE systems. A fraction sample size is available for ANP.

| Dataset | Resource | ECI-1 | ECI-5 | CondFFM | ANP | ProbConserv | DiffusionPDE | D-Flow |
|---|---|---|---|---|---|---|---|---|
| Stokes IC | #sample/#Euler | 128/200 | 128/200 | 128/200 | 32/NA | 128/200 | 128/200 | 2/200 |
| | Time/sample/s | 0.065 | 0.325 | 0.057 | 0.009 | 0.058 | 0.131 | 28.774 |
| | GPU Memory/GB | 5.4 | 5.4 | 5.4 | 7.4 | 5.4 | 10.8 | 26.4 |
| NS equation | #sample/#Euler | 25/100 | 25/100 | 25/100 | 0.2/NA | 25/100 | 25/100 | 1/20 |
| | Time/sample/s | 0.415 | 2.067 | 0.669 | 2.324 | 0.675 | 0.676 | 8.456 |
| | GPU Memory/GB | 16.3 | 16.3 | 16.3 | 11.0 | 16.3 | 27.0 | 27.1 |

## 4.2 REGRESSION TASK

We additionally experiment with regression scenarios (see Section 3.4), where enough constraints reduce the generative task into the standard regression setting in neural operator learning. As the ground truth targets are available in this case, we calculate the MSE and other related evaluation metrics and compare them with traditional NOs.

**Uncertainty Quantification.** In the uncertainty quantification task, random context points from the true solution are sampled together with the conservation laws to pin down the unique solution. Following Hansen et al. (2023), we experiment with our ECI sampling on the Generalized Porous Medium Equation (GPME) family of equations using identical PDE parameter choices. We evaluate

Table 5: Uncertainty quantification metrics on constrained PDEs from the Generalized Porous Medium Equation (GPME) family of equations. All baseline results are taken from Hansen et al. (2023). The best results are highlighted in bold.

| Dataset | Metric | ECI | ANP | SoftC-ANP | HardC-ANP | ProbConserv |
|---|---|---|---|---|---|---|
| Heat Equation | CE / $10^{-3}$ | **0** | 4.68 | 3.47 | **0** | **0** |
| | LL | 1.90 | 2.72 | 2.40 | **3.08** | 2.74 |
| | MSE / $10^{-4}$ | **0.81** | 1.71 | 2.24 | 1.37 | 1.55 |
| PME | CE / $10^{-3}$ | **0** | 6.67 | 5.62 | **0** | **0** |
| | LL | 2.19 | 3.49 | 3.11 | 3.16 | **3.56** |
| | MSE / $10^{-4}$ | 0.19 | 0.94 | 1.11 | 0.43 | **0.17** |
| Stefan Problem | CE / $10^{-2}$ | **0** | 1.30 | 1.72 | **0** | **0** |
| | LL | 3.30 | 3.53 | **3.57** | 2.33 | 3.56 |
| | MSE / $10^{-3}$ | **1.89** | 5.38 | 6.81 | 5.18 | **1.89** |

Table 6: Neural operator learning metrics on two constrained PDEs. We abbreviate OOM for out-of-memory. The results for the best generative models are highlighted in bold.

| Dataset | Metric | ECI | ANP | ProbConserv | DiffusionPDE | D-Flow | FNO |
|---|---|---|---|---|---|---|---|
| Stokes Problem | MMSE / $10^{-3}$ | **0.050** | 16.470 | 103.136 | 5.839 | 89.514 | 0.033 |
| | SMSE / $10^{-2}$ | **0.028** | 0.033 | 7.509 | 0.473 | 136.936 | 0 |
| NS Equation | MMSE / $10^{-2}$ | **0.069** | OOM | 46.673 | 19.015 | 1.310 | 0.380 |
| | SMSE / $10^{-2}$ | **0.002** | OOM | 28.789 | 13.914 | 4.068 | 0 |

the generation results on three instances of the GPME, i.e., the heat equation, porous medium equation (PME), and Stefan problem, where the additional constraints are various physical conservation laws. We also follow the paper to report the pointwise Gaussian log-likelihood (LL) based on the sample mean and variance. The results in Table 5 show that our ECI sampling is able to achieve comparable results with ProbConserv (Hansen et al., 2023), which is specifically tailored for these uncertainty quantification tasks. We also notice that, though being a generative model with intrinsic stochasticity from the prior noise distribution, our method is able to produce quite confident predictions with little variance. It surpasses the other baselines in terms of MSE but at the cost of worse LL. See Appendix E.3 for visualizations and more discussions.

**Neural Operator Learning.** We further experiment with the zero-shot neural operator learning task on the Stokes problem and Navier-Stokes equation. For the Stokes problem, both BC and IC are prescribed to give a unique solution. For the NS equation, we follow Li et al. (2020a) to consider the regression task from the first 15 frames to the other 35 frames. In addition, a standard FNO (Li et al., 2020a) was trained as the regression baseline. Table 6 summarizes the results. The large amount of context pixels in the NS equation makes it infeasible for the ANP model. Our ECI sampling significantly outperforms gradient-based methods and reaches comparable or even better MSEs than the NOs directly trained on these regression tasks. Similar to the uncertainty quantification task, we observe that our ECI sampling is quite confident about its prediction with little variance. We also provide a comparison of ECI sampling with different numbers of frames fixed as the constraint in Appendix E.4 to demonstrate how the variance gradually reduces with more constraint information prescribed.

## 4.3 ABLATION STUDY

Similar to most existing controlled generation methods for diffusion or flow models, the number of mixing iterations is a tunable hyperparameter for our ECI sampling to control the extent of information mixing. Intuitively, an insufficient number of mixing iterations may fail to mix the information between the constrained and the unconstrained regions, leading to more artifacts. On the other hand, an excessively large number of mixing iterations may discard the prior PDE structure learned in the pre-trained generative model, leading to inconsistency with the PDE system. We further test with

the length of noise re-sampling interval $R$, as discussed in Section 3.3, to control stochasticity for different constrained generation tasks.

We test the combination of mixing iterations $M \in \{1, 2, 5, 10, 20\}$, with a re-sampling interval $R \in \{1, 2, 5, 10\}$, and no re-sampling settings on the Stokes problem with the IC or BC prescribed. Figure 4 illustrates the results of different ECI sampling settings together with other baselines. For the IC task, $R = $ None performs the best, whereas for the BC task, $R = $ None performs worse, with $R = 5$ and $R = 2$ achieving lower MSEs. In both cases, the MSEs do not decrease monotonically with increasing mixing iterations. Therefore, the mixing iteration impacts the final performance in a task-specific manner. We hypothesize that the difference in the shifted variance may cause this difference in the performance after imposing a specific constraint. For the Stokes problem, the prior ground truth variance is higher for the BC than for the IC. Prescribing BC values provides more information and reduces the uncertainty, making it easier for the model to be guided. In contrast, prescribing IC values provides less information, which results in a larger variance in the shifted distribution and makes it harder to guide the model.

Inspired by the aforementioned observations, we propose the following heuristic rules for choosing the appropriate number of mixing iterations $M$ and re-sample interval $R$: 1) We usually limit $M$ between 1-10, as a larger number of iterations is both computation-intensive and leads to worse performance; and 2) For easier tasks with lower variance, we choose a smaller $M$ and smaller $R$; and for harder tasks with higher variance, we choose a larger $M$ and larger $R$ or no re-sampling. Indeed, we use 1 mixing iteration for the relatively easy tasks on the heat equation and Darcy flow and use 10 mixing iterations for the harder NS equation.

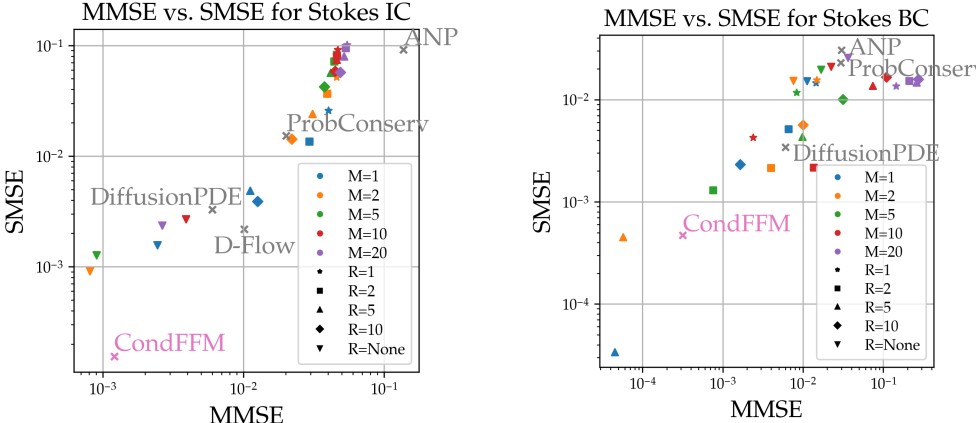

Figure 4: MMSE and SMSE of ECI sampling with mixing iterations $M$ in different colors and re-sampling intervals $R$ in different markers in the Stokes problem with IC (left) or BC (right) fixed. "None" for no re-sampling (the initial noise is used).

## 5 CONCLUSION

In this work, we present ECI sampling as a unified zero-shot and gradient-free framework for guiding pre-trained flow-matching models towards hard constraints. Instantiated on PDE systems, our framework obtains superior sampling quality and efficient sampling time compared to existing zero-shot methods, and it also enjoys the additional benefit of a zero-shot regression model with comparable performance to the traditional regression models. We also highlight the potential of ECI sampling on other domains outside of SciML, including supply chain and times series, which we leave as future work. We further note a limitation of our framework: the number of mixing iterations and the re-sampling interval have a task-specific impact on the final performance. While we have provided empirical evaluations of these hyperparameters and suggested heuristic guidelines for selecting them, theoretical analyses would be beneficial.

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

# Supplementary Material

## A    DETAILS OF CONSTRAINED SAMPLING

In this section, we further discuss additional details of our proposed ECI sampling framework. Our proposed ECI sampling interleaves extrapolation, correction, and interpolation stages at each sampling step for flow-based generative models to enforce the hard constraint. We will elaborate further on the implementation details below.

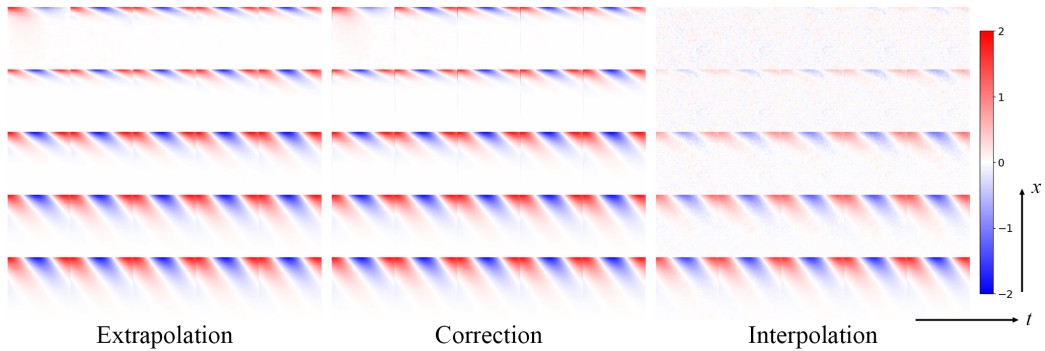

Figure 5: Intermediate results (trajectory) of the extrapolation, correction, and interpolation stages for the Stokes problem with IC fixed. The trajectories demonstrate how the initial random noise gradually transforms into the controlled generation with decreasing artifacts around the boundary.

### A.1    FORMAL DEFINITION OF CONSTRAINED GENERATION

First, we provide a more rigorous measure-theoretic definition of constrained generation for functional data as a generative task. Consider a well-posed PDE system $\mathcal{F}_\phi u(x) = 0, x \in \mathcal{X}$ with PDE parameters $\phi \in \Phi$, then the solution map $\mathcal{T} : \phi \mapsto u(x)$ is a well-defined mapping. For a fixed distribution $p_\Phi$ over the parameter $\phi$, we can naturally define the measure over the solution set as the push-forward of the solution map: $\mu_\mathcal{F} = \mathcal{T}_* p_\Phi$. Now consider some constraint operator $\mathcal{G}u(x) = 0, x \in \mathcal{X}_\mathcal{G} \subseteq \mathcal{X}$ defined on a subset of the PDE domain. Denote the solution set for $\mathcal{G}$ as $\mathcal{U}_\mathcal{G} = \{u(x) : \mathcal{G}u(x) = 0, x \in \mathcal{X}_\mathcal{G}\}$ and the characteristic function over the solution set as $\chi_\mathcal{G}$ such that

$$\chi_\mathcal{G}(u) = \begin{cases} 1, & u \in \mathcal{U}_\mathcal{G} \\ 0, & u \notin \mathcal{U}_\mathcal{G} \end{cases}. \tag{4}$$

With the non-degenerating assumption that $\int_\mathcal{U} \chi_\mathcal{G} d\mu_\mathcal{F} > 0$ (this is essentially saying that we should have at least one solution), the conditional measure can be obtained as

$$\mu_{\mathcal{F}|\mathcal{G}}(A) := \mu_\mathcal{F}(A|\mathcal{G}) = \int_A \chi_\mathcal{G} d\mu_\mathcal{F} \Big/ \int_\mathcal{U} \chi_\mathcal{G} d\mu_\mathcal{F} , \quad \forall A \subseteq \mathcal{U}, \tag{5}$$

where $\mathcal{U}$ is the Hilbert space of all "well-behaving" functions (see Kerrigan et al. (2023) for regularity conditions). We assume $\mu_\mathcal{F}$ can be approximated well by generative priors like FFM, and we want to guide it towards the conditional measure $\mu_{\mathcal{F}|\mathcal{G}}$. In practice, the ground truth solution map $\mathcal{T}$ can be obtained by either exact analytical solutions or by using numerical PDE solvers. With a pre-defined distribution over PDE parameters (often uniform over some interval, see Table 7 and Appendix B for detailed specification), we can easily sample the PDE parameters and apply the solution map to obtain the sampled functions for pre-training.

For constrained sampling, instead of directly calculating the conditional measure $\mu_{\mathcal{F}|\mathcal{G}}$, we can sample the PDE parameters from the pull-back probability $p_{\Phi|\mathcal{G}} = \mathcal{T}^* \mu_{\mathcal{F}|\mathcal{G}}$. In practice, we directly assume such a conditional probability distribution $p_{\Phi|\mathcal{G}}$ over the PDE parameters is known. Indeed, for all the experiments in this paper, we simply choose a subset of PDE parameters and apply the solution map to obtain functions as the ground truth constrained solutions.

## A.2 EXTRAPOLATION OF SOLUTIONS

The continuous-time and deterministic formulation of the flow-matching sampling makes it possible to apply different step sizes to advance the solver for the flow ODE. Specifically, at timestep $t$, we can make a one-step prediction with a step size of $1 - t$ that directly advances the ODE solver to $t = 1$ as $u_1 = u_t + (1 - t)v$. Theoretically, if the optimal transport paths are learned exactly, the flow should be straight (Lipman et al., 2022), and any arbitrary discretization should lead to the same target. Indeed, in the left plot in Figure 5, the extrapolation gives reasonable prediction even at a timestep close to 0. The constraint, on the other hand, is not necessarily satisfied in this stage.

## A.3 SOLUTION CORRECTION

Our ECI sampling scheme is a unified framework for different constraint operators as long as the corresponding correction algorithm at $t = 1$ is readily available. Specifically, we consider constraint operators $\mathcal{G}$ of two following categories: 1) *value constraints* that specify exact values for a subset of the PDE domain: $u(x) = g(x), x \in \mathcal{X}_{\mathcal{G}} \subseteq \mathcal{X}$, and 2) *region constraints* that specify an exact value for the integration over some subset of the domain: $\int_{\mathcal{X}_{\mathcal{G}} \subseteq \mathcal{X}} u(x)dx = a \in \mathbb{R}$. These two classes of constraints encompass a wide range of practical constraints for PDE systems.

*Example* 1. For $\mathcal{X} = \Omega \times [0, T]$ where $\Omega \in \mathbb{R}^{D-1}$, the follow examples are value constraints:

- *Initial condition* (IC): $u(x, 0) = g(x), x \in \Omega$.

- *Boundary condition* (BC): $u(x, t) = f(t), x \in \partial\Omega, t \in [0, T]$.

*Example* 2. For $\mathcal{X} = [0, 1] \times [0, T]$, the following examples are region constraints:

- *Mass conservation*: $\int_0^1 u(x, t)dx = 0, t \in [0, T]$.

- *Periodic boundary condition* (PBC): $u(0, t) = u(1, t), t \in [0, T]$.

Furthermore, any linear conservation law can be understood as a region constraint, specifying a conserved quantity over a region in the PDE domain. As these constraints are linear constraints, we follow Hansen et al. (2023) to apply an oblique projection as a correction. The corresponding correction algorithms are outlined in Algorithm 4 and 5.

---

**Algorithm 4** Value Constraint Correction

1: **Input:** function $u_1$, constraint function $g$, region $\mathcal{X}_{\mathcal{G}}$.
2: $\hat{u}_1 \leftarrow \mathbb{1}[x \in \mathcal{X}_{\mathcal{G}}] \odot g + \mathbb{1}[x \notin \mathcal{X}_{\mathcal{G}}] \odot u_1$
3: **return** $\hat{u}_1$

---

**Algorithm 5** Region Constraint Correction

1: **Input:** function $u_1$, conservation value $a$, region $\mathcal{X}_{\mathcal{G}}$.
2: $g \leftarrow u_1 + (a - \int_{\mathcal{X}_{\mathcal{G}}} u_1(x)dx) / \int_{\mathcal{X}_{\mathcal{G}}} dx$
3: $\hat{u}_1 \leftarrow \mathbb{1}[x \in \mathcal{X}_{\mathcal{G}}] \odot g + \mathbb{1}[x \notin \mathcal{X}_{\mathcal{G}}] \odot u_1$
4: **return** $\hat{u}_1$

---

It is easy to verify that the corrected function $\hat{u}_1$ in these correction algorithms indeed satisfies the corresponding constraint exactly. We can also have multiple (even infinitely many) non-overlapping constraints. For example, consider the generalized mass conservation laws $\int_0^1 u(x, t)dx = f(t) \equiv 0, t \in [0, T]$ for some given function $f(t)$, we have a different conservation law for each PDE timestep $t$. After discretization, we have a finite set of non-overlapping constraints that can be corrected according to Algorithm 5. The correction stage is demonstrated in the middle plot in Figure 5. Note how directly applying the correction algorithm creates noticeable artifacts around the boundary and how such artifacts gradually disappear when advancing the ODE solver.

### A.4 INTERPOLATION OF SOLUTIONS

The iterative nature of flow sampling requires interpolation back to arbitrary timestep $t$ during each sampling step. This can be understood as the forward noising process along the conditional path of measures and can be efficiently calculated as the linear interpolation $u_t = (1-t)u_1 + tu_0$ according to the FFM formulation (Kerrigan et al., 2023). In the right plot in Figure 5, we demonstrate how the initial Gaussian process noise is transformed into the constrained generation. We have also discussed the impact of the re-sampling interval length to control stochasticity in the generation in Section 3.3.

### A.5 PROOF OF EXACT SATISFACTION OF CONSTRAINTS

**Proposition 1.** *Suppose the corrected algorithm $C(u_1, \mathcal{G})$ in Equation 2 satisfies the constraint $\mathcal{G}$ exactly, then for any number of mixing steps $M \geq 1$, the ECI sampling scheme described in Algorithm 3 exactly recovers the constraint in the final generation at $t = 1$.*

*Proof.* As the advancing step for the ODE solver (the last mixing step) will always perform, it suffices to consider $M = 1$. Consider the last Euler step at $t = 1 - 1/N$, the linear interpolation procedure $q(\hat{u}_{t'}|\hat{u}_1) = 0 \cdot u_0 + 1 \cdot \hat{u}_1 = \hat{u}_1$ will be deterministic as the noise will not contribute to the final interpolant at timestep $t' = 1$. Therefore, the interpolation will deterministically produce $q(\hat{u}_{t'}|\hat{u}_1) = \hat{u}_1 = C(u_t + (1-t)v_\theta(u_t), \mathcal{G})$ which satisfies the constraint exactly. □

## B DATASET DESCRIPTION AND GENERATION

This section provides a more detailed description of the datasets used in this work and their generation procedure. In addition to the statistics in Table 2, we further provide additional information in Table 7 for pre-training the FFM as our generative prior. For dataset types, *synthetic* indicates that the exact solutions are calculated on the fly based on the randomly sampled PDE parameters for both the training and test datasets. We manually assign the training set with 5k solutions and the test set with 1k solutions. On the other hand, *simulated* indicates that the solutions are pre-generated using numerical PDE solvers and are different for the training and test datasets.

Table 7: More dataset specifications for pre-training the prior FFM.

| Dataset | PDE Parameter | Split | Spatial Domain | Time Domain | Type |
|---|---|---|---|---|---|
| Stokes Problem | $k \sim U[2, 20]$ $\omega \sim U[2, 8]$ | 5k / 1k | $[0, 1]$ | $[0, 1]$ | Synthetic |
| Heat Equation | $\alpha \sim U[1, 5]$ $\phi \sim U[0, \pi]$ | 5k / 1k | $[0, 2\pi]$ | $[0, 1]$ | Synthetic |
| Darcy Flow | $k$ (see below) $C \sim U[-2, 2]$ | 10k / 1k | $[0, 1]^2$ | NA | Simulated |
| NS Equation | $w_0, f$ (see below) | 10k / 1k | $[0, 1]^2$ | $[0, 49]$ | Simulated |
| PME | $m \sim U[1, 6]$ | 5k / 1k | $[0, 1]$ | $[0, 1]$ | Synthetic |
| Stefan Problem | $u^* \sim U[0.55, 0.7]$ | 5k / 1k | $[0, 1]$ | $[0, 0.1]$ | Synthetic |

### B.1 STOKES PROBLEM

The 1D Stokes problem is given by the heat equation:

$$
\begin{aligned}
u_t &= \nu u_{xx}, & x &\in [0, 1], t \in [0, 1], \\
u(x, 0) &= Ae^{-kx}\cos(kx), & x &\in [0, 1], \\
u(0, t) &= A\cos(\omega t), & t &\in [0, 1],
\end{aligned}
\tag{6}
$$

with viscosity $\nu \geq 0$, oscillation frequency $\omega$, amplitude $A > 0$, and $k = \sqrt{\omega/(2\nu)}$. The analytical solution is given by $u_{\text{exact}}(x, t) = Ae^{-kx}\cos(kx - \omega t)$. Note that $k$ and $\omega$ independently and

uniquely define the IC and BC, respectively, and fixing both values reduces the PDE to a well-posed system with a unique solution. We follow Saad et al. (2023) to construct the dataset by fixing $A = 2$ and sampling $\omega \sim U[2,8], k \sim U[2,20]$ for pre-training. During constrained sampling, we test two different settings of prescribing IC with $k = 5$ or BC with $\omega = 6$, respectively. In the regression task, we provide both the IC and BC values. Figure 1 shows the ground truth solutions for the unconstrained and constrained Stokes problem.

## B.2 HEAT EQUATION

The 1D heat (diffusion) equation with periodic boundary conditions is given as

$$
\begin{aligned}
u_t &= \alpha u_{xx}, & x &\in [0, 2\pi], t \in [0, 1], \\
u(x, 0) &= \sin(x + \varphi), & x &\in [0, 2\pi], \\
u(0, t) &= u(2\pi, t), & t &\in [0, 1],
\end{aligned}
\tag{7}
$$

where $\alpha$ denotes the diffusion coefficient and $\varphi$ denotes the phase of the sinusoidal IC. The exact solution is given as $u_{\text{exact}}(x, t) = e^{-\alpha t} \sin(x + \varphi)$. We sample $\alpha \sim U[1, 5]$ and $\varphi \sim U[0, \pi]$ for pre-training. During constrained sampling, we fix the phase $\phi = \pi/4$. Figure 6 shows the ground truth solution family.

In the uncertainty quantification task, we follow Hansen et al. (2023) to fix $\phi = 0$ and vary only the diffusion coefficient in $U[1, 5]$ during pre-training. During constrained generation, we use the same setting to fix $\alpha = 1, t = 0.5$ for calculating the MSE and LL. The global conservation law for the heat equation in Equation 7 is written as

$$
\int_0^{2\pi} u(x, t) dx = 0, \quad t \in [0, 1].
\tag{8}
$$

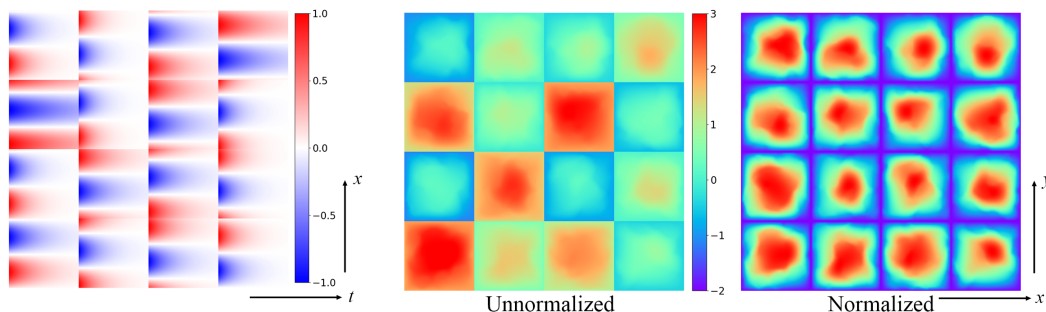

Figure 6: Ground truth solutions for the heat equation with $\alpha \sim U[1, 5], \varphi \sim U[0, \pi]$.

Figure 7: Ground truth solutions for the Darcy flow with $C \sim U[-2, 2]$. Solution values are unnormalized on the left and normalized per-solution on the right.

## B.3 DARCY FLOW

The 2D Darcy flow is a time-independent, second-order, elliptic PDE with the following form:

$$
\begin{aligned}
-\nabla \cdot (k(x)\nabla u(x)) &= f(x), & x &\in D = [0, 1]^2, \\
u(x) &= C, & x &\in \partial D,
\end{aligned}
\tag{9}
$$

where $k$ denotes the permeability field and $f$ denotes the forcing function. We follow Li et al. (2020a) to fix $f(x) \equiv 1$ and sample $k \sim \psi_* \mathcal{N}(0, (-\Delta + 9I)^{-2})$ with zero Neumann boundary conditions on the Laplacian. The mapping $\psi : \mathbb{R} \to \mathbb{R}$ takes the value 12 for positive numbers and 3 for the negative numbers, and the pushforward is defined pointwise. Unlike previous work, the boundary condition is sampled from $C \sim U[-2, 2]$. It can be verified that, for the same $k$, if $u_0$ is the solution for prescribing the boundary condition $u(x) = 0, x \in \partial D$, then $u_0 + C$ is the solution when prescribing the boundary condition $u(x) = C, x \in \partial D$. Therefore, we simulate 1000 samples with different $k$ and the zero boundary condition and sample $C$ on the fly for both the training and testing datasets. During constrained generation, we fix $C = 1$. Figure 7 shows the ground truth solution family.

### B.4 NAVIER-STOKES EQUATION

The 2D Navier-Stokes (NS) equation for a viscous, incompressible fluid in the vorticity form with periodic boundary conditions is given as

$$
\begin{aligned}
\partial_t w(x,t) + u(x,t) \cdot \nabla w(x,t) &= \nu \Delta w(x,t) + f(x), & x \in [0,1]^2, t \in [0,T], \\
\nabla \cdot u(x,t) &= 0, & x \in [0,1]^2, t \in [0,T], \\
w(x,0) &= w_0(x), & x \in [0,1]^2,
\end{aligned}
\tag{10}
$$

where $u$ denotes the velocity field, $w = \nabla \times u$ denotes the vorticity, and $w_0$ denotes the initial vorticity. We follow Li et al. (2020a) to sample $w_0 \sim \mathcal{N}(0, 7^{3/2}(-\Delta + 49I)^{-5/2})$ with periodic boundary conditions. The forcing term is defined as $f(x) = 0.1\sqrt{2}\sin(2\pi(x_1 + x_2) + \phi)$ where $\phi \sim U[0, \pi/2]$, and the viscosity is fixed to be $\nu = 10^{-3}$. We sample 100 initial vorticities and 100 forces with linearly spaced $\phi$ in the interval $[0, \pi/2]$. We mesh-grid the vorticities and forces to build 10000 solutions for the training set. For the testing set, we sample another 10 initial vorticities and mesh-grid them with the same 100 forces, resulting in another 1000 solutions. We use $T = 49$ and sample 50 snapshots of the numerical solutions. We use the same generative NS equation datasets for training for the regression task that maps the first 15 frames to the other 35 frames. The regression evaluation is performed on the first sample in the test set.

### B.5 POROUS MEDIUM EQUATION

The nonlinear Porous Medium Equation (PME) with zero initial and time-varying Dirichlet left boundary conditions is given as

$$
\begin{aligned}
u_t &= \nabla \cdot (u^m \nabla u), & x \in [0,1], t \in [0,1], \\
u(x,0) &= 0, & x \in [0,1], \\
u(0,t) &= (mt)^{1/m}, & t \in [0,1], \\
u(1,t) &= 0, & t \in [0,1],
\end{aligned}
\tag{11}
$$

where $m \geq 1$. The exact solution is given as $u_{\text{exact}}(x,t) = (m\,\text{ReLU}(t - x))^{1/m}$. We follow Hansen et al. (2023) to sample $m \sim U[1, 5]$ for pre-training and fix $m = 1, t = 0.5$ for sampling. The conservation law for the PME in Equation 11 is written as

$$
\int_0^1 u(x,t)dx = \frac{(mt)^{1+1/m}}{m+1}, \quad t \in [0,1].
\tag{12}
$$

Note that this conservation law implicitly contains information about $m$.

### B.6 STEFAN PROBLEM

The Stefan problem is a challenging and nonlinear case of the Generalized Porous Medium Equation (GPME). With fixed Dirichlet boundary conditions, it is given as:

$$
\begin{aligned}
u_t &= \nabla \cdot (k(u)\nabla u), & x \in [0,1], t \in [0,T], \\
u(x,0) &= 0, & x \in [0,1], \\
u(0,t) &= 1, & t \in [0,T], \\
u(1,t) &= 0, & t \in [0,T],
\end{aligned}
\tag{13}
$$

where $k(u)$ denotes the nonlinear step function with respect to a fixed shock value $u^*$:

$$
k(u) = \begin{cases} 1, & u \geq u^* \\ 0, & u < u^*. \end{cases}
\tag{14}
$$

The exact solution is given as

$$
u_{\text{exact}}(x,t) = \mathbb{1}[u \geq u^*]\left(1 - (1 - u^*)\frac{\text{erf}(x/(2\sqrt{t}))}{\text{erf}(\alpha)}\right),
\tag{15}
$$

where $\mathrm{erf}(z) = \frac{2}{\sqrt{\pi}} \int_0^z \exp(-t^2)dt$ is the error function and $\alpha$ is uniquely determined by $u^*$ as the solution for the nonlinear equation $(1 - u^*)/\sqrt{\pi} = u^* \mathrm{erf}(\alpha)\alpha \exp(\alpha^2)$. We follow Hansen et al. (2023) to sample the shock value $u^* \sim U[0.55, 0.7]$ and use $T = 0.1$. During sampling, we fix $u^* = 0.6, t = 0.05$. The conservation law for the Stefan problem in Equation 13 is

$$\int_0^1 u(x,t)dx = \frac{2(1 - u^*)}{\mathrm{erf}(\alpha)}\sqrt{\frac{t}{\pi}}, \quad t \in [0,1]. \tag{16}$$

## C   FUNCTIONAL FLOW MATCHING AS THE GENERATIVE PRIOR

This section provides additional mathematical backgrounds on flow matching, functional flow matching, and our pre-training settings for FFM as the generative prior.

### C.1   FLOW MATCHING

We first provide more details regarding the flow matching framework on common Euclidean data (e.g., pixel values of images) before proceeding to functional flow matching for functional data (e.g., PDE solutions). Flow matching (Lipman et al., 2022) is a generative framework built upon continuous normalizing flows (Chen et al., 2018). This flow-based model can be viewed as the continuous generalization of the score matching (diffusion) model that allows for a more flexible design of the denoising process following the optimal transport formulation (Lipman et al., 2022). Flow matching tries to learn the time-dependent *vector field* $v_t : \mathbb{R}^d \times [0,1] \to \mathbb{R}^d$ that defines a continuous time-dependent diffeomorphism called the *flow* $\psi_t : \mathbb{R}^d \times [0,1] \to \mathbb{R}^d$ via the following *flow ODE*:

$$\frac{\partial}{\partial t}\psi_t(x) = v_t(\psi_t(x)), \quad x_0 \sim p_0(x), \tag{17}$$

where $p_0$ is the initial noise distribution. The flow induces a probability path with the push-forward $p_t = (\psi_t)_* p_0$ for generative modeling. The vanilla flow matching loss can be written as

$$\mathcal{L}_{\mathrm{FM}} = \mathbb{E}_{t \sim U[0,1], x_1 \sim p_1(x)}[\|v_\theta(x_t, t) - u_t(x_t)\|^2], \tag{18}$$

where $x_t := \psi_t(x)$ is the noise data, $p_1$ is the target data distribution, and $u_t(x_t)$ is the ground truth data vector field at $x_t$ and timestep $t$. The vanilla flow matching objective is generally intractable, as we do not know the ground truth vector fields. Lipman et al. (2022) demonstrates a key observation that, when considering the *conditional* probability path $\psi_t(x|x_1)$ conditioned on the target data, the *conditional* vector field $u_t(x_t|x_1)$ can be calculated analytically while sharing the same gradient with the vanilla flow matching objective. The conditional flow matching objective can be written as

$$\mathcal{L}_{\mathrm{CFM}} = \mathbb{E}_{t \sim U[0,1], x_1 \sim p_1(x)}[\|v_\theta(x_t, t) - u_t(x_t|x_1)\|^2]. \tag{19}$$

Following Chen & Lipman (2023), when further conditioned on the noise $x_0 \sim p_0(x)$, the CFM loss can be reparameterized into a simple form:

$$\mathcal{L}_{\mathrm{CFM}} = \mathbb{E}_{t \sim U[0,1], x_0 \sim p_0(x), x_1 \sim p_1(x)}[\|v_\theta(x_t, t) - (x_1 - x_0)\|^2], \quad x_t = (1-t)x_0 + tx_1. \tag{20}$$

The linear interpolation above corresponds to the *optimal-transport probability path* (OT-path) in Lipman et al. (2022), which enjoys additional theoretical benefits of straighter vector fields over other options, including the variance-preserving path (Ho et al., 2020) or the variance-exploding path (Song & Ermon, 2019). During sampling, the flow ODE in Equation 17 is solved with the learned vector field to obtain the final generation during sampling. The deterministic flow ODE makes it potentially easier to guide than the stochastic Langevin dynamics or stochastic differential equations (SDEs) in the diffusion formulation.

### C.2   FUNCTIONAL FLOW MATCHING

Functional flow matching (FFM) (Kerrigan et al., 2023) further extends the conditional flow matching framework to modeling functional data — intrinsically continuous data like PDE solutions. As mentioned in the original work, the major challenge of extending the CFM framework lies in the fact that probability densities are ill-defined over the infinite-dimensional Hilbert space of continuous functions. FFM proposed to generalize the idea of flow matching to define *path of measures*

using measure-theoretic formulations. Specifically, consider a real separable Hilbert space $\mathcal{U}$ of functions $u : \mathcal{X} \to \mathbb{R}$ equipped with the Borel $\sigma$-algebra $\mathcal{B}(\mathcal{U})$. FFM learns a time-dependent vector field operator $v_t : \mathcal{U} \times [0, 1] \to \mathcal{U}$ that defines a time-dependent diffeomorphism $\psi_t : \mathcal{U} \times [0, 1] \to \mathcal{U}$ called the *flow* via the differential equation

$$\partial_t \psi_t(u) = v_t(\psi_t(u)), \quad u_0 \sim \mu_0(u), \tag{21}$$

where $\mu_0$ is some fixed noise measure from which random continuous functions can be sampled. Similar to CFM, the flow $\psi_t$ induces a push-forward measure $\hat{\mu}_t := (\psi_t)_* \mu_0$ for generative modeling. FFM demonstrates that the conditional formulation in CFM to deduce a tractable flow matching objective can also be adapted for the path of measures under some regularity conditions. In this way, also relying on the optimal transport path of measures, the FFM objective shares a similar format as the CFM loss in Equation 20:

$$\mathcal{L}_{\text{FFM}} = \mathbb{E}_{t \sim U[0,1], u_0 \sim \mu_0(u), u_1 \sim \mu_1(u)}[\|v_\theta(u_t, t) - (u_1 - u_0)\|^2], \quad u_t = (1-t)u_0 + tu_1, \tag{22}$$

where $u_t := \psi_t(u|u_1)$ is the interpolation along the *conditional* path of measures and $\mu_1$ is the target data measure. The norm is the standard $L^2$-norm for square-integrable functions. Similarly, sampling for FFM can be thought of as solving the flow ODE with the learned vector field. We demonstrate the Euler method for FFM sampling in Algorithm 1.

We also noted that Lim et al. (2023) proposed the diffusion denoising operator (DDO) as an extension of diffusion models to function spaces. DDO relies on the non-trivial extension of Gaussian measures on function spaces. Compared to DDO, FFM has a more concise mathematical formulation and better empirical generation results. Therefore, we use FFM as our generative prior.

### C.3    PRE-TRAINING FOR FFM

We follow Kerrigan et al. (2023) to use Fourier Neural Operator (FNO) (Li et al., 2020a) as the vector field operator parameterization with additional relative coordinates and sinusoidal time embeddings (Vaswani, 2017) concatenated to the noised function as inputs. Following FFM, for all 2D data (1D PDE with a time dimensional or Darcy flow), the prior noises are sampled from the 2D Gaussian process with a Matérn kernel with a kernel length of 0.001 and kernel variance of 1. For 3D data (2D NS equation), sampling from the Matérn kernel is prohibitively expensive. Though Kerrigan et al. (2023) has indicated that the white noise does not meet the regularity requirement from mathematical considerations, we empirically found that such white noise prior performed well enough. Therefore, for 3D data, we always sample from the standard white noise.

For 2D data, we use a four-layer FNO with a frequency cutoff of $32 \times 32$, a time embedding channel of 32, a hidden channel of 64, and a projection dimension of 256, which gives a total of 17.9M trainable parameters. All FFMs are trained on a single NVIDIA A100 GPU with a batch size of 256, an initial learning rate of $3 \times 10^{-4}$, and 20k iterations (approximately 1000 epochs for a 5k training dataset).

For 3D data, we use a two-layer FNO with a frequency cutoff of $16 \times 16 \times 16$, a time embedding channel of 16, a hidden channel of 32, and a projection dimension of 256, which gives a total of 9.46M trainable parameters for efficiency concerns. This model is trained on 4 NVIDIA A100 GPUs with a batch size of 24 (per GPU) for approximately a total number of 2M iterations (or 5000 epochs) with an initial learning rate of $3 \times 10^{-4}$.

It is worth noting that our proposed ECI sampling framework, as a unified zero-shot approach for guiding pre-trained flow-matching models, does not require conditional training like the conditional FFM baseline. Therefore, we only need to train a separate model for each different PDE system, but not for each different constraint. For example, the two generative tasks and the regression task on the Stokes equation are based on the same pre-trained FFM on the unconstrained Stokes problem dataset.

To evaluate the pre-training quality, we provide MMSE and SMSE between the generation by the pre-trained unconstrained FFM model and the test dataset in Table 8 as the evaluation metrics. It can be seen that most FFMs can achieve a decent approximation of the prior unconstrained distribution over the solution set with small MMSEs and SMSEs. For the 2D Darcy flow, the boundary condition has a more significant impact on the final solution as all pixel values should be shifted by the same value, leading to larger errors in the distributional properties. The generation, on the other hand, does seem reasonable.

Table 8: Mean and standard deviation MSEs between the generation by the pre-trained unconstrained FFM model and the test dataset.

| Dataset | Stokes Problem | Heat Equation | Darcy Flow | NS Equation |
|---|---|---|---|---|
| MMSE / $10^{-2}$ | 0.045 | 0.130 | 5.421 | 0.992 |
| SMSE / $10^{-2}$ | 0.039 | 0.198 | 4.121 | 0.371 |

We also provide the generative metrics of the unconstrained pre-trained FFM without any guidance as a sanity check in Table 3. It can be demonstrated that there is indeed a shift in the distribution from the generative prior learned by the unconstrained FFM, as many errors are significant. Our proposed ECI sampling can successfully capture such a shift in the distribution with significantly lower MSEs and a closer resemblance to the ground truth constrained distribution.

# D EXPERIMENTAL SETUP

In this section, we provide further details regarding the experimental setups and evaluation metrics. We also give a brief introduction to the baselines used in our experiments.

## D.1 SAMPLING SETUP

For the baseline models, we use the adaptive Dopri5 ODE solver (Dormand & Prince, 1980) for CondFFM and ProbConserv. For ECI and DiffusionPDE, we used 200 Euler steps for 2D datasets and 100 steps for 3D datasets. For D-Flow, due to the limitation of GPU memory, we use 100 Euler steps for 2D datasets and 20 Euler steps for 3D datasets. The hyperparameters for sampling are summarized in Table 4.

Generative models have intrinsic stochasticity, as the sampling procedure has randomness in the prior noise distribution. To provide a more robust evaluation of the generated results, we always generate 512 samples for every 2D dataset and baseline for computing all evaluation metrics and the sampling time in Table 4. 100 samples are generated for all the NS equation tasks (3D dataset). In this way, we aim to minimize the impact of randomness in the generation and provide a better estimation of the statistics and metrics.

## D.2 EVALUATION METRICS

As we have discussed in Section 4, for generative tasks, we calculate distributional properties like the mean squared errors of mean and standard deviation (**MMSE** and **SMSE**) to measure the errors from the ground truth statistics. We also use the constraint error (**CE**) to measure the violation of constraints.

The Fréchet Poseidon distance (**FPD**) is inspired by the metric of Fréchet Inception distance (FID) widely used in the image generation domain to evaluate the generative quality using the pre-trained InceptionV3 model (Szegedy et al., 2016). By comparing the Fréchet distance between the hidden activations of the data captured by a pre-trained discriminative model, such a score provides a measurement of the similarity between data distributions. A smaller Fréchet distance to the ground truth data distribution indicates a closer resemblance to the ground truth data and, thus, a better generation quality. Inspired by FID, we propose to use the pre-trained PDE foundation model *Poseidon* (Herde et al., 2024) to generate the hidden activations. We used the base version of Poseidon with 157.8M model parameters. Poseidon takes the initial frame, $x$-velocity field, $y$-velocity field, pressure field, and timestep as inputs and outputs predictions of the frame after the timestep. We always zeroed out the $x$-velocity field, $y$-velocity field, and pressure field in the inputs. For 2D data, the generated solutions are directly fed into Poseidon as the initial frames, and the timestep is always fixed to 0. For 3D data, each time frame of the solution trajectories is fed separately into Poseidon. The Fréchet distances are calculated separately for different time frames and are averaged to obtain the final FPD score. As we want to leverage the pre-trained Poseidon to extract high-level representations of the generated solutions, we extract the last hidden activations of the encoder for Fréchet distance calcu-

lation. The hidden activation size for the Poseidon base model is $784 \times 4 \times 4$, which we mean-pool into a 784-dimensional vector for FPD calculation.

For regression tasks, we can directly calculate the **MSE** with respect to the ground truth and the **CE** for constraint violation. Following Hansen et al. (2023), the log-likelihood (**LL**) is calculated pointwise as

$$\text{LL} = \log p \left( u_1 | \mathcal{N}(\hat{\mu}, \hat{\sigma}^2 I) \right) = -\frac{u_1 - \hat{\mu}}{2\hat{\sigma}^2} - \log \hat{\sigma} - \frac{1}{2} \log(2\pi), \qquad (23)$$

where $u_1$ is the ground truth data and $\hat{\mu}, \hat{\sigma}$ are the generation mean and standard deviation, respectively.

### D.3 CONSTRAINT ENFORCEMENT

For value constraints, we can easily enforce the constraint as described in Algorithm 4. Specifically, the initial condition is defined as the first column values after discretization, and the boundary condition is defined as the first row values. For 2D Darcy flow, the boundary condition is defined as the outer-most pixels in all four directions, thus leading to an additional resolution along each dimension. For conservation laws, the integral is approximated with the Riemann sum across the constrained region with uniform discretization along each dimension. The constraint error is also calculated using the same Riemann sum.

In the uncertainty quantification task, we follow Hansen et al. (2023) to experiment with the zero-shot superresolution setting. While the pre-trained FFM is trained on a spatiotemporal resolution of $100 \times 100$, during constrained generation, a resolution of $200 \times 200$ is instead enforced. Following Hansen et al. (2023), 100 random context points (0.25%) of the ground truth solution are provided as the value constraint, with additional conservation laws given in Appendix B.

### D.4 BASELINE MODELS

**CondFFM.**   The conditional FFM model (Kerrigan et al., 2023) assumes the family of constraints is known *a priori*. During training, the constraints are fed to the FFM model as additional information. In this way, CondFFM can only handle value-based constraints but not conservation laws. In practice, we copy the input function and set the pixels in the constrained region to the corresponding values and the pixels in the unconstrained region to zero. This condition is then concatenated to the input channel-wise as additional information. Similarly, this condition is fed into the model during the constrained sampling.

We use an almost identical FNO-based encoder for CondFFM except for one additional channel for the condition. For a fair comparison, the training hyperparameters are the same as the corresponding generative FFM. As CondFFM has additional information on the constraint in both training and sampling, we naturally expect it to exhibit better performance than other zero-shot (training-free) baselines. As a drawback for all models that require training adjustment, different CondFFM models need to be trained separately for different constraints even on the same PDE system (e.g., IC and BC for the Stokes problem), making it less flexible to different constraints.

**ANP.**   The Attentive Neural Process (ANP) (Kim et al., 2019) models the conditional distribution of a function $u$ at a specific set of target points $\{x_i\}_{i \in T}$ given another set of context points $\{x_i\}_{i \in C}$. The ANP uses the attention mechanism and variational approach to maximize the variational lower bound for the data likelihood. As it is a special case of neural processes, the ANP can take both target and context points of arbitrary sizes.

In practice, we use 100 random context points and 1000 random target points during training. The prior model is trained on the same amount of data as the corresponding FFM models. During sampling, the context points are fixed to be the values in the constraints, and all discretized spatiotemporal coordinates are used as the target points in the generation. ANP model is also zero-shot and can be applied to different value constraints.

**ProbConserv.**   ProbConserv (Hansen et al., 2023) is a general black-box posterior sampling method specifically designed for the exact satisfaction of various PDE constraints. ProbConserv adopts the gradient-free approach that directly projects the generation to the corresponding solution

space using the least squares method. When applied to value constraints, ProbConserv (the version that ensures the exact satisfaction of constraints) directly modifies the corresponding pixels to the constraint values, often leaving noticeable artifacts between the constrained and the unconstrained regions. For a fair comparison, the ProbConserv model in our experiments is built upon the same pre-trained FFM model. As ProbConserv directly operates on the final generation, it does not offer fine-grained control over the intermediate steps of the iterative sampling process of flow-matching models.

**DiffusionPDE.** DiffusionPDE (Huang et al., 2024) uses gradient guidance from the constraint loss and the PINN loss (Raissi et al., 2019; Li et al., 2024) to modify the vector field at each step to minimize these losses while maintaining the physical structure of the PDE system. This gradient-based approach is also representative of many models for inverse problems, specifically, the diffusion posterior sampling approach (Chung et al., 2022). As our PDE system may have variations in the PDE parameters, the PINN loss is not applicable. Thus, we only use the constraint loss to guide the model at each time step. In practice, we follow the original paper to test several guidance strengths from $10^2$ to $10^3$ and choose the value with the lowest MSE.

**D-Flow.** D-Flow (Ben-Hamu et al., 2024) guides the final generation via optimizing the initial noise $u_0$ rather than modifying the vector field. Assuming the constraint loss $L(\hat{u}_1)$ is differentiable, D-Flow backpropagates the gradient of the loss through the ODE solver to the initial noise as $\nabla_{u_0} L(\hat{u}_1)$, hoping such a direct optimization can reduce the final constraint loss. Such a direct gradient-based approach is extremely memory-demanding and time-consuming, as the simplest Euler solver requires about 100 steps to achieve decent generation results. We follow the original work to use the LBFGS optimizer with a learning rate of 1 and 20 maximum iterations.

**FNO.** The Fourier Neural Operator (FNO) (Li et al., 2020a) is used as the additional baseline in regression tasks of neural operator learning. We use an almost identical architecture as the encoder for FFM, except for the missing time embedding part. Similar to CondFFM, the pixels in the constrained region are set to the constraint values, while other pixels are set to zero. For a fair comparison, the FNO model tries to predict all the pixel values for the solution domain, including the constrained ones already given as the input. As the regression models are easier to overfit, we apply early-stopping techniques to choose the best model checkpoint for evaluation. The FNO was trained for 10k iterations for the Stokes problem and 500k iterations for the NS equation when the valid loss plateaued.

# E   ADDITIONAL RESULTS AND VISUALIZATIONS

In this section, we provide additional experimental results and ablation studies to further demonstrate the effectiveness of our proposed ECI sampling scheme. Additional visualizations of the generations and evaluation metrics are also provided in this section.

## E.1   GENERATIVE TASKS

### E.1.1   ADDITIONAL VISUALIZATION

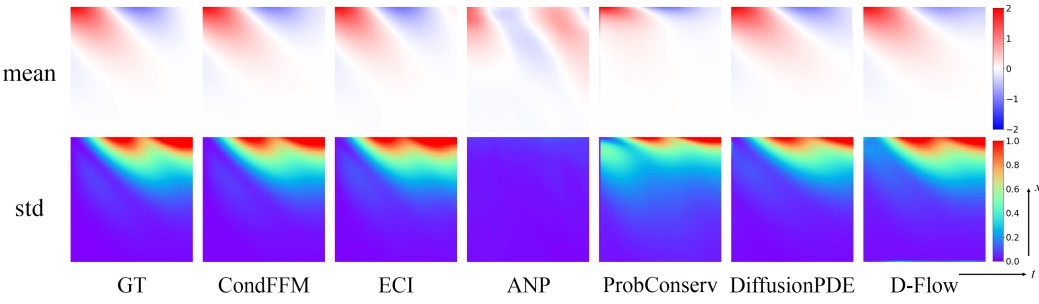

Figure 8: Generation mean and standard deviation for the Stokes problem with IC fixed.

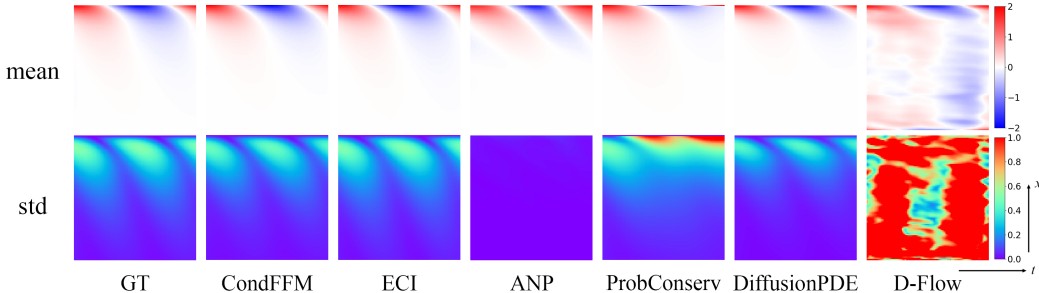

Figure 9: Generation mean and standard deviation for the Stokes problem with BC fixed.

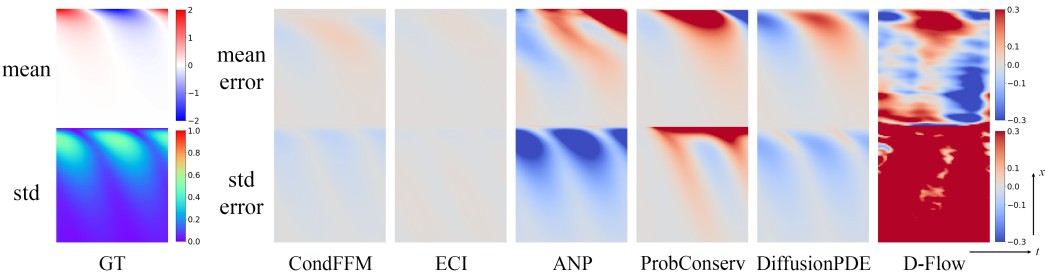

Figure 10: Generation mean and standard deviation errors for the Stokes problem with BC fixed.

In addition to the error plots in Figure 2 and 3, we further provide visualizations of the generation statistics in Figure 8 (Stokes problem with IC fixed), 9 (Stokes problem with BC fixed), 11 (heat equation), 13 (Darcy flow), and 15 (2D NS equation).

Errors for generation statistics for different systems are also provided in Figure 10 (Stokes problem with BC fixed), 12 (heat equation), and 14 (Darcy flow). It can be demonstrated more clearly that gradient-based methods like DiffusionPDE and D-Flow had more artifacts around the boundary and did not satisfy the constraint exactly. For ProbConserv, though its gradient-free approach indeed ensured the exact satisfaction of the constraint, the non-iterative control of the prior flow model left noticeable artifacts between the constrained and the unconstrained regions. Similar trends can also be observed in other PDE systems.

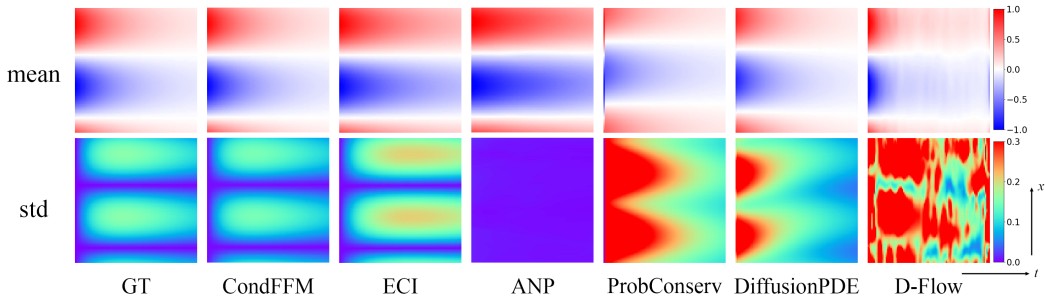

Figure 11: Generation mean and standard deviation for the heat equation with IC fixed.

We also noticed that D-Flow tended to lead high-variance generations. This is probably because a large number of Euler steps leads to an exceedingly complex dynamic that is difficult to optimize. Also, note that directly taking the gradient with respect to the initial noise may break the initial structure sampled from the Matérn kernel and may lead to initial noises never seen by the model during training. These two reasons may also account for the better performance of the D-Flow for the NS equation, in which we used fewer Euler steps and the standard white noises as the prior noise functions.

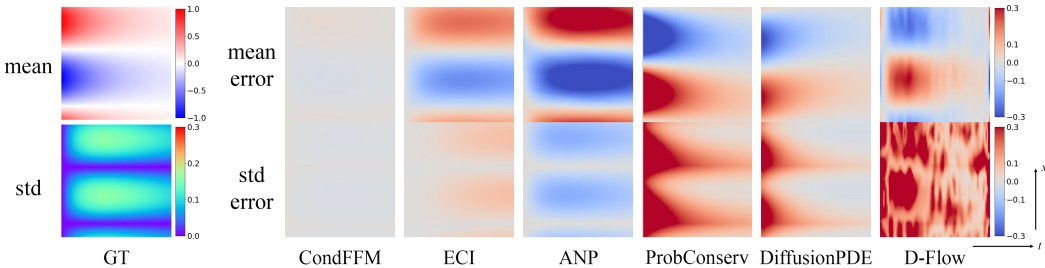

Figure 12: Generation mean and standard deviation errors for the heat equation with IC fixed.

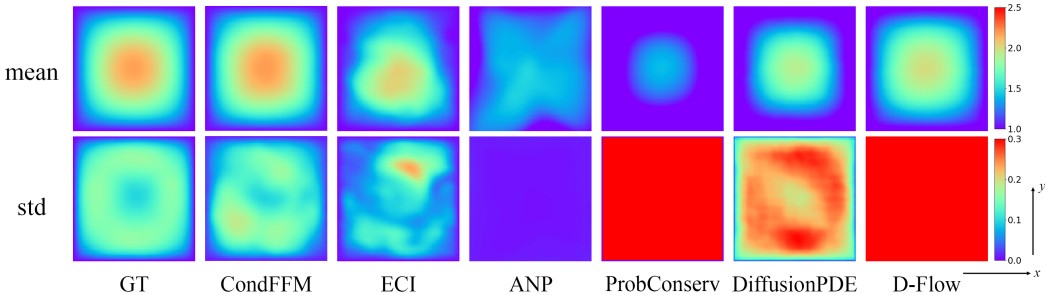

Figure 13: Generation mean and standard deviation for the Darcy flow with BC fixed. Note that the scaling range is different and large variance values are cropped for a better comparison.

## E.2 STOKES PROBLEM WITH DIFFERENT CONSTRAINTS

We provide additional generative evaluation on the Stokes problem with various ICs/BCs in Table 9 in addition to the results in Table 3. Recall that during unconstrained pre-training, we varied the PDE parameter as $\omega \sim U[2, 8], k \sim U[2, 20]$, and in Table 3 we fixed $k = 5$ or $\omega = 6$. Combined with the results Table 9, it can be clearly demonstrated that our proposed ECI sampling can achieve consistent performance improvement over the baselines in a zero-shot manner with the flexibility to all different IC settings $k = 5, 10, 15$ and BC settings $\omega = 4, 6, 8$.

### E.2.1 GENERATIVE SUPERRESOLUTION

In Table 10, we provide the evaluation results under the zero-shot superresolution setting, where the generation resolution is $200 \times 200$, four times larger than the training set. It is interesting to see that the conditional FFM is the most sensitive method in such a zero-shot setting, probably due to the hard-coded connection of the masked input as the condition.

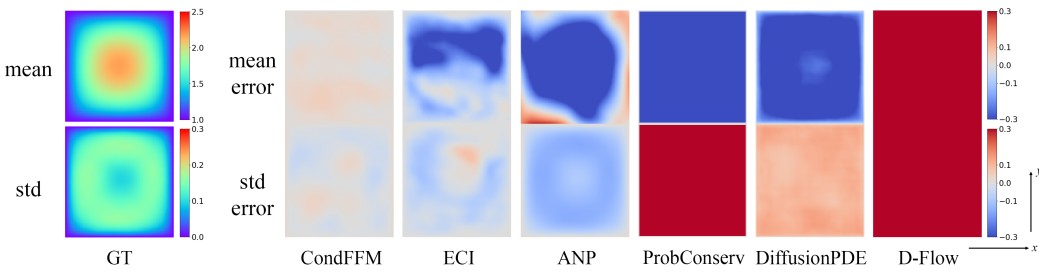

Figure 14: Generation mean and standard deviation errors for the Darcy flow with BC fixed. Note that large error values are cropped for a better comparison.

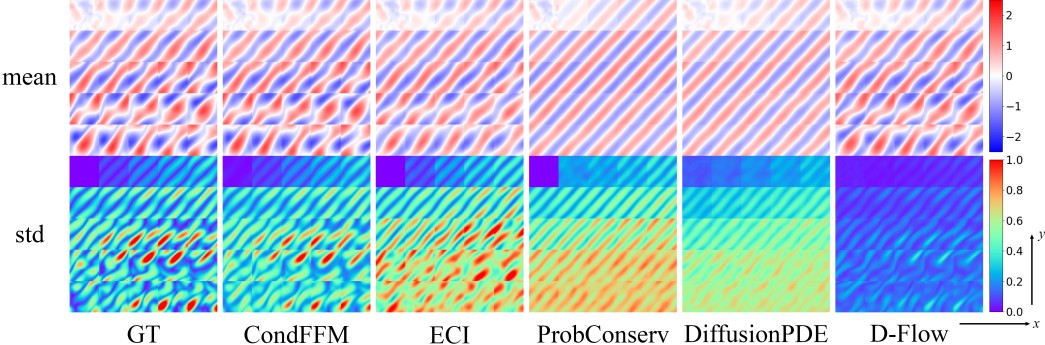

Figure 15: Generation mean and standard deviation for the 2D NS equation with IC fixed.

Table 9: Generative metrics on the Stokes problem with different constraints. The best results for zero-shot methods (CondFFM is not zero-shot) are highlighted in bold.

| Dataset | Metric | ECI | ANP | ProbConserv | DiffusionPDE | D-Flow | CondFFM |
|---|---|---|---|---|---|---|---|
| Stokes IC $k = 10$ | MMSE / $10^{-2}$ | **0.113** | 4.616 | 1.792 | 0.512 | 0.774 | 0.307 |
| | SMSE / $10^{-2}$ | **0.124** | 12.400 | 0.356 | 0.301 | 59.377 | 0.060 |
| | CE / $10^{-2}$ | **0** | 13.032 | **0** | 0.068 | 3.004 | 0.011 |
| | FPD | **0.156** | 6.694 | 3.619 | 0.340 | 11.013 | 0.454 |
| Stokes IC $k = 15$ | MMSE / $10^{-2}$ | 0.190 | 11.148 | 0.797 | 0.216 | **0.088** | 0.125 |
| | SMSE / $10^{-2}$ | **0.183** | 3.354 | 2.751 | 0.470 | 12.063 | 0.010 |
| | CE / $10^{-2}$ | **0** | 18.278 | **0** | 0.194 | 13.149 | 0.010 |
| | FPD | **0.105** | 6.509 | 6.425 | 0.832 | 4.745 | 0.020 |
| Stokes BC $\omega = 4$ | MMSE / $10^{-2}$ | **0.676** | 9.273 | 3.034 | 1.488 | 1.728 | 0.031 |
| | SMSE / $10^{-2}$ | **0.378** | 2.777 | 2.462 | 2.278 | 6.343 | 0.081 |
| | CE / $10^{-2}$ | **0** | 115.335 | **0** | 39.99 | 14.553 | 0.012 |
| | FPD | **1.671** | 5.309 | 1.686 | 3.954 | 10.043 | 0.071 |
| Stokes BC $\omega = 8$ | MMSE / $10^{-2}$ | **0.042** | 4.115 | 9.008 | 5.794 | 3.801 | 0.028 |
| | SMSE / $10^{-2}$ | **0.026** | 2.462 | 2.710 | 3.288 | 22.376 | 0.010 |
| | CE / $10^{-2}$ | **0** | 73.916 | **0** | 160.397 | 51.056 | 0.017 |
| | FPD | **0.109** | 6.663 | 0.843 | 9.356 | 13.202 | 0.111 |

### E.2.2 ADDITIONAL METRICS FOR NS EQUATION

We provide the per-frame FPD scores for the NS equation generation results in Figure 16. We noted that ECI sampling tended to have more accurate generations for frames close to the initial condition, with the FPD scores increasing with the time frames. In gradient-based models of DiffusionPDE and D-Flow, we also noticed a peak in the FPD scores at an early stage, which should have been

Table 10: Generative metrics on the Stokes problem with the zero-shot superresolution setting to $200 \times 200$. The best results are highlighted in bold.

| Dataset | Metric | ECI | ANP | ProbConserv | DiffusionPDE | D-Flow | CondFFM |
|---|---|---|---|---|---|---|---|
| Stokes IC | MMSE / $10^{-2}$ | **0.274** | 14.120 | 4.338 | 5.109 | | 4.242 |
| | SMSE / $10^{-2}$ | **0.515** | 8.720 | 2.933 | 4.167 | OOM | 4.152 |
| | CE / $10^{-2}$ | **0** | 3.563 | **0** | 5.392 | | 12.508 |
| | FPD | **2.647** | 26.965 | 15.977 | 19.654 | | 24.798 |
| Stokes BC | MMSE / $10^{-2}$ | **0.863** | 2.819 | 5.223 | 1.610 | | 4.669 |
| | SMSE / $10^{-2}$ | **0.418** | 2.870 | 1.491 | 1.768 | OOM | 2.706 |
| | CE / $10^{-2}$ | **0** | 29.168 | **0** | 0.068 | | 21.567 |
| | FPD | **1.903** | 5.282 | 2.021 | 3.830 | | 9.504 |

easier for the model. As expected, CondFFM outperformed all zero-shot guidance models by a large margin in such a complex task for almost all frames.

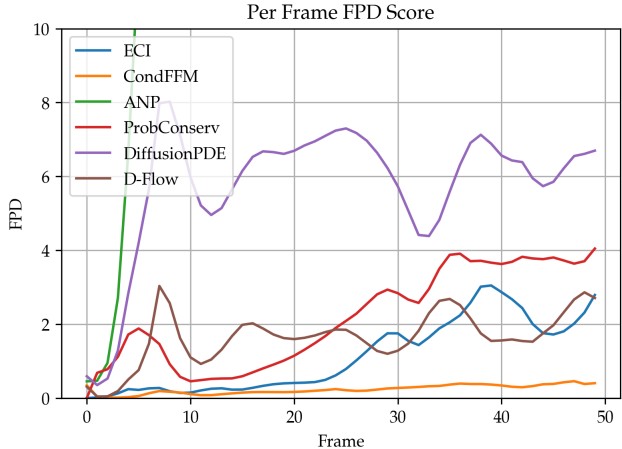

Figure 16: Per frame FPD score for the NS equation with IC fixed.

### E.3 REGRESSION TASKS

#### E.3.1 UNCERTAINTY QUANTIFICATION

For the uncertainty quantification tasks, the specific snapshot at the test time as described in Appendix B (following the same settings in (Hansen et al., 2023)) is plotted in Figure 17, with the generation mean and 3 times of the generation standard deviation as the confidence interval. Our results share the similar trend described in (Hansen et al., 2023), in which they also observed more uncertainty with the increasing difficulty from the heat equation, PME, to the Stefan problem. Specifically, the solution in the Stefan problem has a shock in the function value, which leads to higher uncertainty around the shock position. For other linear parts of the solution, our ECI sampling generation is able to achieve quite good predictions with low variance and low uncertainty.

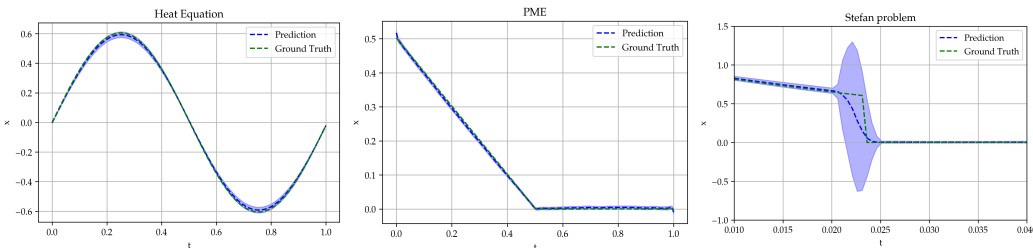

Figure 17: Uncertainty quantification visualization results for three instances of the Generalized Porous Medium Equation (GPME). The $3\sigma$ confidence intervals are plotted.

#### E.3.2 NEURAL OPERATOR LEARNING

For NO learning tasks, we provide additional visualizations of the generation statistics and the reference FNO predictions in Figure 18 (Stokes problem) and 19 (NS equation). It can be demonstrated that, though not directly trained on such regression tasks, our ECI sampling naturally produces generation results with low variance in such scenarios. In comparison, other models often have a variance scale similar to those in the generative tasks, making them less suitable for such zero-shot regression tasks.

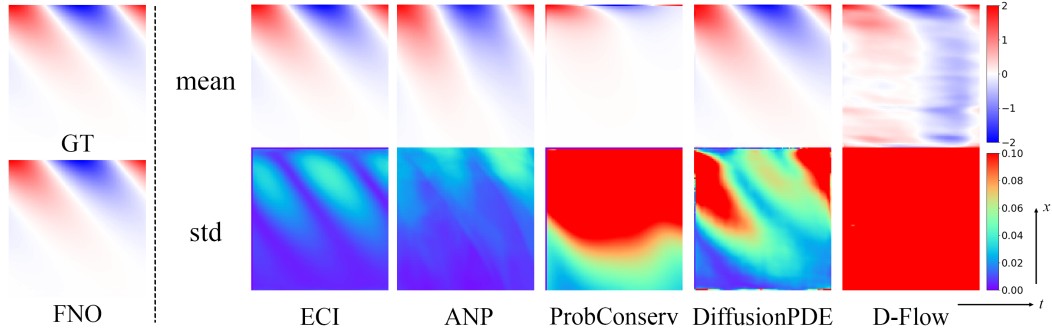

Figure 18: Regression mean and standard deviation for the Stokes problem with both IC and BC fixed. Note that the scaling range for the standard deviation is different from that in the generative task.

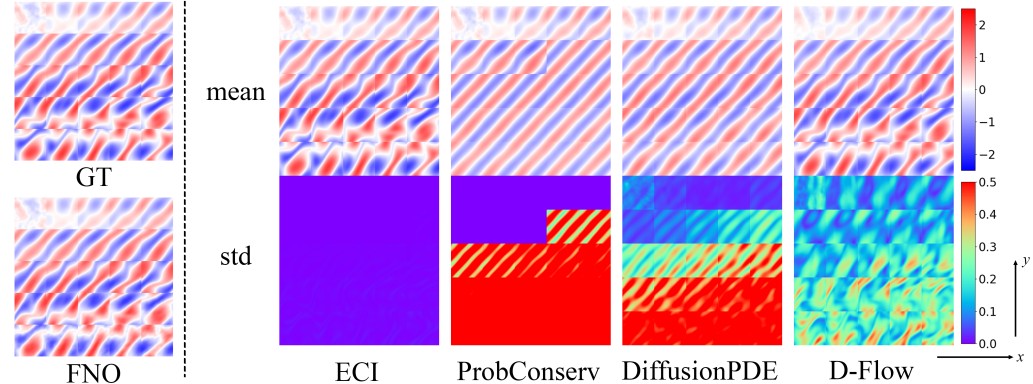

Figure 19: Regression mean and standard deviation for the NS equation with the first 15 frames as the constraint. Note that the scaling range for the standard deviation is different from that in the generative task.

### E.4 MORE ABLATION STUDIES

#### E.4.1 STOKES PROBLEM FPD

To provide a more thorough investigation into the impact of mixing iterations $M$ and re-sampling interval $R$ on the final performance, we further calculate the FPD scores in Figure 20 for the two settings for the Stokes problem in addition to Figure 4. The FPD scores are more consistent than the MMSE and SMSE. For the Stokes IC problem with higher variance, generations with no re-sampling work the best, with the FPD scores peaking at 10 mixing iterations. For the Stokes BC problem with lower variance, the best performance combination of hyperparameters has a diagonal pattern with more mixing iterations pairing best with a smaller re-sampling interval. Such results are also consistent with our heuristic suggestions on choosing the best $M$ and $R$ for different tasks with different variances.

#### E.4.2 PRIOR GENERATIVE MODEL

We further provide the ablation studies on the impact of the quality of the prior generative model. Intuitively, a stronger generative prior that better captures the underlying information of the PDE system can be easier to guide. The results are demonstrated in Table 11, with checkpoints from 5k, 10k, and 20k training iterations. A clear trend of better guided-generation performance can be observed in both IC and BC constraints. In this way, the quality of FFM indeed plays an important role in our ECI sampling framework.

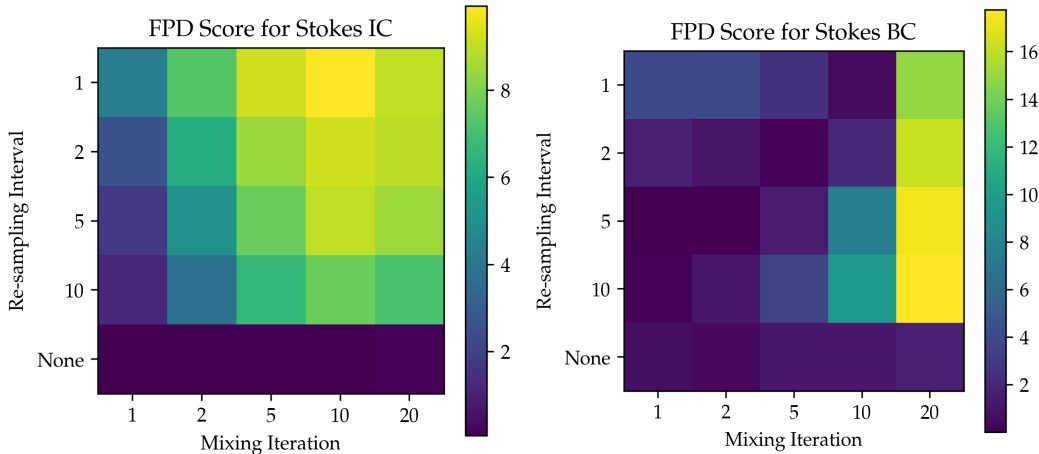

Figure 20: FPD score of ECI sampling with different mixing iterations and re-sampling intervals in the Stokes problem with IC (left) or BC (right) fixed.

Table 11: Generative metrics on the Stokes problem with different constraints using the different FFM checkpoints with different training iterations.

| Dataset | Stokes IC | | | Stokes BC | | |
|---|---|---|---|---|---|---|
| #iter | 10k | 15k | 20k | 10k | 15k | 20k |
| MMSE / $10^{-2}$ | 1.273 | 0.601 | 0.090 | 0.888 | 0.011 | 0.005 |
| SMSE / $10^{-2}$ | 1.693 | 0.177 | 0.127 | 0.868 | 0.015 | 0.003 |
| FPD | 1.863 | 1.031 | 0.076 | 3.101 | 0.009 | 0.010 |

### E.4.3 AMOUNT OF CONSTRAINT INFORMATION

We also provide additional ablation studies on ECI sampling behavior as a regression model. Specifically, we explore the regression losses when more information is provided in the constraint. Intuitively, serving as a deterministic regression model, the ECI generation should exhibit fewer errors and less variance with more information provided. In our experimental setup, we fixed the number of mixing iterations to 5 and the re-sampling interval to 1. To control the information in the constraint, we varied the number of initial frames fed into the model as the value constraint from 5 to 25. As our ECI sampling guarantees the exact satisfaction of the constraint, we instead calculated the mean MSE and standard deviation MSE based on the unconstrained region for a fair comparison between different settings. The losses are plotted in Figure 21, in which a clear and almost linear trend in the log scale can be distinguished for both metrics as the number of constrained frames increases. Similarly, the visualization of the mean error and the logarithm of the standard deviation are provided in Figure 22. The model made more accurate predictions with less uncertainty as the number of constrained frames increased. The regions with large mean errors also matched those with large variance (uncertainty). In this sense, our ECI sampling provides a bonus to transform the pre-trained unconditional FFM model into a decent zero-shot uncertainty quantification model with potentially wider applications.

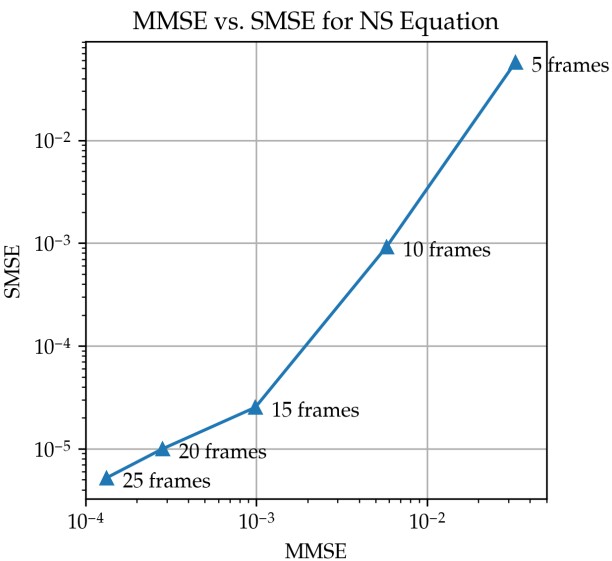

Figure 21: Regression MMSE and SMSE for the NS equation with different numbers of frames fixed as the constraints. Notice a clear and almost log-linear trend for the decrease in both metrics as the number of constrained frames increases.

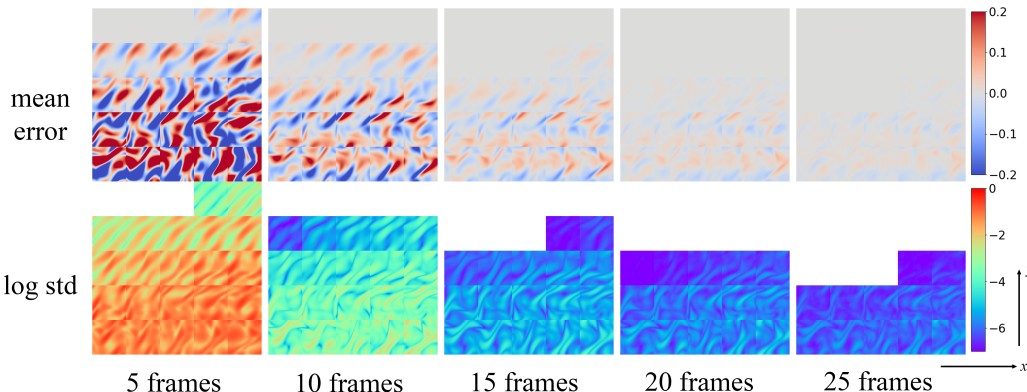

Figure 22: Regression error of the mean and logarithm of the standard deviation for the NS equation with different numbers of frames fixed as the constraints. We use the same scale for plotting different settings.

