# OpenReview forum: "Gradient-Free Generation for Hard-Constrained Systems"
_ICLR.cc/2025/Conference — ICLR 2025 Poster_

### Official Review · Reviewer_11X9 · 2024-10-31

**Soundness:** 2
**Presentation:** 1
**Contribution:** 2
**Rating:** 5
**Confidence:** 3

**Summary:**

The authors introduce a novel framework for adapting pre-trained, unconstrained
generative models to exactly satisfy constraints of PDEs in a zero-shot manner.
Sampling which alternates between extrapolation (E), correction (C), and interpolation
(I) stages during each iterative sampling step to ensure accurate integration of constraint information of PDEs
is proposed
and called ECI sampling.
The new method is evaluated against various existing approaches and for different PDEs with both value constraints
(initial and boundary conditions) and conservation conditions.
The topic is interesting, but the paper requires a complete rewriting focusing on the main contribution and towards a sound mathematical notation.

**Strengths:**

- interesting and relevant research question

- extensive evaluation of the new ECI sampling method versus other zero-shot guidance models for both generative and regression tasks
and experiments on various 1d/2D and one 3D PDE;
however I am not an expert in all these PDE learning methods

**Weaknesses:**

The writing of the paper is  unsatisfactory, everything boils down to Alg. 2 together with the
beginning of p. 5 (Extrapolation, Correction, Interpolation),
but the authors
try to explain something in a non understandable, fuzzy way.
In particular, often the notation is not introduced when it is needed.
For example in formula (2) it is not clear how $q$ is defined.

 The paper could be considerable shortened or better some parts from the appendix could be moved to the main part
to make the methods better understandable.\\
Prop 1 is straightforward and holds just by construction; it could go into the main part

 Alg. 2  is obscured by notational
overload. Basically what is happening in Algorithm 2, line 6 is the
following:

- drawing $u_1\sim p_\theta(u_1|u_t^{m-1})$ amounts to computing
$$u_1=u_t^{m-1}+ (1-t)v_{t,\theta}(u_t^{m-1})$$ In particular
$p_\theta(u_1|u_t^{m-1})$ is a delta measure resulting from the one step
application of the vector field.
- then compute $\hat{u}_1:=C(u_1,\mathcal{G})$ in order to project to
a function $\hat{u}_1$ that satisfies the constraints

- sample from $q(u_t^m|\hat{u}_1)$ by sampling from $u_0\sim\mu_0$ and
calculating $u_t^m=(1-t)u_0+t\hat{u}_1$. This means that one assumes that
$q(u_t^m|\hat{u}_1)$ is distributed as $(1-t)U_0+ \hat{u}_1$ where
$U_0\sim \mu_0$.

In line 7 the only difference is that in the last step one computes
$u_{t+1\backslash N}=(1-(t+1\backslash N))u_0+(t+1\backslash N)\hat{u}_1$.

**Questions:**

Minor remarks:

- p. 3: PDE family is $\mathcal F_\phi$, but then $\mathcal U_{|\mathcal F}$ without $\phi$
(skip the $x$ in $u(x)$ in definition of $\mathcal U$ since you mean the function and not the function value);
the family could be $\mathcal F:=\{\mathcal F_\phi: \phi \in \Phi\}$

- in the definition of $\mathcal U_{|\mathcal G}$ I am missing the domain of $x$ in $\mathcal F_\phi u(x) = 0$

- what is the reason for the clumsy notation, why not calling it just $\mathcal U_{\mathcal G}$ and $\mathcal U_{\mathcal F}$

- there are errors in English, e.g. and of p. 3 ''denote'' instead of ''denotes''

- when the authors use $:=$ (push-forward measure) and when not (most of all other definitions)

- formula (2) is not readable since $q$ is not defined here; later the authors try to explain the formula

- put important formulas on an extra line

- While Alg 1 is superfluous, how Alg 3 and 4 are embedded in Alg 2

---

> ### Author Response · Authors · 2024-11-13
> **Rebuttal to 11X9**
>
> We thank your recognition of our work's **extensive evaluations** and your constructive suggestions to improve the presentation of our work. We apologize for some not-well-explained notations with our assumption that all reviewers were familiar with generative models, especially flow matching. We have included a more detailed mathematical introduction to flow matching in the common rebuttal session, and we will further elaborate on some additional details to address your questions and concerns.
>
> ## Q1 Notations
>
> We first clarify some notations in addition to Section 3.1. We use $u:\mathcal{X}\subseteq\mathbb{R}^d\to \mathbb{R}$ to denote a continuous function as the solution. The subscript always indicates the time step in the flow. We follow the flow matching convention such that $t=0$ corresponds to the noise and $t=1$ corresponds to the target (data). As the space of continuous functions has an infinite dimensionality, we follow FFM [1] to adopt measure-theoretical terms of $\mu_0,\mu_1$ as probability measures over the continuous functions, instead of using the common $p_0,p_1$ in computer vision domains. All $p,q$ in our work are probability densities defined as the  Radon–Nikodym derivative (see the FFM paper for technical details). Specifically, at each time step $t$, the noised data $u_t:=\psi_t(u_0|u_1)$​​ can be obtained with *interpolation*. We noted that **all of the above definitions are standard notations in flow matching context**.
>
> We do agree Equation 2 is less well-defined as we postponed the definitions in the bulletin list later. We will **modify the order in our revised manuscript** to first introduce and define the notations to avoid potential confusion. We will also follow your suggestion to move Proposition 1 to the main text to make it more understandable.
>
> ## Q2 Algorithm
>
> Your understanding of our Algorithm 2, the core algorithm for ECI sampling, is definitely correct. We originally put it in a more concise way mainly because **all notations are quite standard in diffusion or flow-based generative models**. For example, [2, 3] also adopts an iterative formulation for guiding diffusion models with a double loop. We will consider reformatting the algorithm to include more details following your outline.
>
> Still, we would like to mention that both extrapolation and interpolation stages for flow matching have clear definitions with respect to the probability path induced by the flow. Specifically, the linear interpolation corresponds to the *OT-path* (OT for optimal transport) in flow matching literature [4] with theoretical benefits of optimal transport. It is **not our whimsical assumption but theoretically a good option** in choosing the probability paths.
>
> Also, due to the iterative nature of flow sampling, ECI sampling needs to advance the solver for the flow ODE (see common rebuttal session). Therefore, the last step of advancing the solver to a large $t'=t+1/N$​ is also necessary.
>
> ## Q3 Other Remarks
>
> - Regarding $\mathcal{F}_\phi$. Your understanding is correct. We will clarify it.
> - Regarding the domain of $x$. We assume the normal domain of $\mathcal{X}$ for $\mathcal{F}_\phi$.
> - We used the notation $\mathcal{U_{|F}}$ to indicate the solution set is *restricted* by the constrained to a narrower subset.
> - We used $:=$ for definition (of new quantities). For the push-forward, we use the standard notation of $(\psi_t)_*$​, e.g., the second part of Section 3.1.
> - For the correction algorithms in Algorithm 3&4, they serve as concrete examples of the correction $C(u_1,\mathcal{G})$ in Equation 3&4 and Algorithm 2.
>
> We also thank your other suggestions regarding the notations and formatting for better conciseness and understandability. We will add more clarifications on notations, more mathematical backgrounds, and better presentations in our revised manuscript to make our work more accessible to a larger audience.
>
>
>
> [1] Kerrigan, Gavin, Giosue Migliorini, and Padhraic Smyth. "Functional flow matching." *arXiv preprint arXiv:2305.17209* (2023).
>
> [2] Lugmayr, Andreas, et al. "Repaint: Inpainting using denoising diffusion probabilistic models." *Proceedings of the IEEE/CVF conference on computer vision and pattern recognition*. 2022.
>
> [3] Kawar, Bahjat, et al. "Denoising diffusion restoration models." *Advances in Neural Information Processing Systems* 35 (2022): 23593-23606.
>
> [4] Lipman, Yaron, et al. "Flow matching for generative modeling." *arXiv preprint arXiv:2210.02747* (2022).

---

> > ### Comment · Reviewer_11X9 · 2024-11-17
> > **So far there is no update of the paper. But the authors did a lot of comments what they want to do and what ALL reviewers do not understand, acknowledging at the same time that my consistent rewriting of their main algorithm 2 was completely correct.**
> >
> > First let my emphazise that my confidence level 3 was on PDEs, not on generative modeling and in particular flow matching,
> > where I am confident 5.
> > 1. My concern was not that the authors use standatd notations, but they do not do this in a unfied way.
> > 2. As the authors acknowledge my understanding of Alg 2, which is a main modeling contribution of the paper, was absolutely correct. Still the main contribution was hidden in the bad writing style of the paper.
> > I do not understand what the authors want to say me with their comment, since I said nothing in this direction.
> > ''
> >  Sadly, we note a common misunderstanding and misinterpretation among all the reviewers and we would first like to clarify this. Our proposed ECI sampling framework is NOT a regression model (e.g., FNO [1]), nor a plugin for existing regression models (e.g., BOON [2]). Instead, ECI sampling is a generative framework for functional data based on flow matching.We believe all reviewers have misinterpreted this setting and have asked for a comparison with regression models or using regression metrics, which are not applicable to our problem setup. We do apologize for our assumption that this paper's audience is familiar enough with generative models, especially flow matching. We now add a more detailed introduction to flow matching here to make our paper more accessible to the audience in the SciML domains.''
> > 3. So far there is no revision available, where I see my concerns addressed. I know the references!

---

> > > ### Author Response · Authors · 2024-11-18
> > > **Updated manuscript**
> > >
> > > We sincerely thank you for your timely response regarding our rebuttal, and we are more than happy to continue such a constructive dialogue with you. In the past few days, we have been working on running additional experiments and modifying our manuscript to a better presentation.
> > >
> > > We appreciate your detailed suggestions on the improvements to the presentation of our work. We have taken your suggestion on the notations, algorithms, and motivations to significantly **refactor Section 3.2 in a more concise, clear, and accessible way**. Specifically, we follow your suggestion to add an **additional algorithm** called the *ECI step* and redefine the notations in a more unified and concise way prior to using them. Other suggestions are also reflected in our revised manuscript. Please refer to our **general rebuttal** for other major changes.
> > >
> > > We would also like to clarify a few things regarding your new response. We mentioned the common issues regarding our contributions in generative modeling as it was not highlighted in your summary or strength. However, as you have mentioned that your confidence for the generative model is 5, we are glad you understand our framework in the generative domain and reserve the previous common rebuttal for reviewer rhkG.
> > >
> > > In this regard, we have carefully followed your suggestions to update the manuscript, and we believe our updated manuscript has addressed your current concerns about the presentation of our work. We are happy to address any further concerns of yours.

---

> > > > ### Comment · Reviewer_11X9 · 2024-11-23
> > > > **Second answer**
> > > >
> > > > ,,We mentioned the common issues regarding our contributions in generative modeling as it was not highlighted in your summary or strength. ''
> > > >
> > > > I still do not understand this way of rebuttal. Note that in you bold contribution points you neither wrote that a main contribution is the incorporation of generative models.
> > > >
> > > > I had a look at the revision.
> > > > Unfortunately, the authors did not wrote the new parts in another color so that it would take some effort to really see the differences. I do not appreciate the hint t ,,our general rebuttal for other major changes''.
> > > > So I seems that they somehow strengthen the paper towards the novel contribution which was one of my concerns.
> > > > So I update my score to 5, but not more, since  I still think that the novelty impact is very moderate.

---

> > > > > ### Author Response · Authors · 2024-11-25
> > > > > **Thanks for your decision to raise the score**
> > > > >
> > > > > We first thank your recognition of our updated rebuttal and your willingness to raise the score. We further clarify some points as follows, hopefully further addressing your remaining concerns.
> > > > >
> > > > > Regarding our contributions, we confine our work to guiding pre-trained **flow matching** models, a specific class of generative framework that has drawn wide attention recently with its success. We noticed that none of the initial summaries mentioned the term *flow*, so we further clarified this to avoid potential misunderstanding. Given your response to our rebuttal, we now fully believe you did not misunderstand our work, and we instead focused more on updating our revised manuscript to include more experimental results and a more concise and clear way of delivering our method.
> > > > >
> > > > > Again, we sincerely thank your detailed review to help improve our work and your recognition of our contributions. We noted that, however, **your score stays at 3, not 5 as you have mentioned**. We would appreciate it if you could modify the score instead.

---

> > > > > > ### Comment · Reviewer_11X9 · 2024-11-26
> > > > > >
> > > > > > tried to raise the score to 5 in the system - for unknown reasons thsi does not work.

---

### Official Review · Reviewer_KM5e · 2024-11-01

**Soundness:** 3
**Presentation:** 2
**Contribution:** 2
**Rating:** 5
**Confidence:** 4

**Summary:**

This work proposes an ECI sampling framework that strictly satisfies hard constraints such as IC/BCs and conservation laws through extrapolation, correction, and interpolation stages. The method has gradient-free generation and zero-shot inference for parametric PDE and dynamical systems.

**Strengths:**

1. There are many experimental results, sufficient comparison with baselines, and complete test metrics.
2. The analysis presented in line 260 is basically confirmed by experiments: when the physical constraints are strong or contain a lot of information, the generation variance is small, and conversely the variance is large.
3. Compared with Boundary enforcing Operator Network (BOON), this work implements hard constraints more concisely. The ECI sampling framework satisfy physical laws and system requirements strictly.

**Weaknesses:**

1. Although the results in Table 3 is excellent, Figures 6-8 may not clearly reflect the advantages of ECI sampling, and further explanation of the figures and settings is needed.
2. This method can achieve zero-shot performance and exact satisfaction of different IC/BCs. However, Figures 5-7 are shown with IC or BC fixed. It is necessary to supplement the visualization of final predictions under different ICs. For the Stokes problem in Appendix B.1, we recommend testing more k and w values to demonstrate the generalization and applicability of ECI sampling.
3. There is a lack of an overall framework diagram, and the entire training and generating process is not clear or intuitive enough. The PDE problems considered in the experiment are limited to 2-3 dimensions. And those experimental descriptions, data formats and prediction tasks should have a clearer and unified presentation.

**Questions:**

For pre-trained operator networks, fusing BC/ICs to enforce hard constraints can indeed improve predictions at initial time and boundaries, but can ECI sampling directly correct or improve the prediction values in the domain? And how effective is this ability under zero-shot inference?
In addition, u_1 in line 2 of Algorithm 4 should be in brackets and needs to be corrected.

---

> ### Author Response · Authors · 2024-11-13
> **Rebuttal to KM5e**
>
> We thank your recognition of our work's **comprehensive experimental results** and **concise formulation**. We are happy to address your questions and concerns. Still, we noted the potential misinterpretation of our proposed approach as a regression model but not as a generative model. See the common rebuttal for more details, clarifications, and mathematical backgrounds.
>
> ## Q1 Results in Figure 6-8
>
> Figure 6-8 demonstrates the generation statistics of point-wise mean and standard deviation, together with the ground truth statistics. Similar to Figure 1&2, a closer resemblance to the ground truth statistics indicates a better generation result. It is worth noting, though, that MMSE and SMSE only reflect the first two moments of the distribution but not higher-order information. Therefore, FPD scores should be preferred, in which the pre-trained PDE foundation model Poseidon is used to extract the hidden representations (see Appendix D.1). We have included the detailed experimental setups in Appendix B and we will add more explanations for Figure 6-8.
>
> ## Q2 Experiments with different IC/BC
>
> Thanks for the insightful suggestion. We are running experiments on the Stokes dataset with different IC/BC values to further demonstrate the generalizability and effectiveness of our approach. We will update the results once they are readily available.
>
> ## Q3 Overall framework
>
> The overall framework of our ECI sampling can be understood as a modification to the vanilla flow sampling process in Algorithm 1. With minimum modifications, we demonstrate the ECI sampling in Algorithm 2. Previous work in guiding diffusion models also adopts an iterative formulation [1, 2], which is in fact **standard in the context of diffusion or flow-based generation models**. We understand that the reviewer may not be familiar with the generative models and we have provided more details in the common rebuttal session.
>
> Regarding the datasets used in our work, we noted all datasets and their spatiotemporal resolution setups are directly **taken from previous papers** (see Appendix B for reference of the datasets in previous papers). Therefore, we believe such an experimental setup is **standard, consistent, and comparable with existing works**. To the best of our knowledge, **4D datasets in PDE systems do not exist so far**. It is also a **standard practice** to test on 2D and 3D datasets in the PDE domain, e.g., neural operator learning [3, 4], foundation models [5, 6], and existing constrained generation [7, 8]. Therefore, we believe we have already provided enough evidence of the superior performance of our proposed method.
>
> We already included the dataset descriptions, specifications, and task formulation in Table 2&7, Appendix B&D.2, for which our ECI sampling serves as a unified constrained generation framework.
>
> ## Q4 Improvement in the whole domain
>
> We note that our ECI sampling was designed to offer **iterative control** in each flow sampling step to mitigate the artifacts between the constrained and unconstrained regions. Indeed, our generation result demonstrated **remarkable consistency in the whole spatiotemporal domain**, with hardly noticeable artifacts around the boundary in Figure 1, 6, 7, and 8. This is in sharp contrast with the naive projection algorithm like ProbConserv, which can be only applied to the final prediction without fine-grained control over each sampling step. In the close-up demonstration in Figure 5, noticeable artifacts can be found for ProbConserv and other gradient-based approaches.
>
> The iterative control of ECI sampling that gradually pushes the intermediate noised data into generations with hard constraints can be further demonstrated in Figure 4. Note how the generation trajectory gradually exhibits more consistency between the constrained and unconstrained regions.
>
> ## Q5 Zero-shot effectiveness
>
> We emphasize that **all experiments are carried out in a zero-shot manner** except for the conditional FFM baseline. Specifically, an unconstrained FFM was first trained as the generative prior. Different approaches including our ECI sampling were then applied in a zero-shot manner (see Appendix C&D.2). In this way, our framework provides a **unified approach for different constraints** without finetuning or conditional training (e.g., IC and BC in the Stokes problem). The generative evaluation metrics also demonstrated the state-of-the-art performance across all zero-shot models (conditional FFM is not zero-shot and should be viewed as an upper bound, as we have discussed in Section 4.1).

---

> > ### Author Response · Authors · 2024-11-13
> > **Rebuttal to KM5e (Reference)**
> >
> > [1] Lugmayr, Andreas, et al. "Repaint: Inpainting using denoising diffusion probabilistic models." *Proceedings of the IEEE/CVF conference on computer vision and pattern recognition*. 2022.
> >
> > [2] Kawar, Bahjat, et al. "Denoising diffusion restoration models." *Advances in Neural Information Processing Systems* 35 (2022): 23593-23606.
> >
> > [3] Li, Zongyi, et al. "Fourier neural operator for parametric partial differential equations." *arXiv preprint arXiv:2010.08895* (2020).
> >
> > [4] Li, Zongyi, et al. "Anima Anandkumar, Physics-informed neural operator for learning partial differential equations." *arXiv preprint arXiv:2111.03794* (2021).
> >
> > [5] Herde, Maximilian, et al. "Poseidon: Efficient Foundation Models for PDEs." *arXiv preprint arXiv:2405.19101* (2024).
> >
> > [6] Sun, Jingmin, et al. "Towards a Foundation Model for Partial Differential Equation: Multi-Operator Learning and Extrapolation." *arXiv preprint arXiv:2404.12355* (2024).
> >
> > [7] Huang, Jiahe, et al. "DiffusionPDE: Generative PDE-solving under partial observation." *arXiv preprint arXiv:2406.17763* (2024).
> >
> > [8] Négiar, Geoffrey, Michael W. Mahoney, and Aditi S. Krishnapriyan. "Learning differentiable solvers for systems with hard constraints." *arXiv preprint arXiv:2207.08675* (2022).

---

> > > ### Author Response · Authors · 2024-11-18
> > > **Updated manuscript and additional experimental results**
> > >
> > > We have uploaded our revised manuscript and have provided additional clarifications and experimental results in our general rebuttal. We provide additional details as follows.
> > >
> > > ## Additional visualizations
> > >
> > > We provide visually more intuitive plots of **errors of the statistics** in Figure 1, 2, 10, 12, 14. In these plots, the grey regions indicate a smaller absolute error, whereas red and blue regions indicate positive and negative errors, respectively. It can be seen that gradient-based approaches like DiffusionPDE and D-Flow exhibit more noticeable artifacts around the boundary.
> > >
> > > ## Experiments with different IC/BC
> > >
> > > We further provide additional generative evaluation on the Stokes problem with various ICs/BCs below. It can be clearly demonstrated that our proposed ECI sampling can achieve **consistent performance improvement** over the baselines in a zero-shot manner with the flexibility to all different IC settings $k=5,10,15$ and BC settings $\omega=4,6,8$. ($k=5$ and $\omega=6$ are already provided in the original experiments.)
> > >
> > > | Dataset | Metric | ECI | ANP | ProbConserv | DiffusionPDE | D-Flow | CondFFM |
> > > |---|---|---|---|---|---|---|---|
> > > | Stokes IC $k=10$ | MMSE / $10^{-2}$ | **0.113** | 4.616 | 1.792 | 0.512 | 0.774 | 0.307 |
> > > |  | SMSE / $10^{-2}$ | **0.124** | 12.400 | 0.356 | 0.301 | 59.377 | 0.060 |
> > > |  | CE / $10^{-2}$ | 0 | 13.032 | 0 | 0.068 | 3.004 | 0.011 |
> > > |  | FPD | **0.156** | 6.694 | 3.619 | 0.340 | 11.013 | 0.454 |
> > > | Stokes IC $k=15$ | MMSE / $10^{-2}$ | 0.190 | 11.148 | 0.797 | 0.216 | 0.088 | 0.125 |
> > > |  | SMSE / $10^{-2}$ | **0.183** | 3.354 | 2.751 | 0.470 | 12.063 | 0.010 |
> > > |  | CE / $10^{-2}$ | 0 | 18.278 | 0 | 0.194 | 13.149 | 0.010 |
> > > |  | FPD | **0.105** | 6.509 | 6.425 | 0.832 | 4.745 | 0.020 |
> > > | Stokes BC $\omega=4$ | MMSE / $10^{-2}$ | **0.676** | 9.273 | 3.034 | 1.488 | 1.728 | 0.031 |
> > > |  | SMSE / $10^{-2}$ | **0.378** | 2.777 | 2.462 | 2.278 | 6.343 | 0.081 |
> > > |  | CE / $10^{-2}$ | 0 | 115.335 | 0 | 39.99 | 14.553 | 0.012 |
> > > |  | FPD | **1.671** | 5.309 | 1.686 | 3.954 | 10.043 | 0.071 |
> > > | Stokes BC $\omega=8$ | MMSE / $10^{-2}$ | **0.042** | 4.115 | 9.008 | 5.794 | 3.801 | 0.028 |
> > > |  | SMSE / $10^{-2}$ | **0.026** | 2.462 | 2.710 | 3.288 | 22.376 | 0.010 |
> > > |  | CE / $10^{-2}$ | 0 | 73.916 | 0 | 160.397 | 51.056 | 0.017 |
> > > |  | FPD | **0.109** | 6.663 | 0.843 | 9.356 | 13.202 | 0.111 |
> > >
> > > ## Scalability
> > >
> > > See our **general rebuttal** for additional experimental results on higher-resolution data and the analysis on time and memory consumption. With this evidence,  we believe our ECI framework enjoys better scalability than other baselines.

---

> > > ### Comment · Reviewer_KM5e · 2024-11-25
> > > **Response after rebuttal**
> > >
> > > Thank you for your rebuttal. While some of my concerns have been addressed, after carefully reviewing the other comments, I believe the paper would benefit from another revision to improve clarity in writing and include detailed explanations, along with illustrative figures where necessary. Therefore, I maintain my original score.

---

> > > > ### Author Response · Authors · 2024-11-25
> > > > **Thanks for your response**
> > > >
> > > > We sincerely thank your response and your recognition of our contributions in the generative domain after reviewing our rebuttals and additional experimental results. We are also grateful for your detailed suggestions on further improving the conciseness of clarity of our work, making it potentially more accessible to a larger audience.

---

### Official Review · Reviewer_rhkG · 2024-11-04

**Soundness:** 1
**Presentation:** 3
**Contribution:** 1
**Rating:** 5
**Confidence:** 4

**Summary:**

This paper tackles the problem of conditional generation of physical systems with physical constraints (such as physical conservation laws). They propose “a framework for adapting pretrained zero-shot, unconstrainted generative models to satisfy exacts constraints” in a computationally efficient manner. Subsequently, the apply the framework to Partial Differential Equation (PDE) systems and demonstrate empirical results on a variety of 1D and 2D physical problems.

**Strengths:**

* The method does seem to operate as advertised. The generated results have 0 constraint error, compared to > 0 constraint error of other soft-constrained methods. At the very least, the claim that ECI guarantees hard-constraint satisfaction is well supported
* The ablation study that shows the effect of the resampling intervals and the mixing iterations is quite good.
* Reporting both generation quality as well as runtime for various experiments helps objectively evaluate the proposed methodology.
* Overall, the presentation of the paper if quite good. The problem statement and context are very clearly outlined at the start of the paper. The advantages of the methodology are plain described, and the research problem is clearly state and motivated within existing literature. Thorough and well written description of experiments and results. Nice figures though could benefit from more detailed captions describing what we are looking at.

**Weaknesses:**

__Theoretical Concerns__
* While the authors do a great job of citing the existing context for SciML applications and flow-matching models, they do not cite the large existing bodies of work related to constraint satisfaction/constrained optimization. In particular, many methods share many similarities with ECI in that they perform unconditional update steps, and then project back into the feasibility region. As such, the method lacks theoretical motivation and context of existing works.

__Experimental Concerns__
* Many of the experiments performed are quite simple, with either low spatial resolution, low dimensionality, or a small number of time-steps. In order to validate that the correction step is not overly destructive, it would be beneficial to perform experiments with higher resolution, and a greater number of time steps. Additionally, there is a concern that ECI performs poorly in complex settings, as it is out-performed by one model or another on all 2D experiments (in terms of MMSE), and thus it would be good to see a 3D fluid setting to evaluate how ECI performs in higher-dimension problems.
* Additionally, fluid flow can exhibit quite complex behaviour. It is not clear why an increased variance in the solutions is a sign of a bad solution, as opposed to the system exhibiting complex dynamics (such as turbulence). Thus, the justification for SMSE as an evaluation metric is not quite clear.

__Contribution Concerns__
* First, the concept of perform Extrapolate-Correct-Interpolate steps in order to solve constraint satisfaction is not a novel idea. In fact, and is well studied in constraint satisfaction settings. For example, projected gradient descent functions very similarly by first taking a step to minimize the objective (Extrapolate) and then taking a step to project back into the feasible region (Correct).
* Second, as mentioned in the paper, the extrapolate step assumes a pre-trained generative model (and thus is not a novel contribution), the interpolate step simply adds noise to the sample (and thus no novel contribution contribution), so the only real contribution is the correction step. However, the correction step is extremely trivial. In the case of the boundary conditions, the method simply sets the boundary values to the desired constraint. In the case of conservation, the method subtracts the deviation from the conserved amount from all cells equally. Both approaches are extremely simple, and don’t really capture realistic physics. We can construct pathological examples where this type of correction leads to systems which satisfy the constraints but are completely implausible and useless for any sort of SciML application. In fact, this correction step is more or less the most trivial way to project back into the feasibility region.
* Furthermore, to address new problems, you need to implement your own correction steps, which may be difficult, computationally intensive or simply intractable for more interesting problems. For example, there is no treatment for conservation of a vector quantity (i.e. momentum, angular momentum), which would occur in many settings of interest.
* Additionally, the paper deals with a pretty niche problem. Notably hard-constrained generation using flow-matching models for physical system simulation. However, when digging into the ECI framework, we note that the E and I steps come from pre-existing generative models, and the C step is incredibly simple. Thus, the overall contribution is very limited in scope of application, as being quite simple from a theoretical and practical point of view.
* Overall, given a pre-trained flow matching model, ECI is a somewhat simple extension that allows for hard constraints to be satisfied. This is guaranteed by the correction step, which is quite simplistic. In particular, we would expect that such coase correction steps would lead to poor solution quality. In fact, from the experimental results, we can see that ECI performs well in 1D settings, but under performs compared to other models in the more complex 2D problems. In fact, we would expect it to perform even worse with high spatio-temporal resolution or in 3-dimensional problems.


Overall, the paper is very well written and presented, but the ECI framework (in particular the correction step) do not present a significant theoretical or practical contribution.

**Questions:**

* Is it possible to address the conservation laws in a more principled way? For example, minimal local perturbations to achieve the desired conservation (rather than a global correction), or monitoring which cells are causing the violation and thus correct the violations in a more targeted manner?
* Can we apply ECI to a 3D problem/higher resolution/more complex problem to see if the correction step is much worse? It would also be good to see an experiment at a much higher resolution and with many more time-steps.
* Why are we using Frechet Poseidon Distance? For these problems, we know the physics and so would it not be better to simply compute the ground truth for evaluation?

---

> ### Author Response · Authors · 2024-11-12
> **Rebuttal to rhkG (Part I)**
>
> Thank you for your appreciation of our work's **clear presentation**, **sufficient ablation studies**, and **effectiveness regarding hard constraints**. We will address your concerns as follows.
>
> # Significant misunderstanding and misinterpretation of our work
>
> We would first like to point out a **significant misunderstanding and misinterpretation** of our work. We address the constrained generation of the flow matching framework. The guided flow matching model during the iterative sampling process is still a **generative model** whose output is a **distribution over solutions** instead of a single ground truth solution. Sadly, we noted the reviewer emphasized the projected gradient approaches which are **infeasible for flow-based generative models**. The reviewer also **misinterpreted our generative evaluation metrics** and **emphasized constrained models in regression models** like projected gradient, which was not our claimed contribution.
>
> We further distinguish our **problem and experimental setup** from existing work, especially for the projected gradient-related methods, in the following aspects.
>
> - **Generative modeling**. Unlike projected gradient-related methods, our proposed framework is a generative framework that captures the distribution of solutions instead of a single solution.
> - **Challenge**. The challenge for guiding flow (or diffusion) based models lies in its iterative sampling process, as we have mentioned in the second part of related work and Table 1. The iterative nature makes it infeasible to employ normal projected gradient approaches.
> - **Gradient-free** guidance. Our proposed ECI-sampling is completely gradient-free. We demonstrated in our experimental results that our gradient-free approach still outperformed gradient-based methods while achieving significant speed-up.
> - **Evaluation** metrics. The generative nature requires different evaluation metrics that capture distributional properties instead of simple MSE to the ground truth. In this regard, we calculate the point-wise MSE of the mean and standard deviation between the ground truth distribution and the generation distribution. We further apply Poseidon, a PDE foundation model to evaluate the Frechet distance.
>
> We further reiterate our **contributions**, which we believe may have also been misinterpreted by the reviewer, as follows:
>
> - We proposed ECI sampling for hard-constrained generation for pre-trained **flow-based generative models**. We emphasize *flow-based generative models*, which impose unique challenges due to their iterative sampling nature. Existing approaches for *constrained prediction* in regression models are infeasible or perform badly. To the best of our knowledge, we are the first to deal with the zero-shot flow-based constrained generation problems for functional data.
> - **The extrapolation and interpolation concepts are unique to flow-based models**, which are completely different from any non-iterative approaches like project gradient. Indeed, we do not claim contributions in the correction — we even manually placed it in the Appendix as it is less important. Instead, the **iterative control framework** unique to flow-based models using the interleaved ECI steps is our major claim for a better generation result.
> - Our approach guarantees the exact satisfaction of constraints **without the expensive gradient information**.
> - Our approach achieves **state-of-the-art generative performance** on most evaluation metrics. We emphasize *generative evaluation metrics*, as the reviewer has drastically misinterpreted our metrics as regression MSEs. Instead, distributional properties are calculated and the foundation model Poseidon is used to give the most comprehensive metric.
>
> We understand the reviewer might **not be familiar with generative modeling** in the PDE domain, which, as we have mentioned in the first part of our related work, is a newly emerged realm for functional data. However, as we have already pointed out the reviewer's misunderstanding and have distinguished our problem setup and contributions from existing work, we urge the reviewer to give a more fair and comprehensive evaluation of our work.

---

> > ### Author Response · Authors · 2024-11-12
> > **Rebuttal to rhkG (Part II)**
> >
> > # Theoretical Concerns
> >
> > ## Q1 Relation to projected gradient
> >
> > We have already included a large amount of related work in the original manuscript. In the second part of related work, we reviewed existing work in the PDE domain for zero-shot constrained generation for **diffusion models**. In the third part, we also included related work in the computer vision domains which can be adapted for PDE tasks. We also include constrained prediction (*NOT generation*) work (e.g., [1-4]), which is essentially a special case of the projected gradient. However, we emphasize that we focus on **flow-based generative models**, so these regression approaches are **fundamentally different** from our problem step and can only serve as the correction stage of our framework.
> >
> > We respectfully disagree with the reviewer's claim that "*many methods share many similarities with ECI in that they perform unconditional update steps, and then project back into the feasibility region*" and the lack of novelty. Though constrained prediction for regression models does exist (e.g., [1-4], as we have already cited them), they failed in the context of guiding diffusion or flow-based models that rely on iterative sampling. To the best of our knowledge, **we do not know any existing work for diffusion or flow-based models that share a similar idea with ours**. We would appreciate it if the reviewer could give more concrete references.
> >
> >
> >
> > # Experimental Concerns
> >
> > ## Q2 Experimental Setup
> >
> > We understand that the reviewer may be also not familiar with PDE domains. We noted all datasets and their spatiotemporal resolution setups are directly **taken from previous papers** (see Appendix B for reference of the datasets in previous papers). Therefore, we believe such an experimental setup is **standard, consistent, and comparable with existing works**. The datasets of PDE systems used in our work have also been the standard test cases in various papers (e.g. [5, 6]), which in turn demonstrated they are not "simple" cases, as the reviewer tried to suggest.
> >
> > We further point out that the Navier-Stokes equation is a 3D dataset with 2D spatial and 1D temporal resolutions. To the best of our knowledge, **4D datasets in PDE systems do not exist so far**. It is also a **standard practice** to test on 2D and 3D datasets in the PDE domain, e.g., neural operator learning [5, 6], foundation models [7, 8], and existing constrained generation [2, 9]. Therefore, we believe we have already provided enough evidence of the superior performance of our proposed method.
> >
> > We also believe the reviewer's claim that "*ECI performs poorly in complex settings*" is **ungrounded** due to their misinterpretation of our evaluation metrics (also see Q3). The MSE of the point-wise mean between the ground truth and the generation only partially reflects the resemblance of distributions. The MSE of the point-wise standard deviation is another metric, in which our model **significantly outperforms existing approaches**. Furthermore, inspired by the *Frechet inception distance* (FID), a widely used metric in image generation domains, we proposed to use the foundation model Poseidon to give a more comprehensive evaluation of the distributional property, in which our model excels over baselines. See Appendix D.1 for details.
> >
> > ## Q3 Variance in the solution
> >
> > We have noted this is one of the **misinterpretations** regarding our evaluation metrics. We do not claim a higher variance is a sign of bad solutions, as the reviewer suggested. Instead, SMSE **measures the difference in the point-wise standard deviation** of the generation with respect to the ground truth solution set. Therefore, a smaller SMSE indicates **a closer resemblance of variance** to the ground truth solution distribution, thus a better generation result.

---

> > > ### Author Response · Authors · 2024-11-12
> > > **Rebuttal to rhkG (Part III)**
> > >
> > > # Contribution Concerns
> > >
> > > ## Q4 Concept of ECI
> > >
> > > We respectfully disagree with the reviewer's claim that ECI has been "*well studied*". We have noted in the previous clarification of the reviewer's misunderstanding that **extrapolation and interpolation have unique meanings in flow-based models** and are **completely different from concepts in regression-based methods** like the projected gradient.
> > >
> > > In flow matching, extrapolation and interpolation are geometric concepts related to the push-forward measures $\mu\_t=(\psi\_t)\_* \mu\_0$ on the probability path $\\{\mu_t,0\le t\le 1\\}$ [10] . Here $\mu_0$ is the initial (fixed) noise measure, $\psi_t$ is the flow (possibly defined with the learned vector field), and $(\psi_t)_*$ denotes the push-forward. When conditional on the target solution $u_1$​, the extrapolation along the conditional probability path can be written in a variational way in Equation 2. Similarly, interpolation are defined along the probability path. In this way, extrapolation is never a step to "minimize the objective", and interpolation has no correspondence with the projected gradient. Except for the correction stage which we do not claim as our contribution (see Q5), **we do not see any similarity between our ECI and the projected gradient, nor do we know any existing flow-based constrained generation model that uses a similar approach**.
> > >
> > > Furthermore, with an iterative sampling process, the intermediate generation is partially noises (see Figure 4), making the projected gradient infeasible, as it operates on the final prediction in regression models. This further distinguishes our framework from existing works.
> > >
> > > ## Q5 Novelty of ECI
> > >
> > > The reviewer has a **significant misunderstanding of our contributions**. We do NOT claim any contribution in the correction stage  — we even manually placed it in the Appendix as it is less important. Instead, the **iterative control framework** with the concepts of extrapolation and interpolation unique to flow-based models (see Q4) using the interleaved ECI steps is our major claim for a better generation result. Furthermore, we almost **deliberately designed the correction step to be as simple as possible** to solidly demonstrate that the **good performance of the final generation comes from our iterative control framework** instead of from the correction algorithms like the projected gradient.
> > >
> > > Indeed, **even with such a naive correction algorithm, our ECI sampling still achieves remarkable generation results with hardly noticeable artifacts around the constrained region**. This is in sharp contrast with the ProbConserv baseline — a projection-based approach that directly operates on the final prediction without fine-grained iterative control — which exhibits noticeable and sharp change around the constrained region (see Figure 5 for a close-up result). Furthermore, in the correction stage in Figure 4, we demonstrate how the artifacts around the constrained boundary are gradually mitigated with our iterative ECI framework.
> > >
> > > ## Q6 Correction Step
> > >
> > > We have already clarified in Q4 & Q5 that **the correction stage is never our contribution**. We also explain in Q5 why we deliberately designed the correction step as simple as possible. Regardless, ECI sampling still achieves state-of-the-art generation performance thanks to its **iterative control** over the flow sampling. New correction algorithms are always readily applicable to our framework but are not necessary. For experiments in our paper, we already demonstrated simple correction algorithms suffice.
> > >
> > > ## Q7 Hard-Constraint Generation
> > >
> > > The hard-constraint generation task has **crucial importance** in physical systems like PDE, supply chains, and time series, where certain constraints like conservation laws have clear physical meanings and should always be respected. Previously, many regression models already tried to address this problem [1-4]. Our ECI sampling further extends such settings to generative modeling. Therefore, we respectfully disagree with the "*limited scope*" suggested by the reviewer, as we have already demonstrated the **effectiveness of ECI sampling across various 2D and 3D PDE datasets** and the potentially **wide applications in diverse SciML domains**.
> > >
> > > We reiterate that extrapolation and interpolation have different and unique meanings in diffusion and flow-based generative models and that our contributions lie in the **iterative control of the flow sampling process by interleaving ECI steps**. This is completely different from any existing works, especially regression-based approaches like the projected gradient. In fact, though the extrapolation and interpolation for flow models (or the forward and backward diffusion processes for diffusion models) have been well-defined, **guiding diffusion or flow-based models is non-trivial**, and many existing works in the computer vision domain all adopted a similar zero-shot formulation [11, 12].

---

> > > > ### Author Response · Authors · 2024-11-12
> > > > **Rebuttal to rhkG (Part IV)**
> > > >
> > > > ## Q8 Generation Performance
> > > >
> > > > We have noted the **misunderstanding of our contributions** of ECI sampling, especially in terms of the correction stage in Q5 & Q6, and we have also noted the reviewer's **concerns for poor performance were ungrounded** in Q2. Indeed, across all 2D and 3D generative datasets, ECI sampling achieves state-of-the-art performance on most evaluation metrics. It is unclear to us **why the reviewer emphasized solely the MMSE for the NS equation but deliberately ignored all of the other metrics and datasets** in which our ECI-sampling achieved the best results.
> > > >
> > > >
> > > >
> > > > # Other Questions
> > > >
> > > > ## Q9 Conservation Laws
> > > >
> > > > It is always possible to apply alternative correction algorithms for conversation laws, but **unnecessary**, as 1) this correction stage is not our contribution; and 2) even the simple projection algorithm works pretty well with our iterative framework.
> > > >
> > > > ## Q10 Higher Dimensional Data
> > > >
> > > > We reiterate that the correction stage is the least important stage in our algorithm, as even a simple projection algorithm can lead to decent results. Therefore, we believe our ECI sampling can be scaled to higher dimensional data while preserving its competitive performance. However, as we have mentioned in Q2, such data are, if not none, scarce for training a generative model. We believe our existing experiments across various 2D and 3D PDE systems already demonstrated the superior performance of ECI sampling.
> > > >
> > > > ## Q11 Frechet Poseidon Distance
> > > >
> > > > As we have mentioned at the very beginning of our rebuttal, our ECI sampling is still a **generative framework** for which generative evaluation metrics for distributional properties should be used instead of the normal regression metrics. In this sense, **the ground truth is a set of solutions** with no one-to-one correspondence, making it impossible to calculate the regression metrics.
> > > >
> > > > In contrast, inspired by the *Frechet inception distance* (FID), a widely used metric in image generation domains, we proposed to use the foundation model Poseidon to give a **more comprehensive evaluation of the distributional property**, in which our model excels over baselines. See Appendix D.1 for details.
> > > >
> > > > [1] Saad, Nadim, et al. "Guiding continuous operator learning through physics-based boundary constraints." *arXiv preprint arXiv:2212.07477* (2022).
> > > >
> > > > [2] Négiar, Geoffrey, Michael W. Mahoney, and Aditi S. Krishnapriyan. "Learning differentiable solvers for systems with hard constraints." *arXiv preprint arXiv:2207.08675* (2022).
> > > >
> > > > [3] Li, Zongyi, et al. "Anima Anandkumar, Physics-informed neural operator for learning partial differential equations." *arXiv preprint arXiv:2111.03794* (2021).
> > > >
> > > > [4] Mouli, S. Chandra, et al. "Using Uncertainty Quantification to Characterize and Improve Out-of-Domain Learning for PDEs." *arXiv preprint arXiv:2403.10642* (2024).
> > > >
> > > > [5] Li, Zongyi, et al. "Fourier neural operator for parametric partial differential equations." *arXiv preprint arXiv:2010.08895* (2020).
> > > >
> > > > [6] Li, Zongyi, et al. "Anima Anandkumar, Physics-informed neural operator for learning partial differential equations." *arXiv preprint arXiv:2111.03794* (2021).
> > > >
> > > > [7] Herde, Maximilian, et al. "Poseidon: Efficient Foundation Models for PDEs." *arXiv preprint arXiv:2405.19101* (2024).
> > > >
> > > > [8] Sun, Jingmin, et al. "Towards a Foundation Model for Partial Differential Equation: Multi-Operator Learning and Extrapolation." *arXiv preprint arXiv:2404.12355* (2024).
> > > >
> > > > [9] Huang, Jiahe, et al. "DiffusionPDE: Generative PDE-solving under partial observation." *arXiv preprint arXiv:2406.17763* (2024).
> > > >
> > > > [10] Kerrigan, Gavin, Giosue Migliorini, and Padhraic Smyth. "Functional flow matching." *arXiv preprint arXiv:2305.17209* (2023).
> > > >
> > > > [11] Lugmayr, Andreas, et al. "Repaint: Inpainting using denoising diffusion probabilistic models." *Proceedings of the IEEE/CVF conference on computer vision and pattern recognition*. 2022.
> > > >
> > > > [12] Kawar, Bahjat, et al. "Denoising diffusion restoration models." *Advances in Neural Information Processing Systems* 35 (2022): 23593-23606.

---

> > > > > ### Comment · Reviewer_rhkG · 2024-11-16
> > > > > **Addressing Q8 - Q11**
> > > > >
> > > > > # Q8
> > > > >
> > > > > As previously mentioned, the description of the SMSE in the paper is lacking, and the comment providing additional details has helped address my concerns with the results to a certain degree. However, as previously mentioned in my comments, the lack of error bars detracts from the results as it's not clear if the differences are significant. This, in addition to the fact the ECI performs much worse on MMSE (which I believe is the most important metric) call into question whether ECI is an improvement over existing methods, or if there are deeper trade-offs that make the novelty less clear.
> > > > >
> > > > > # Q9 & Q10
> > > > >
> > > > > Addressed in other comments, no further discussion from my end.
> > > > >
> > > > > # Q11
> > > > >
> > > > > While Frechet Poseidon Distance (FPD) is justified in the context of existing metrics such as Frechet Inception Distance (FID) for images, images are a very different domain from PDEs. Notably, there isn't a clear closed-form ground truth model for images, while physical laws are known and thus we can evaluate PDE solutions with a known, closed-form "perfect" oracle. As such, my emphasis on MMSE over other metrics (in particular the FPD) is that the dynamics of such PDE systems are known, and can be evaluated with arbitrary precision. And thus it does not make sense to emphasis foundation models as more important to than the true underlying distribution. In fact, one could even perform monte-carlo simulations of the physical system with a typical PDE solver to obtain an estimate of the true distribution of solutions. In my view, when dealing with physical systems, we should be emphasize alignment with reality, rather than a foundation model with will very likely differ from the underlying physical systems.

---

> > > > > > ### Author Response · Authors · 2024-11-18
> > > > > > **Response to rhkG (Part I)**
> > > > > >
> > > > > > We sincerely thank you for your timely and detailed response regarding our rebuttal, and we are more than happy to continue such a constructive dialogue with you.
> > > > > >
> > > > > > We are glad that our previous rebuttal can successfully clarify some potential misunderstandings and address some of your concerns. We have uploaded the revised manuscript with additional clarifications, more concise formulation, additional experiments, and highlights of the contributions of our proposed ECI sampling in the generative domain. Please refer to our updated general rebuttal for details. With such updated information, we further address your remaining concerns as follows.
> > > > > >
> > > > > > ## Q1 Contribution and Novelty
> > > > > >
> > > > > > We fully agree that novelty and contributions are one of our top priorities and what we would like to convey in our paper. We appreciate that the reviewer has agreed to view our work as a **cohesive framewor**k to offer iterative control over the flow matching sampling process instead of simple stacking of stages.
> > > > > >
> > > > > > Although we do agree our method shares some conceptual similarities with some existing works for diffusion models, — as we have included a wide range of related works in Section 2.3, — our **problem setup and application scenario are completely different** from previous work, which we further elaborate as follows:
> > > > > >
> > > > > > - We focus on **flow matching models** instead of diffusion models. The deterministic nature of flow ODE makes some existing variational approaches [1, 2] for diffusion models inapplicable.
> > > > > > - We focus on the **hard constraints in functional data** with **unique challenges** different from computer vision domains. We have a detailed discussion on such challenges in Section 1 and why common gradient-based methods may fail in such a scenario.
> > > > > > - We propose a **gradient-free approach** to ensure hard constraints. This is fundamentally different from inverse problems where constraints are suggestions instead of requirements, and multiple permissible solutions can be obtained.
> > > > > >
> > > > > > Based on the above difference, we respectfully **disagree that our proposed ECI sampling is a simple adaptation** of existing approaches. Similarly, we also **disagree that the simplicity of our approach indicates insufficient contributions or novelty**. Our framework is specifically tailored for the unique challenges and is therefore well-motivated for our problem setup. The ECI sampling, as a cohesive framework for guiding flow matching models, adapts a novel gradient-free approach that differentiates our framework from all existing works. As concrete evidence, our contributions in generative modeling are **clearly recognized by the two new reviewers who are domain experts**.
> > > > > >
> > > > > > We noted that, though the reviewer kept referring to our framework as "*applying a well-known pattern*", they did not provide any reference. In fact, to the best of our knowledge, we do not know of any previous work adopting a similar approach for flow matching models. We, therefore, **request the reviewer to provide concrete references** in this regard to support their claim instead of **unfairly overshadowing our contributions** in the generative domains.

---

> > > > > > > ### Author Response · Authors · 2024-11-18
> > > > > > > **Response to rhkG (Part II)**
> > > > > > >
> > > > > > > ## Q2 Experimental Results
> > > > > > >
> > > > > > > Regarding the evaluation metrics, we first point out **a clear factual mistake** in the reviewer's response. The MSE for statistics like mean and standard deviation, referred to as MMSE and SMSE respectively in our work, **is clearly used in the FFM paper** as their evaluation metrics for generative quality. Furthermore, the DDO paper [3], cited in our original manuscript, **uses Frechet Inception distance (FID)** as the evaluation metrics on the MNIST-SDF dataset. The adaptation of the FID score for PDE solutions is exactly our defined FPD, as we have clearly stated their connection in Appendix D.1. Therefore, we have to assume that the reviewer did not carefully look into any related generative literature as their concerns regarding the validity of the evaluation metrics used in our generative framework are **completely ungrounded**. We highly recommend the reviewers look into related generative literature, as such metrics over the distributional properties are the **standard approaches** in generative modeling. Furthermore, if the reviewer has any alternative evaluation metric with solid references, — we have clearly explained in our previous rebuttal why regression metrics are not applicable in our generative settings, — we will be more than happy to provide additional results.
> > > > > > >
> > > > > > > Regarding the error bar, we follow the standard practice in previous constrained generation papers including [1-3] and FFM which also do not have error bars. To minimize the impact of intrinsic stochasticity, we generated **a large number of solutions** (512) for each model on each dataset to calculate the statistics (see Appendix D.3). In this way, the provided evaluation metrics are already amortized metrics that accurately depict the generation quality.
> > > > > > >
> > > > > > > We reiterate that MMSE is only one of our generation metrics. Specifically, MMSE only reflects the first-order behavior of the generation. Therefore, as we have also mentioned in the previous rebuttal, we find the reviewer's claim of "*underperforming*" **completely unsupported**, as our ECI sampling achieves the **best scores for 16 out of 20 metrics**. Furthermore, on the additional experiments for the scalability of ECI sampling in our general rebuttal, ECI sampling achieves the **best scores for 23 out of 24 metrics** in Table 10 and 11 in the revised manuscript. It remains unclear why the reviewer deliberately focused only on the few metrics where ECI is worse as if **overshadowing our overall state-of-the-art performance**.
> > > > > > >
> > > > > > >
> > > > > > >
> > > > > > > ## Q3 Additional Response
> > > > > > >
> > > > > > > #### Q1, 3, 4, 8
> > > > > > >
> > > > > > > We have clearly stated in the original manuscript that "*we calculate the pointwise mean and standard deviation (std) of the ground truth solutions and the solutions generated from various sampling methods.*" From the generative modeling perspective, such a distributional approach is also **standard** in previous work like FFM.
> > > > > > >
> > > > > > > #### Q11
> > > > > > >
> > > > > > > > There isn't a clear closed-form ground truth model for images
> > > > > > >
> > > > > > >  **This is not true**. The ground truth images (e.g., class-conditioned digits in MNIST) themselves are used to calculate ground truth statistics, as was done in the DDO paper.
> > > > > > >
> > > > > > > > While physical laws are known and thus we can evaluate PDE solutions with a known, closed-form "perfect" oracle
> > > > > > >
> > > > > > > While this is true, we point out one significant misunderstanding of the reviewer of our evaluation method. We never claim that ground truth solutions are not available. Instead, we also calculate these ground truth solutions based on the PDE system. Our emphasis is that similar to the FID score for image generation, a foundation model can be used to **extract meaningful high-level information** to project the solutions to latent space. As a concrete example, the hypothetical generation $\mathcal{U}\_\text{gen}=\\{\sin(x)-100,\sin(x)+100\\}$ perfectly recovers the data $\mathcal{U}\_\text{gt}=\\{\sin(x)\\}$ in terms of MMSE, but clearly they are divergent. MMSE only captures the first-order information, SMSE the second, but **FPD captures high-level information**. This is essentially the problem of why we use FID for image generation instead of the errors in the pairwise mean.

---

> > > > > > > > ### Author Response · Authors · 2024-11-18
> > > > > > > > **Response to rhkG (Part III)**
> > > > > > > >
> > > > > > > > We sadly discovered that the **reviewer's unfamiliarity with the generative domains** has required us to provide clarifications on some **standard practices and basic concepts** in generative modeling, including but not limited to, the use of FID-like scores in previous work for generative modeling, our disparate problem setup for the flow matching models, and even some factual mistakes regarding evaluation metrics. In contrast, our contributions to generative modeling have been **comprehensively evaluated and unanimously recognized** by the two emergency reviewers from the generative domain. Given the current scores and the reviewer's **lack of concrete references** to support their concerns on both our contributions and experimental results, we have to, though very reluctantly, assume that the reviewer is deliberately **overshadowing our contributions**, as we have already provided concrete evidence and explanations above why we believe these concerns are completely ungrounded. We highly recommend the reviewer look into generative literature. Still, we always maintain an open and collaborative attitude to all reviews, and concrete suggestions on the limitations of work have been properly addressed in our revised manuscript.
> > > > > > > >
> > > > > > > >
> > > > > > > >
> > > > > > > > [1] Lugmayr, Andreas, et al. "Repaint: Inpainting using denoising diffusion probabilistic models." *Proceedings of the IEEE/CVF conference on computer vision and pattern recognition*. 2022.
> > > > > > > >
> > > > > > > > [2] Kawar, Bahjat, et al. "Denoising diffusion restoration models." *Advances in Neural Information Processing Systems* 35 (2022): 23593-23606.
> > > > > > > >
> > > > > > > > [3] Lim, Jae Hyun, et al. "Score-based diffusion models in function space." *arXiv preprint arXiv:2302.07400* (2023).

---

> > > > ### Comment · Reviewer_rhkG · 2024-11-16
> > > > **Q4-Q7**
> > > >
> > > > # Q4 - Q7
> > > >
> > > > After reviewing the previous comments, some of my concerns with the results have been addressed. However, I am still concerned that ECI performs much worse than other methods in terms of MMSE across all 3D datasets and thus do not necessarily that ECI "achieves remarkable generation results" or that it significantly outperforms other methods. But rather there seems to be a fundamental trade-off in generation quality (as quantified by the MMSE).
> > > >
> > > > While I still have reservations regarding the theoretical and conceptual novelty, I will not belabor the point and will give the opportunity for the authors to respond.

---

> > > ### Comment · Reviewer_rhkG · 2024-11-16
> > > **Addressing Q1-Q3**
> > >
> > > # Q1
> > >
> > > As outlined in my previous comment, the ECI framework is quite conceptually simple, consists of pre-existing operations and steps that are not novel contributions, and lacks a principled theoretical justification and/or analysis.
> > >
> > > # Q2
> > >
> > > While I am quite familiar with PDEs in the context of engineering and physical sciences, I admit that I have not done much work on generative models for PDEs. Thank you for clarifying that these datasets are standard practice and are in-line with existing works.
> > >
> > > However, it is more or less trivial to construct 4D datasets, as many (or all) of the PDEs have 3D + Time formulations (such as navier-stokes). That being said, constructing a novel 4D PDE dataset could very well be outside the scope of this work. This concern is addressed, thank you.
> > >
> > > # Q3
> > >
> > > Thank you for the clarification. I was not aware that it was the SMSE measures the difference between the variance of the generated solutions and the ground truth variance. However, this is a failure of the paper to properly describe the error metrics used. In fact, the only descriptions of SMSE are given on pages 6 ("standard deviation (SMSE)") and 19 (" mean squared errors of mean and standard deviation (MMSE and SMSE)"). As such, while this addresses my concerns about the SMSE used as a metric, it does raise the issue that the paper lends itself to misunderstanding the methodology through inadequate description of the error metrics.

---

> > ### Comment · Reviewer_rhkG · 2024-11-16
> > **Clarifying Points**
> >
> > Thank you for your detailed rebuttal, I will address each comment's content individually through comments.
> >
> > As a first comment, I want to clarify my reservations and questions about this work. The concern is not so much that the results are incorrect, but rather that there is not significant enough novelty. When evaluating a paper, I look at theoretical novelty (novel ideas, novel analysis, and novel theoretical results such as bounds) and empirical novelty (such as improved performance on existing baselines, or extension to previously unsolved tasks).
> >
> > As I understand (and as the authors have highlighted both in the paper and comments), the ECI framework allows for zero-shot gradient-free conditional generation using flow matching models. There is a clear challenge that they are aiming to solve, and the proposed methodology is sufficiently well defined.
> >
> > However, when we look at the components of ECI (namely the extrapolation, correction, and interpolation steps), my understanding is that none of these steps are novel. The extrapolation and interpolation steps are pre-existing operations coming from flow matching models. Additionally, in a following comment it is stated that "the correction stage is never our contribution". So we can conclude that none of the individual steps are novel contributions of this work. Thus, the remaining contribution must be the combination of these individual steps into a cohesive framework.
> >
> > When looking at the ECI framework, some parallels with existing methods are quite clear. In my original comment I mention methods such as constrained optimization and projected gradient descent. While the authors point out that projected gradient descent is a gradient-based method that requires a regressor, ECI is zero-shot, gradient free and does not require a regressor. However, there are some deeper conceptual similarities. The core concept for ECI is that at each step of the flow matching model, they perform an extrapolation step, then a correction step, then an interpolation step. Again, while the mechanism through which other methods (such as projected gradient descent) perform these steps are different, the intuition is not novel. Put another way, taking a single unconditional update step, correcting the resulting point, and then performing subsequent iterations is an approach that has been applied in many settings and many models types. Thus, I do not find that the framework itself to be significantly novel, other than applying a well-known pattern to a new setting.
> >
> > Finally, we can talk about theoretical and empirical results. In terms of theoretical results, there is no analysis performed for bounds or guarantees on ECI, and there is a lack of a principled justification for ECI from a theoretical standpoint. Consequently, I do not believe that there is significant theoretical novelty or results. In terms of empirical results, the authors provide various metrics such as Mean Squared Error of the mean (MMSE), Constraint Error (CE), and standard deviation (SMSE). On the one hand, ECI performs quite well in terms of constraint error (though this is accomplished through a somewhat rudimentary adjustment), and outperforms all methods on metrics such as Frechet Poseidon Distance (FPD) and SMSE. However, when reviewing related works such as Conditional FFM, ANP, and others from Table 1, I note that many of the metrics in this paper do not occur in any of the related works. In fact, most of the related works report performance mainly as a function of the MMSE. Thus, I am not quite convinced of the validity of the SMSE and FPD metrics when declaring significant improvement over other methods. Additionally, no error bars are reported for any of the metrics, and so it is difficult to ascertain whether an improvement of 1.0 to 0.6 is significant (SMSE for Darcy Flow experiment). In light of this, it may be the case that ECI presents a significant empirical result over existing methods, but I find the lack of error bars, underperforming in terms of MMSE, and lack of principled justification for the other metrics to be open questions that would need to be addressed.
> >
> > Overall, I would like to commend the authors for their well-written and presented paper, but I feel the need to highlight several aspects and weaknesses that call into question the novelty of the work.

---

> ### Comment · Reviewer_rhkG · 2024-12-03
> **Reviewed score**
>
> I would like to thank the authors for their detailed rebuttal. While I still believe that the theoretical component of the paper is somewhat lacking, the method does present some practical benefit both in terms of hard constraint satisfaction and some performance improvements (as described in the follow-up comments/results).
>
> Overall, I think the paper is well written and presents some empirical evidence, but would benefit from more theoretical analysis and motivation. I have revised my score.

---

> > ### Author Response · Authors · 2024-12-03
> > **Thank you for your revision of your scores**
> >
> > We are grateful that our clarifications have successfully addressed your concerns regarding the contributions of our work. Your review and response regarding the clarity of our delivery and other reviewers' suggestions have been instrumental in improving our work's comprehensiveness and soundness for a larger audience. These modifications have been made in the revised manuscript, and we are committed to further polishing our work.
> >
> > We fully agree that, even though our framework demonstrated empirically superior performance over the baselines, theoretical analyses are still preferred, especially regarding the hyperparameters controlling the sampling process. Such a limitation was also noted in our manuscript, and we will continue exploring theoretical aspects in our future work.

---

### Official Review · Reviewer_bR4B · 2024-11-14

**Soundness:** 3
**Presentation:** 3
**Contribution:** 2
**Rating:** 6
**Confidence:** 3

**Summary:**

The paper introduces ECI sampling, a framework that adapts pre-trained, unconstrained generative models to exactly satisfy hard constraints in a zero-shot manner without requiring gradient computations or fine-tuning. This framework alternates between extrapolation, correction, and interpolation stages during each iterative sampling step to integrate constraint information while preserving the validity of generated outputs. Empirical results demonstrate that ECI sampling strictly adheres to physical constraints across various PDE systems and outperforms baseline approaches in zero-shot constrained generative and regression tasks.

**Strengths:**

1. The paper is easy to follow with high readability. The problem setting is well motivated and important.
2. The paper provides quite comphrenseive numerical results with comparison to relevant benchmarks. The extension to regression setting is also quite impressive.
3. The paper also provides a quite detailed ablation study on the choice of algorithm hyperparameters.

**Weaknesses:**

*1. Clarification of Problem Setup*

The problem setup is not sufficiently explained, which may lead to confusion. Although the authors repeatedly emphasize that ECI is intended for the generative modeling of constrained PDE solutions, it remains unclear what distribution of constrained solutions ECI aims to recover. Specifically, is it targeting a uniform distribution over the space of constrained solutions $\mathcal{U}_{|\mathcal{G}}$? The authors should invest more effort in formally clarifying the problem setup.

*2. Discussion of ECI’s Advantages and Comparisons*

The paper appears to lack a section that discusses the benefits and potential sources of ECI’s superior performance, as well as comparisons with other existing approaches mentioned in the work. Including such a discussion would help readers better understand both the relevant literature and the proposed method.

*3.Validation of PDE System Satisfaction*

While ECI ensures the exact satisfaction of the constrained operator by applying a correction operator after each one-step extrapolation, it is unclear why the solution obtained at time 0 still satisfies the PDE system $\mathcal{F}_{\phi} u(x) = 0$. In Algorithm 2, the next-step solution is generated by forward noising starting from the predicted solution at time 1. Does this noising process preserve the satisfaction of the PDE system? Additionally, is the linear interpolation of two PDE solutions a valid PDE solution, implying that the PDE system is linear? Further explanations are needed to clarify these points.

*4.Missing References on Gradient-Free Guidance*

Some references on gradient-free guidance are missing, such as [1] and [2]. It appears that the proposed method is related to [2], involving functional FM and additional projection steps. Providing more clarification on these connections would be appreciated.

[1]Yujia Huang, Adishree Ghatare, Yuanzhe Liu, Ziniu Hu, Qinsheng Zhang, Chandramouli S Sastry, Siddharth Gururani, Sageev Oore, and Yisong Yue. Symbolic music generation with nondifferentiable rule guided diffusion. arXiv preprint arXiv:2402.14285, 2024.

[2]Chung, Hyungjin, et al. "Diffusion posterior sampling for general noisy inverse problems." arXiv preprint arXiv:2209.14687 (2022).

**Questions:**

1. This is the most critical question. How does ECI guarantee that the achieved solution satisfies the PDE system? In computer vision tasks where differences from ground truth images are permissible.  In this work, boundary conditions are guaranteed, but how does ECI ensure that the generated data truly represents solutions to the PDE? More specifically, why the linear interpolation between two PDE solutions is still a valid PDE solution?

2. Can you elaborate more on the stochasticity of generated solutions, as is discussed in Sec 3.3. Does more stocahsticity mean the generated distribution is distribution-wise closer to the uniform distribution over the space of constrained PDE solutions?
3. For the extrapolation step, will the algorithm performance be better if more a refined scheme is used? For example, mutiple step extrapolation when time $t$ is far from $1$ and one step when $t$ is close to $1$. From my understanding, the quality of predicted $\hat u_1$ is crucial to the algorithm performance. An additional ablation study could be performed here.
4. Can you comment on the computational efficiency of ECI compared with existing approaches?

The current score assumes a positive answer to the first question. My score will be adjusted accordingly based on the authors' response.

---

> ### Author Response · Authors · 2024-11-18
> **Rebuttal to bR4B (Part I)**
>
> We sincerely appreciate your recognition of our work's high readability and comprehensive experiments and ablation studies. We also thank your recognition of our ECI sampling as a generative model. We will clarify and address your questions as follows.
>
> ## Q1 Problem Setup
>
> We will further elaborate on our problem setup in more rigorous measure-theoretic terms as follows. Consider a well-posed PDE system $\mathcal{F}\_\phi u(x)=0, x\in\mathcal{X}\subseteq \mathbb{R}^D$ with PDE parameters $\phi\in\Phi$, then the solution map $\mathcal{T}:\phi\mapsto u(x)$ is a well-defined mapping. For a fixed distribution $p_\Phi$ over the parameter $\phi$, we can naturally define the measure over the solution set as the push-forward of the solution map: $\mu\_\mathcal{F}=\mathcal{T}\_* p\_\Phi$. Now consider some constraint operator $\mathcal{G}u(x)=0,x\in\mathcal{X_G}\subseteq \mathcal{X}$ defined on a subset of the PDE domain. Denote the solution set for $\mathcal{G}$ as $\mathcal{U_{G}}$ and the characteristic function over the solution set as $\chi_\mathcal{G}$. With the non-degenerating assumption that $\int_\mathcal{U} \chi_\mathcal{G}d\mu_\mathcal{F}>0$ (this is essentially saying that we should have at least one solution), the conditional measure can be obtained as $\mu_\mathcal{G}(A):=\mu_\mathcal{F}(A|\mathcal{G})=\int_A\chi_\mathcal{G}d\mu_\mathcal{F}/\int_\mathcal{U} \chi_\mathcal{G}d\mu_\mathcal{F}$. We assume $\mu_\mathcal{F}$ can be approximated well by generative priors like FFM, and we want to guide it towards the conditional measure $\mu_\mathcal{G}$.
>
> It is straightforward from the above measure-theoretic formulation that whether the measure $\mu_\mathcal{G}$ over the constrained solution set $\mathcal{U_{G}}$ is uniform **depends on the prior choice of the distributions over PDE parameters** $p_\Phi$. In our current experiments, all PDE parameters are sampled uniformly over a fixed interval (see Table 7 and Appendix B for details). This setting indicates the target measure over $\mathcal{U_{G}}$ is indeed uniform in our case. However, we can easily alter the parameter distribution, e.g., sampling some PDE parameters from the standard normal distribution instead. In this way, the target measure over the solution set is non-uniform but transforms accordingly with the push-forward.
>
> The problem setup can be understood more intuitively, as shown in the example of the Stokes problem demonstrated in Figure 5 in our updated manuscript. In the unconstrained pre-trained stage, the FFM is trained with **both variable BC and IC**. During zero-shot constrained generation, **either IC (middle) or BC is fixed (right) but not both of them**. In this way, solutions in the middle figure share the same left-column values while the variable BC still allows for a family of solutions. Similarly, solutions in the right figure share the same top-row values with variable IC as a family of solutions.
>
> ## Q2 ECI's Advantages and Comparisons
>
> We have briefly discussed the potential advantage of ECI sampling in Section 3.2 and we further provide a more systematic summary here.
>
> - Compared to existing correction algorithms for regression NOs (e.g., ProbConserv), ECI sampling offers **fine-grained iterative control** in the flow sampling process, leading to a more consistent generation. As demonstrated in our visualizations, directly applying the correction algorithm to the final generation will lead to noticeable artifacts. ECI sampling, on the other hand, can generate predictions with hardly noticeable artifacts.
> - Compared to existing gradient-based algorithms (e.g., DiffusionPDE, D-Flow), ECI sampling adopts a gradient-free approach that **ensures the exact satisfaction of the constraints** while enjoying a **significant speedup and memory saving** (see our common rebuttal). We also mentioned in Section 1 why gradient-based methods may fail in PDE scenarios.
> - Similar to some previous work (D-Flow, RePaint, DDRM), ECI sampling can be also **applied recursively** in a single sampling step, leading to better information mixing between the constrained and unconstrained regions. This is because the correction information is "propagated" back to the current timestep via interpolation, and multiple rounds of correction can potentially improve consistency with the constraints.
>
> These points are further supported by our extensive experiments over various 2D and 3D PDE systems, in which our ECI sampling was able to achieve state-of-the-art performance on most generative metrics.
>
> We already provide a fairly comprehensive related work in Section 2, where existing zero-shot guided generation models in computer vision domains are also included. An itemized comparison with the existing baselines is provided in Table 1, and further details regarding the baselines are provided in Appendix B.

---

> > ### Author Response · Authors · 2024-11-18
> > **Rebuttal to bR4B (Part II)**
> >
> > ## Q3 Validation of PDE System Satisfaction
> >
> > We do not guarantee the satisfaction of the PDE system for intermediate noised functions. In fact, this is the **standard and normal assumption** in all diffusion and flow matching models. As a more familiar example,  the diffusion model for image generation (e.g., Stable Diffusion) also adds noises to produce **intermediate noised images with little meaning**, and its standard point is standard Gaussian which certainly cannot be viewed as a valid image.
> >
> > The noising process in flow matching models is related to the conditional probability path from the prior noise measure to the target measure to facilitate generative modeling [1]. Specifically, the linear interpolation in our ECI step is the same as the training stage for FFM (the *OT-path* in [1] and FFM) where **the Gaussian process as the prior noises and ground truth solutions are interpolated** (instead of "two PDE solutions", as suggested in the review Q1). Such an interpolation also does not guarantee the satisfaction of the PDE system but is perfectly fine as the generative mode learns the continuous flow dynamics for intermediate noised data and can produce meaningful generation at timestep $t=1$.
> >
> > We further note that the exact satisfaction of the PDE system, on the other hand, is the goal for all machine learning models in the PDE domains and can never be exactly satisfied — if so, all regression neural operators do not need to exist as such an exact satisfaction of the PDE system provides the exact solution. Instead, **all current models seek to minimize such an error** (noticeably, the PINN loss). We assume the prior generative model like FFM already captures the important information of the PDE system with decent generations, and we expect our constrained generation to also respect the underlying PDE system. It can be demonstrated in our generation results in various figures that, compared with other baselines, our ECI sampling can produce results with much fewer artifacts, thus indicating a better consistency with the PDE system.
> >
> > ## Q4 Additional References
> >
> > We thank you for mentioning these two insightful papers that we missed. We will add them to our revised manuscript and we discuss the difference in our ECI sampling from these works.
> >
> > The SCG in the first paper is related to our setting in the sense that 1) it offers gradient-free control over potentially non-differentiable rules, 2) it also tries to estimate the clean sampling and apply the scoring rule. However, there are also some major differences from our ECI sampling. 1) Sampling for flow matching models is **deterministic once the initial noise is chosen**. This means we can not follow SCG to sample multiple plausible predictions and choose the best one. 2) The underlying theory of SCG comes from optimal control, whereas we adopt a variational approach to decompose the conditional measure with additional variables. In fact, we also noted some related work using gradient-free approaches in our original manuscript, such as RePaint and DDRM. However, they all rely on the stochastic nature of diffusion sampling. Adapting them to flow matching models is non-trivial and is not a major point of this work.
> >
> > The DPS paper is more classical. We first noted that **DPS is not gradient-free**. Instead, it relies on the gradient of some final loss function with respect to the current noise data. Although we did not directly cite the DPS paper, it is worth mentioning that the DiffusionPDE baseline in our paper, when the PINN loss is not available, reduces fully to a special case of DPS. Therefore, as we have already demonstrated the superior performance of ECI sampling over DiffusionPDE, it is safe to claim that we also outperform the vanilla DPS approach.
> >
> >
> >
> > ## Q5 Stochasticity
> >
> > Given the explanation in Q1 regarding the uniform distribution over the constrained solution set, your understanding of stochasticity in our context is mostly correct. We state such a claim more accurately in the following way: **less stochasticity means the distribution over the constrained generation is closer to that over the unconstrained generation**. In other words, if the constrained solution set is more different from the prior unconstrained solution set, more stochasticity is expected during the constrained generation to better capture the distribution shift, as we have suggested as a heuristic rule for choosing hyperparameters.
> >
> > We noted that, however, an excessively large number of mixing iterations (and thus very high stochasticity) may lead to failure cases with very large values. This is probably because the pre-trained FFM is not perfect, and high stochasticity may lead to regions never explored during training and thus highly inaccurate vector fields. Therefore, we often limit the number of mixing iterations below 10.

---

> > > ### Author Response · Authors · 2024-11-18
> > > **Rebuttal to bR4B (Part III)**
> > >
> > > ## Q6 Multiple Extrapolation Steps
> > >
> > > We thank you for pointing out such an insightful viewpoint of our framework. We carry out additional ablation studies on the effect of the number of extrapolation steps. As the Stokes BC results are already good, we did the ablation study with the Stokes IC and NS equation. The results are summarized in the table below. `#extrapolation` stands for the number of steps when we do extrapolation, with 1 equivalent to our current setting (i.e., one-step extrapolation). It can be demonstrated that the impact of this is minor, if not negative on the final performance. It is probably because our iterative framework can already offer fine-grained control over the sampling process, so the extrapolation stage itself does not impact the final performance. This can be also demonstrated in Figure 4, left, where the extrapolation already achieves decent prediction at a timestep close to zero.
> > >
> > > | Dataset | Stokes IC | Stokes IC | Stokes IC | NS Equation | NS Equation | NS Equation |
> > > |---|---|---|---|---|---|---|
> > > | #extrapolation | 1 | 2 | 3 | 1 | 2 | 3 |
> > > | MMSE / $10^{-2}$ | 0.090 | 0.126 | 0.093 | 7.961 | 8.345 | 10.739 |
> > > | SMSE / $10^{-2}$ | 0.127 | 0.108 | 0.150 | 5.846 | 7.138 | 7.296 |
> > > | FPD | 0.075 | 0.048 | 0.103 | 1.131 | 1.787 | 2.024 |
> > >
> > >
> > > ## Q7 Computational Efficiency
> > >
> > > We provide a more comprehensive evaluation of our method's computational efficiency in terms of sampling time and GPU memory and additional experiments on higher-resolution data in our **general rebuttal**. It can be demonstrated in the table that our ECI sampling has **little time and memory overhead** over the original flow sampling scheme. Furthermore, we noted a tremendous **x440 speedup and x330 memory saving** compared to D-Flow on the Stokes problem. Such a tremendous computational efficiency benefit can be attributed to our gradient-free design of the framework, making it more scalable to larger datasets where gradient-based methods will have memory issues.
> > >
> > > [1] Lipman, Yaron, et al. "Flow matching for generative modeling." *arXiv preprint arXiv:2210.02747* (2022).

---

> > > > ### Comment · Reviewer_bR4B · 2024-11-19
> > > > **Some Clarifications**
> > > >
> > > > Thank you for your response.
> > > >
> > > > There are a few things I would like to clarify:
> > > >
> > > > 1. The statement that flow matching is a deterministic model, and therefore the technique in [1] cannot be applied, is incorrect. I also noticed that you made a similar claim to another reviewer. In fact, diffusion models and flow matching can be transformed into each other using equation 8 in [2] and re-paratermization of the score/flow function. I do not expect any comparison with [1] due to the limited time during the rebuttal phase, but I hope you include this clarification in the paper to avoid misleading readers who are not familar with generative models.
> > > >
> > > > 2. Regarding the most important aspect—the correctness of the PDE solution—I am fully convinced by the numerical results you provided, although I am not an expert in this area. However, it is essential to mention that the work cannot guarantee the correctness of the generated solutions. I understand that prior work also lacks this guarantee, but I believe it's critical to start addressing this issue. This is particularly important for readers who are experts in the scientific machine learning (SciML) area and may not be familiar with generative models.
> > > >
> > > > Therefore, I will maintain my score of 6, and I strongly hope that you add the two aforementioned clarifications in the main paper to assist audiences in the generative modeling (1) and scientific domains (2).
> > > >
> > > > [1] Yujia Huang, Adishree Ghatare, Yuanzhe Liu, Ziniu Hu, Qinsheng Zhang, Chandramouli S Sastry, Siddharth Gururani, Sageev Oore, and Yisong Yue. Symbolic music generation with nondifferentiable rule guided diffusion. arXiv preprint arXiv:2402.14285, 2024.
> > > >
> > > > [2]Ma, Nanye, et al. "Sit: Exploring flow and diffusion-based generative models with scalable interpolant transformers." arXiv preprint arXiv:2401.08740 (2024).

---

> > > > > ### Author Response · Authors · 2024-11-19
> > > > > **Response to bR4B**
> > > > >
> > > > > We sincerely thank you prompt response and your willingness to engage in a productive dialogue. We further comment on your clarifications.
> > > > >
> > > > > Regarding the connection between flow and diffusion models, we completely agree that they can be unified under one framework. In fact, we already mentioned the stochastic interpolant work [1] in our original manuscript, which is also the basis for the SiT model you mentioned. The reason why we believe the existing work cannot be adopted for flow matching is that we **confine ourselves to the classical deterministic flow ODE sampling instead of considering the stochastic differential equation (SDE)** formulation. Therefore, based on such an assumption, the SCG work you mentioned cannot be adapted for the standard flow ODE, as it requires sampling several different next steps and picking up the best one. Of course, this is still possible if we adopt the SDE formulation in the stochastic interpolant work. We thank you for pointing out such a connection.
> > > > >
> > > > > Regarding the correctness of the PDE system, we fully agree that it is indeed the central problem in the PDE domain of neural operator learning. We will follow your suggestions to add more clarifications in our final revision in this regard to make our work more accessible to a larger audience.
> > > > >
> > > > >
> > > > >
> > > > > [1] Albergo, Michael S., Nicholas M. Boffi, and Eric Vanden-Eijnden. "Stochastic interpolants: A unifying framework for flows and diffusions." *arXiv preprint arXiv:2303.08797* (2023).

---

### Official Review · Reviewer_MQJx · 2024-11-14

**Soundness:** 3
**Presentation:** 2
**Contribution:** 2
**Rating:** 6
**Confidence:** 3

**Summary:**

In this work, the authors propose ECI for gradient-free generation for constraint systems. ECI is zero-shot and doesn't need tuning the model. It works through alternating between extrapolation (E), correction (C), and interpolation (I) in each sample step. The empirical results show the model achieves competitive performance across various PDE systems compared with baseline models.

**Strengths:**

1. The work is well-motivated, it's important in scientific applications to have constraint generation
2. The proposed method is effective and does have constrained generation. Also, the method is zero-shot which is a big benefit for constraint sampling method. Although it's based on functional FM not FM in general
3. The experimental results show advantages over other baseline models.

**Weaknesses:**

1. The writing is also a bit confusing. Actually comments from reviewer KM5e help me better understand the algorithm 2.
2. The authors are recommended to better formulate the contribution of this work. In particular, the model works in functional space and applies projection to constraint spaces to guarantee hard-constraint met.
3. The authors mention supply chain optimization in abstract and introduction as a hard-constrained system. However, it lacks experiments on such problems. It would be better to showcase some applications beyond PDE learning (which I believe is broad).

**Questions:**

1. How does proposed ECI compared with FFM without ECI in experiments? It would better show the improvement from ECI in sampling.
2. In section 3.3, the authors mention resampling noise during sampling. Can the authors explain why this helps better generative results?
3. The model assumes a perfectly trained FFM, however there is inevitable error in the trained generative model. How does such error affect the performance? Will such error break the constraint in sampling?
4. Empirical results are on low-resolution problems. Can the authors comment on how ECI work at larger scale?

---

> ### Author Response · Authors · 2024-11-18
> **Rebuttal to MQJx (Part I)**
>
> We appreciate your recognition of our work's **clear motivation** in scientific domains, the benefit of our **zero-shot** formulation, and **competitive performance**. We will address your questions and concerns as follows.
>
> ## Q1 ECI Algorithm
>
> We thank your suggestion. As we have mentioned in our **general rebuttal**, we have updated the revised manuscript with clearer definitions of notations, an additional description of the *ECI step* in Algorithm 2, and more explanations to better describe our algorithm in a more concise and accurate way.
>
> ## Q2 Contribution
>
> We formulate our contributions more clearly in the following sense:
>
> - We proposed ECI sampling for hard-constrained generation for pre-trained **flow-based generative models**. We emphasize *flow-based generative models*, which impose unique challenges due to their iterative sampling nature. Existing approaches for *constrained prediction* in regression models are infeasible or perform badly. To the best of our knowledge, we are the first to deal with the zero-shot flow-based constrained generation problems for functional data.
> - Our proposed ECI sampling is an **iterative control framework** unique to flow-based models using the interleaved ECI steps. Such a fine-grained iterative control better facilitates information flow between the constrained and the unconstrained regions, leading to a more consistent generation with fewer artifacts.
> - Our approach guarantees the exact satisfaction of constraints **without the expensive gradient information**. Such a gradient-free formulation achieves significant speedup and memory-saving compared to existing gradient-based approaches. In this way, our framework enjoys easy scalability with little time or memory overhead.
> - Our approach achieves **state-of-the-art generative performance** on most evaluation metrics for zero-shot constrained generation. We emphasize *generative evaluation metrics*, for which distributional properties are calculated and the foundation model Poseidon is used to give the most comprehensive metric of FPD.
>
> The changes are also reflected in our revised manuscript.
>
> ## Q3 Supply Chain Applications
>
> We noted that our framework, in principle, can be applied to different functional data including those in supply chain applications. However, we also noted similar data are scarce in such a domain, so we instead instantiate our approach on PDE domains where more well-established data are available and baseline-constrained generation models are also available. We will leave additional applications in other scientific domains as future work and **we have carefully modified our manuscript to avoid overclaiming our contributions**.
>
>
> ## Q4 Vanilla FFM Performance
>
> As a sanity check, we provide the generative metrics of the pre-trained FFM without any guidance in the table below. It can be demonstrated that there is indeed a shift in the distribution from the generative prior learned by the unconstrained FFM, as many errors are large. Our proposed ECI sampling can successfully capture such a shift in the distribution with significantly lower MSEs and a closer resemblance to the ground truth constrained distribution.
>
> | Dataset | MMSE / $10^{-2}$ | SMSE / $10^{-2}$ | CE / $10^{-2}$ | FPD |
> |---|---|---|---|---|
> | Stokes IC | 2.068 | 1.572 | 9.792 | 13.342 |
> | Stokes BC | 3.695 | 3.320 | 177.297 | 2.824 |
> | Heat Equation | 3.954 | 4.737 | 49.296 | 2.799 |
> | Darcy Flow | 58.879 | 69.432 | $7.48\times 10^4$ | 189 |
> | NS Equation | 18.749 | 8.423 | 11.451 | 2.185 |
>
> ## Q5 Resampling & Stochasticity
>
> We further elaborate on the effect of the resampling interval length and its relationship to sampling stochasticity. We first noted that the **"magnitude" of the distribution shift is different** for different constraints. This can be also understood by the results in Q4, where tasks like Stokes IC have smaller original FPDs, but tasks like Stokes BC have larger ones. We noted that such a property is intrinsic to the constraint. Therefore, it might be difficult for a single framework to deal with different magnitudes of distribution shifts simultaneously. In this sense, stochasticity during our constrained sampling should be larger for tasks with larger distribution shifts to guide the model toward further exploration, and vice versa.
>
> With this observation, we introduced the length of the resampling interval as **a tunable hyperparameter to offer control over such stochasticity** for each individual task. In this way, when the stochasticity matches those in the task, the model is easier to be guided toward the shifted distribution with more consistency and thus better generation results. We also provided a discussion of some heuristic rules for choosing this value in Section 3.3. However, a more theoretical analysis is always preferred, as we also mentioned as our limitation in Section 5.

---

> > ### Author Response · Authors · 2024-11-18
> > **Rebuttal to MQJx (Part II)**
> >
> > ## Q6 Assumptions on Pre-Trained FFM
> >
> > We thank you for your insightful questions regarding the quality of the pre-trained FFM. Intuitively, the quality or the expressiveness of the pre-trained model does have an impact on the final performance. To further explore this issue, we first provide MMSE and SMSE between the generation by the pre-trained unconstrained FFM model and the test dataset as the evaluation metric for our pre-training stage.
> >
> > | Dataset | Stokes Problem | Heat Equation | Darcy Flow | NS Equation |
> > |---|---|---|---|---|
> > | MMSE / $10^{-2}$ | 0.045 | 0.130 | 5.421 | 0.992 |
> > | SMSE / $10^{-2}$ | 0.039 | 0.198 | 4.121 | 0.371 |
> >
> > It can be seen that most FFMs are able to achieve a **decent approximation of the prior unconstrained distribution** over the solution set with small MMSE and SMSE. For the 2D Darcy flow, the boundary condition has a greater impact on the final solution such that all pixel values should be shifted by the same value, leading to a larger error in the distributional properties. The generation, on the other hand, does seem reasonable.
> >
> > We then provide additional evaluation metrics on the intermediate model checkpoints using the same hyperparameter set on the Stokes IC dataset to test the impact of the quality of the generative prior. Intuitively, a stronger generative prior that better captures the underlying information of the PDE system can be easier to guide. The results are demonstrated below, with checkpoints from 5k, 10k, and 20k training iterations. A clear trend of better guided-generation performance can be observed in both IC and BC constraints. In this way, the quality of FFM indeed plays an important role in our ECI sampling framework.
> >
> > | Metric | Stokes IC | Stokes IC | Stokes IC | Stokes BC | Stokes BC | Stokes BC |
> > |---|---|---|---|---|---|---|
> > | #iter | 10k | 15k | 20k | 10k | 15k | 20k |
> > | MMSE / $10^{-2}$ | 1.273 | 0.601 | 0.090 | 0.888 | 0.011 | 0.005 |
> > | SMSE / $10^{-2}$ | 1.693 | 0.177 | 0.127 | 0.868 | 0.015 | 0.003 |
> > | CE / $10^{-2}$ | 0 | 0 | 0 | 0 | 0 | 0 |
> > | FPD | 1.863 | 1.031 | 0.076 | 3.101 | 0.009 | 0.010 |
> >
> >
> > ## Q7 Scalability
> >
> > See our **general rebuttal** for additional experimental results on higher-resolution data and the analysis on time and memory consumption. With this evidence, we believe our ECI framework enjoys better scalability than other baselines.

---

> > > ### Comment · Reviewer_MQJx · 2024-11-25
> > >
> > > I thank the authors for the response and adding additional results. The response and updated manuscript help clarify my concerns. I would suggest the authors adding vanilla FFM performance to the main table to help readers better understand the improvement. Overall I stay positive about this paper.

---

### Author Response · Authors · 2024-11-13
**Rebuttal to all reviewers**

Dear Reviewers,

We sincerely appreciate your reviews that help make our work more concrete and clear. We thank you for pointing out the strengths in our work. Sadly, we note a common **misunderstanding and misinterpretation among all the reviewers** and we would first like to clarify this.

### **Our proposed ECI sampling framework is NOT a regression model** (e.g., FNO [1]), **nor a plugin for existing regression models** (e.g., BOON [2]). **Instead, ECI sampling is a generative framework for functional data based on flow matching.**

We believe all reviewers have misinterpreted this setting and have asked for a comparison with regression models or using regression metrics, which are not applicable to our problem setup. We do apologize for our assumption that this paper's audience is familiar enough with generative models, especially flow matching. We now add a more detailed introduction to flow matching here to make our paper more accessible to the audience in the SciML domains.

## Flow Matching

Flow matching [3] is a generative framework built on continuous normalizing flows. Flow matching tries to learn the time-dependent *vector field* $v_t:\mathbb{R}^d\times [0,1]\to\mathbb{R}^d$ that defines a continuous time-dependent diffeomorphism called the flow $\psi_t:\mathbb{R}^d\times [0,1]\to\mathbb{R}^d$ via the following *flow ODE*:
$$
\frac{\partial}{\partial t}\psi_t(x_0)=v_t(\psi_t(x_0)),\quad x_0\sim p_0(x)
$$
The flow induces a probability path with the push-forward $p_t=(\psi_t)_*p_0$ for generative modeling. [3] demonstrated the conditional vector field $u_t(x|x_1)$ can be calculated simulation-free while sharing the same gradient with the flow matching objective. In short, flow matching models try to learn the conditional vector field which defines the dynamics of the flow and the probability path.

[3] further demonstrated the conditional probability path can be manually defined with linear interpolation between the noise and target (the *OT-path*). [4] further extended flow matching to Riemannian manifolds where the interpolation has a more intuitive interpretation of geodesic interpolation. Either way, it is clear that **in all flow-matching literature, extrapolation and interpolation have well-defined meanings with respect to the probability path induced by the flow**.

Based on the above observation, we can clearly summarize the difference between regression models (neural operators) and our generative framework as follows:

- ECI sampling, as a flow-based generative framework, can **output a distribution of solutions**. In contrast, any regression model (neural operator) can only output one deterministic prediction.
- As a special case of the flow matching model, ECI sampling is **an iterative process that gradually *denoises*** the current noised samples into meaningful ones by solving the flow ODE above using the Euler method or any ODE solvers. See Algorithm 1 and Figure 4 for demonstration. In contrast, neural operators directly output the final prediction.
- The **learning objective of flow matching is completely different** from neural operators. The former tries to minimize the vector field discrepancy whereas the latter directly predicts the final outcome.
- The generative nature requires **generative evaluation metrics instead of regression metrics**. As both the output and the ground truth are a set of solutions instead of a single solution, distributional properties should be compared as generative metrics.

We further noted that the **iterative nature** of diffusion and flow-based models makes it **non-trivial to guide**, as intermediate steps are noises instead of final predictions. Indeed, we have included existing constrained generation and inverse problem works in computer vision in the related work. However, the continuous nature of flow matching makes their approaches not directly applicable.

We understand the reviewers may not be familiar with the generative domains. As our work lies in the intersection between scientific ML and generative models, we urge the reviewers to **take the contributions in generative modeling of our work — which is our major contribution — into consideration** to give a more comprehensive evaluation of this work.

Warm regards,

Authors.

[1] Li, Zongyi, et al. "Fourier neural operator for parametric partial differential equations." *arXiv preprint arXiv:2010.08895* (2020).

[2] Saad, Nadim, et al. "Guiding continuous operator learning through physics-based boundary constraints." *arXiv preprint arXiv:2212.07477* (2022).

[3] Lipman, Yaron, et al. "Flow matching for generative modeling." *arXiv preprint arXiv:2210.02747* (2022).

[4] Chen, Ricky TQ, and Yaron Lipman. "Riemannian flow matching on general geometries." *arXiv preprint arXiv:2302.03660* (2023).

---

### Author Response · Authors · 2024-11-18
**Updated Manuscript and General Rebuttal (Part I)**

First, we sincerely thank the Area Chair (AC) for accommodating our request to include additional emergency reviewers from the generative modeling domain, ensuring a fairer and more comprehensive evaluation of our work as a generative model. We also extend our gratitude to all the reviewers for their insightful questions. Special thanks go to the two emergency reviewers who dedicated extra time to provide timely reviews.

To incorporate the feedback, add additional experiments, and improve the presentation, we have updated the manuscript with a revised version. The major changes are summarized as follows:

- **Clearer problem setup, algorithm presentation, and notation definition**. We revise our problem setup description in Section 3.1 and provide a mathematically rigorous definition in Appendix A.1. We extract another algorithm for which we term *ECI Step* (Algorithm 2) and also modify our ECI sampling in Algorithm 3 to make it more concise with clearer notation definitions. A visual demonstration of the problem setup is also provided in Figure 5 to give a more intuitive interpretation.
- **Additional experimental results, visualizations, and ablation studies**. These include
  - A more comprehensive evaluation of **time and memory consumption** analysis in Table 4.
  - Visually more intuitive plots of **errors of the statistics** in Figure 1, 2, 10, 12, 14.
  - Examples of the PDE systems and problem setup in Figure 5, 6, 7.
  - Evaluation of the **quality of the pre-trained FFM model** in Table 8, 9, as requested by MQJx.
  - Additional generative results on the Stokes problem with **different constraints** in Table 10, as requested by KM5e.
  - Additional generative results on **higher-resolution data** in Table 11 to demonstrate scalability.
  - Additional ablation studies on the **impact of the quality of FFM** in Table 12, as requested by MQJx.
- **Contributions more focused on the generative domain**. We now highlight our contributions as a generative framework for guiding pre-trained generative models for constrained generation. More detailed mathematical background on flow matching and FFM is also provided in Appendix C.1 and C.2 to make our work more self-contained as a generative model.

We will now use the new indices for tables and figures in the revised manuscript. We also address the common concerns for the scalability of our framework here.  Other questions and concerns will be addressed in separate rebuttals.

---

> ### Author Response · Authors · 2024-11-18
> **Updated Manuscript and General Rebuttal (Part II)**
>
> ## Scalability
>
> We first noted that all datasets and their spatiotemporal resolution setups are directly **taken from previous papers** (see Appendix B for reference of the datasets). Therefore, we believe such an experimental setup is **standard, consistent, and comparable with existing works**. It is also a standard practice to test on 2D and 3D datasets in the PDE domain, e.g., neural operator learning [1, 2], foundation models [3, 4], and existing constrained generation [5, 6]. Therefore, we believe we have already provided enough evidence of the superior performance of our proposed method. Furthermore, as our proposed ECI sampling is model-agnostic, **the time and memory bottleneck lie with the vector field encoder**, which is parameterized by some neural operator. In this work, we used the Fourier neural operator, but any other new encoder architecture can be readily integrated within our framework to further boost the scalability of the overall framework.
>
> We further provide extensive benchmarking of the sampling time and GPU memory usage of different sampling methods on the 2D (Stokes IC) and 3D (NS equation) datasets in the table below. Unfavorable values are highlighted in bold. Compared with gradient-based methods, our gradient-free approach enjoys sampling efficiency in both 2D and 3D cases with **little overhead** compared to the original flow sampling. Noticeably, ECI-1 was able to achieve **x440 acceleration and x310 memory saving** compared to gradient-based D-Flow on the Stokes problem. Such a significant advantage of computation and memory efficiency ensures the scalability of our proposed ECI sampling to larger datasets or data points where the gradient-based approach may fail.
>
> | Dataset | Resource | ECI-1 | ECI-5 | CondFFM | ANP | ProbConserv | DiffusionPDE | D-Flow |
> |---|---|---|---|---|---|---|---|---|
> | Stokes IC | \#sample/\#Euler | 128/200 | 128/200 | 128/200 | 32/NA | 128/200 | 128/200 | **2**/200 |
> |  | Time/sample/s | 0.065 | 0.325 | 0.057 | 0.009 | 0.058 | 0.131 | **28.774** |
> |  | GPU Memory/GB | 5.4 | 5.4 | 5.4 | 7.4 | 5.4 | 10.8 | **26.4** |
> | NS equation | \#sample/\#Euler | 25/100 | 25/100 | 25/100 | 0.2/NA | 25/100 | 25/100 | **1/20** |
> |  | Time/sample/s | 0.415 | 2.067 | 0.669 | 2.324 | 0.675 | 0.676 | **8.456** |
> |  | GPU Memory/GB | 16.3 | 16.3 | 16.3 | 11.0 | 16.3 | **27.0** | **27.1** |
>
> To further provide a more concrete example of scaling up to higher resolutions, we adopt the zero-shot superresolution setting to test all of the methods with a generation resolution of 200×200 on the Stokes problem. The results are summarized in the following table:
>
> | Dataset | Metric | ECI | ANP | ProbConserv | DiffusionPDE | D-Flow | CondFFM |
> |---|---|---|---|---|---|---|---|
> | Stokes IC | MMSE / $10^{-2}$ | **0.274** | 14.120 | 4.338 | 5.109 | OOM | 4.242 |
> |  | SMSE / $10^{-2}$ | **0.515** | 8.720 | 2.933 | 4.167 |  | 4.152 |
> |  | CE / $10^{-2}$ | **0** | 3.563 | **0** | 5.392 |  | 12.508 |
> |  | FPD | **2.647** | 26.965 | 15.977 | 19.654 |  | 24.798 |
> | Stokes BC | MMSE / $10^{-2}$ | **0.863** | 2.819 | 5.223 | 1.610 | OOM | 4.669 |
> |  | SMSE / $10^{-2}$ | **0.418** | 2.870 | 1.491 | 1.768 |  | 2.706 |
> |  | CE / $10^{-2}$ | **0** | 29.168 | **0** | 0.068 |  | 21.567 |
> |  | FPD | **1.903** | 5.282 | 2.021 | 3.830 |  | 9.504 |
>
> **D-Flow runs out of memory even with a batch size of 1**. On the other hand, our ECI sampling continues to achieve the best generation results, though slightly worse than its 100×100 resolution counterpart. It is also interesting to see that the conditional FFM model is the most sensitive approach, probably because it has hard-coded the resolution information for a better generation performance during training. In this way,  we can safely conclude that **our framework demonstrates better scalability on higher-resolution data** than gradient-based baselines.
>
>
>
> [1] Li, Zongyi, et al. "Fourier neural operator for parametric partial differential equations." *arXiv preprint arXiv:2010.08895* (2020).
>
> [2] Li, Zongyi, et al. "Anima Anandkumar, Physics-informed neural operator for learning partial differential equations." *arXiv preprint arXiv:2111.03794* (2021).
>
> [3] Herde, Maximilian, et al. "Poseidon: Efficient Foundation Models for PDEs." *arXiv preprint arXiv:2405.19101* (2024).
>
> [4] Sun, Jingmin, et al. "Towards a Foundation Model for Partial Differential Equation: Multi-Operator Learning and Extrapolation." *arXiv preprint arXiv:2404.12355* (2024).
>
> [5] Huang, Jiahe, et al. "DiffusionPDE: Generative PDE-solving under partial observation." *arXiv preprint arXiv:2406.17763* (2024).
>
> [6] Négiar, Geoffrey, Michael W. Mahoney, and Aditi S. Krishnapriyan. "Learning differentiable solvers for systems with hard constraints." *arXiv preprint arXiv:2207.08675* (2022).

---

### Author Response · Authors · 2024-12-03
**Thank you for your valuable reviews and response**

Dear Reviewers,

We are sincerely grateful for your reviews and active engagement in the discussion period, which have been valuable and instrumental in improving the comprehensiveness and conciseness of our work. We are also grateful that our clarifications have successfully resolved most existing concerns regarding the contributions of our proposed approach in the generative domain. Our contributions in the generative domain have been generally acknowledged by all reviewers now, especially the two emergency reviewers from the generative domains, which has been highly encouraging for us.

We genuinely thank all the reviewers for their insightful suggestions in improving the clarity of our work for a larger audience with diverse backgrounds. We have followed your detailed suggestion to adopt a more concise notation definition, a clearer algorithm description, and a more focused delivery of our contributions in the generative domain. Modifications have been reflected in the revised manuscript, and we are committed to further polishing our work.

We also sincerely thank the AC for their help in finding emergency reviewers for a more comprehensive evaluation of our work in the generative domain and the emergency reviewers for their additional efforts in reviewing our work.

Best regards,

Authors.

---

### Meta-Review · Area_Chair_dcKb · 2024-12-19

**Metareview:**

This paper addresses inverse problems common to scientific data (and potentially other areas like manufacturing) where the solution to the PDE must adhere to hard constraints specified. The main contribution is to introduce a method to constrain samples produced by a generative (flow-based) model so that they adhere to specified boundary conditions and initial values. The method does not require expensive gradient computations or model tuning and can be run zero-shot both for generative and regression tasks. The proposed ECI approach is reminiscent of some non-generative constraint satisfaction methods, but is unique in that it is formulated specifically for a generative approach. The paper presents thorough zero-shot evaluations of their method against published baselines for generative and regression tasks on standard scientific PDE benchmarks, demonstrating competitive performance on most tasks.

Strengths:
The paper is written well and the problem is well motivated, which should be of interest particularly for ML for science applications. In addition, the approach does work as intended to satisfy hard constraints during generation. The paper demonstrates that constraint satisfaction, even without gradient information, can operate well in a zero-shot setting, which may have efficiency value. Results are comprehensive on standard scientific/PDE benchmarks serve to validate the method, both for generative and regression settings. It’s also nice to see a thorough ablation study of modeling choices, and the runtime analysis showing the method is efficient. In theory, this work should also be relevant beyond ML for science to any areas that require hard constraint satisfaction (e.g., design and manufacturing.)

Weaknesses:
Theoretical aspects of the paper could be strengthened, including ensuring readers know that the ECI method still does not guarantee correctness at t=0, as well as the motivation for specifically following a flow-matching ODE formulation for constraint satisfaction (as opposed to non-generative methods or pursuing adapting an SDE formulation as pointed out by one reviewer). The paper could benefit from experiments beyond PDEs, potentially even including CV examples (which may not benefit from the method but would still be interesting to compare). Novelty appears to reviewers to be limited, due to the ECI method bearing superficial similarity to existing methods for constraint satisfaction.

Decision reasoning:
While I believe the novelty of the ECI formulation is somewhat limited, and that non flow-based formulations could potentially be adapted in a similar way, I think the contribution is solid and of practical value to the community (in particular, the relaxation of the need for gradient information and the ensuing efficiency). The experiments are comprehensive and demonstrate the effectiveness of the method — even though it is not guaranteed to always result in hard constraint satisfaction (though this is true of previous guidance/satisfaction methods in the literature as well). The authors did a good job incorporating feedback from reviewers, which led to better highlighting of the contributions, reduction of unsupported claims, and more presentation clarity. Therefore I recommend acceptance.

**Additional Comments On Reviewer Discussion:**

While I find it a bit troubling that the original 3 reviewers were not fully convinced by the rebuttal, I do appreciate that they engaged in dialogue with authors and all expressed some positivity after the rebuttal — two of them revised their scores upward accordingly, indicating that some conerns were addressed. I also found it a bit strange that the authors quickly claimed the original 3 reviewers misunderstood their work — but I was glad to see constructive dialogue ensure, which I believe enhanced the presentation of the paper in the end. In the end, reviewers are on the fence — even the two emergency reviewers who are experts in generative models are only borderline accept and did not revise their scores upward based on the rebuttal. From my own reading, I do not dispute the main concerns of the reviewers about limited novelty and need for further theoretical strengthening. However, I also feel that the

---

### Decision · Program_Chairs · 2025-01-22

Accept (Poster)